# Evolution of the Family Equidae, Subfamily Equinae, in North, Central and South America, Eurasia and Africa during the Plio-Pleistocene

**DOI:** 10.3390/biology11091258

**Published:** 2022-08-24

**Authors:** Omar Cirilli, Helena Machado, Joaquin Arroyo-Cabrales, Christina I. Barrón-Ortiz, Edward Davis, Christopher N. Jass, Advait M. Jukar, Zoe Landry, Alejandro H. Marín-Leyva, Luca Pandolfi, Diana Pushkina, Lorenzo Rook, Juha Saarinen, Eric Scott, Gina Semprebon, Flavia Strani, Natalia A. Villavicencio, Ferhat Kaya, Raymond L. Bernor

**Affiliations:** 1Laboratory of Evolutionary Biology, Department of Anatomy, College of Medicine, Howard University, Washington, DC 20059, USA; 2Earth Science Department, Paleo[Fab]Lab, University of Florence, Via La Pira 4, 50121 Firenze, Italy; 3Earth Sciences Department, University of Oregon, 100 Cascade Hall, Eugene, OR 97403, USA; 4Instituto Nacional de Antropología e Historia Laboratorio de Arqueozoología “M. en C. Ticul Álvarez Solórzano”, Subdirección de Laboratorios y Apoyo Académico, Ciudad de Mexico 00810, Mexico; 5Quaternary Palaeontology Program, Royal Alberta Museum, 9810 103a Ave NW, Edmonton, AB T5J 0G2, Canada; 6Clark Honors College, University of Oregon, Eugene, OR 97401, USA; 7Department of Geosciences, University of Arizona, 1040 E 4th St., Tucson, AZ 85721, USA; 8Department of Paleobiology, National Museum of Natural History, Smithsonian Institution, 10th St and Constitution Ave NW, Washington, DC 20013, USA; 9Division of Vertebrate Paleontology, Yale Peabody Museum of Natural History, 170 Whitney Ave, New Haven, CT 06520, USA; 10Department of Earth Sciences, University of Ottawa, 25 Templeton Street, Ottawa, ON K1N 6N5, Canada; 11Laboratorio de Paleontología, Facultad de Biología, Universidad Michoacana de San Nicolás de Hidalgo, Edif. R 2. Piso. Ciudad Universitaria, Morelia 58030, Mexico; 12Department of Science, University of Basilicata, Via dell’Ateneo Lucano 10, 85100 Potenza, Italy; 13Department of Geosciences and Geography, University of Helsinki, FI-00014 Helsinki, Finland; 14Cogstone Resource Management, Inc., 1518 W., Taft Avenue, Orange, CA 92865, USA; 15Department of Biology, California State University, 5500 University Parkway, San Bernardino, CA 92407, USA; 16Department of Biology, Bay Path University, 588 Longmeadow Street, Longmeadow, MA 01106, USA; 17PaleoFactory, Dipartimento di Scienze della Terra, Sapienza Università di Roma, Piazzale Aldo Moro 5, 00185 Rome, Italy; 18Departamento de Ciencias de La Tierra, Instituto Universitario de Investigación en Ciencias Ambientales de Aragón (IUCA), Universidad de Zaragoza, 50009 Zaragoza, Spain; 19Corporación Laguna de Taguatagua, Av. Libertador Bernardo O’Higgins 351, Santiago 1030000, Chile; 20Instituto de Ciencias Sociales, Universidad de O’Higgins, Av., Libertador Bernardo O’Higgins 611, Rancagua 2852046, Chile; 21Biomolecular Laboratory, Institut Català de Paleoecologia Humana i Evolucio Social, 43007 Tarragona, Spain; 22Department of Archaeology, University of Oulu, FI-90014 Oulu, Finland; 23Human Origins Program, Department of Anthropology, National Museum of Natural History, Smithsonian Institution, Washington, DC 20013, USA

**Keywords:** Equidae, Equinae, hipparionini, protohippini, equini, paleoecology, paleoclimatology, biochronology, phylogeny, evolution

## Abstract

**Simple Summary:**

The family Equidae enjoys an iconic evolutionary record, especially the genus *Equus* which is actively investigated by both paleontologists and molecular biologists. Nevertheless, a comprehensive evolutionary framework for *Equus* across its geographic range, including North, Central and South America, Eurasia and Africa, is long overdue. Herein, we provide an updated taxonomic framework so as to develop its biochronologic and biogeographic frameworks that lead to well-resolved paleoecologic, paleoclimatic and phylogenetic interpretations. We present *Equus*’ evolutionary framework in direct comparison to more archaic lineages of Equidae that coexisted but progressively declined over time alongside evolving *Equus* species. We show the varying correlations between body size, and we use paleoclimatic map reconstructions to show the environmental changes accompanying taxonomic distribution across *Equus* geographic and chronologic ranges. We present the two most recent phylogenetic hypotheses on the evolution of the genus *Equus* using osteological characters and address parallel molecular studies.

**Abstract:**

Studies of horse evolution arose during the middle of the 19th century, and several hypotheses have been proposed for their taxonomy, paleobiogeography, paleoecology and evolution. The present contribution represents a collaboration of 19 multinational experts with the goal of providing an updated summary of Pliocene and Pleistocene North, Central and South American, Eurasian and African horses. At the present time, we recognize 114 valid species across these continents, plus 4 North African species in need of further investigation. Our biochronology and biogeography sections integrate Equinae taxonomic records with their chronologic and geographic ranges recognizing regional biochronologic frameworks. The paleoecology section provides insights into paleobotany and diet utilizing both the mesowear and light microscopic methods, along with calculation of body masses. We provide a temporal sequence of maps that render paleoclimatic conditions across these continents integrated with Equinae occurrences. These records reveal a succession of extinctions of primitive lineages and the rise and diversification of more modern taxa. Two recent morphological-based cladistic analyses are presented here as competing hypotheses, with reference to molecular-based phylogenies. Our contribution represents a state-of-the art understanding of Plio-Pleistocene *Equus* evolution, their biochronologic and biogeographic background and paleoecological and paleoclimatic contexts.

## 1. Introduction

Studies on the evolution of the family Equidae started in the middle of the 19th century following the opening of the western interior of the United States. Marsh [1] produced an early orthogenetic scheme of Cenozoic horse evolution detailing changes in the limb skeleton and cheek teeth. Gidley [2] challenged the purported orthogonal evolution of horses with his own interpretation of equid evolution. Osborn [3] chose not to openly debate the phylogeny of Equidae but rather displayed Cenozoic horse diversity in his 1918 treatise on the unparalleled American Museum of Natural History’s collection of fossil equids. Matthew [4], however, did produce a phylogeny detailing 10 stages, actual morphological grades, ascending from *Eohippus* to *Equus.* Stirton [5] provided a widely accepted augmentation of Matthew’s earlier work with his revised orthogonal scheme of North American Cenozoic Equidae. Simpson [6] published his book on horses, which was the most authoritative account up to that time. His scheme was vertical rather than horizontal in its view of North American equid evolution, with limited attention paid to the extension of North American taxa into Eurasia and Africa. MacFadden [7] updated in a significant way Simpson’s [6] book, depicting the phylogeny, geographic distribution, diet and body sizes for the family Equidae.

This work principally focused on the evolution of *Equus* and its close relatives in North and South America, Eurasia and Africa during the Plio-Pleistocene (5.3 Ma–10 ka). We documented several American lineages that overlap *Equus* in this time range, whereas in Eurasia and Africa, only hipparionine horses co-occurred with *Equus* beginning at ca. 2.6 Ma, which we included in this work for their biogeographical and paleoecological significances. These taxa were reviewed by MacFadden [7] and Bernor et al. [8] including the references therein. 

In the present manuscript, we aimed to revise and discuss the most recent knowledge on the Equinae fossil record, with the following goals:(1)Provide a systematic revision of tridactyl and monodactyl horses from 5.3 Ma to 10 ka. These taxa are discussed in chronological order, from the oldest to the youngest occurrence in each region starting from North and Central America, South America, Eastern Asia (i.e., Central Asia, China, Mongolia and Russia), Indian Subcontinent, Europe and Africa, with their temporal ranges, geographic distributions, time of origins and extinctions. Part of this information is also reported in Appendix A;(2)Integrate the distribution of the fossil record with paleoclimate and paleoecological data in order to provide new insights into the evolution of Equinae and their associated paleoenvironments;(3)Compare and discuss the latest morphological and genetic-based phylogenetic hypotheses on the emergence of the genus *Equus*. Recently, Barrón–Ortiz et al. [9] and Cirilli et al. [10] provided phylogenies of *Equus*, including fossil and extant species, with different resulting hypotheses. We compare and discuss the results of these two competing hypotheses here;(4)Provide a summary synthesis of the major patterns in the evolution, adaptation and extinction of Equidae 5.3 Ma–10 ka.

## 2. Materials and Methods

We provide a revised taxonomy of all Plio-Pleistocene *Equus* across the Americas, Eurasia and Africa summarizing previously published research, as a group of 19 researchers from Europe and North and South America. We provide an updated chronology and geographic distribution for these species. We followed the international guidelines for fossil and extant horse measurements published by Eisenmann et al. [11] and Bernor et al. [12]. Over the last 30 years, these methods have been applied to several case studies in Equinae samples from North and South America, Eurasia and Africa, which led to the identification of new species as well as to the clarification of the taxonomic fossil record. More recently, Cirilli et al. [13] and Bernor et al. [14] provided a combination of analyses to analyze fossil and extant samples including univariate, bivariate and multivariate analyses on cranial and postcranial elements using boxplots, bivariate plots, Log10 ratio diagrams and PCA. We found that robust, overlapping statistical and analytical methods led to a finer resolution of the taxonomy and, ultimately, biogeographical and paleoecological studies. We provide essential information on those species we recognized in the record under consideration. 

The taxonomic revision of the 5.3 Ma to 10 ka equid genera and species is given below and includes the authorship, chronological and paleobiogeographic ranges and some historical and evolutionary considerations on the taxon. A reduced emended diagnosis of the species is reported in the Appendix A in order to offer the most relevant anatomical features to identify the taxon. Additional information is reported in Appendix A. 

We compiled a global Neogene dataset of Equinae body mass estimates and paleodietary information for extensive palaeoecological analyses. These data were collected during several museum visits and complemented with data from publications and databases. Paleodietary information included results from traditional mesowear [15], low-magnification microwear [16] and isotopic analyses of equids from American, Eurasian and African localities (Appendix A). Net primary productivity (NPP) estimates were calculated for equine localities from the mean hypsodonty and mean longitudinal loph counts of large herbivorous mammal communities using the equation of Oksanen et al. [17]. The equation was as follows: NPP = 2601 − 144HYP − 935 LOP, where HYP is the mean ordinated hypsodonty, and LOP is the mean longitudinal loph count.

For the selected localities, body mass estimates based on metapodial measurements of equine paleopopulations [18,19], univariate mesowear scores calculated using the method of Saarinen et al. [20], and NPP estimates [17] were included to test whether the diet and productivity of the paleoenvironments were connected with body size patterns in equine horses. For this purpose, we used ordinary least squares linear models with body mass estimates as the dependent variable and the mean mesowear scores and NPP estimates as the explaining variables. These analyses were based on Eurasian and African *Equus* because of the large amount of data available and high variation in the ecology and environments of that genus during the Pleistocene, particularly in Europe but also, to some extent, in Asia and Africa. Because of slight methodological differences concerning the North American equine data [21] (Appendix A), we discuss them separately from Eurasian and African *Equus* in the context of the patterns revealed by the Eurasian and African *Equus* models. We also discuss the paleoenvironmental and habitat properties of key equine species based on what is known regarding the vegetation type and climate in their environments/paleoenvironments. Furthermore, we compared the patterns in the equine body size evolution, dietary ecologies and paleoenvironments between the continents and discuss how differential changes in biome distribution on the different continents could explain the observed differences in the body size patterns between continents.

We assembled data on the large herbivorous mammals (i.e., Artiodactyla, Perissodactyla, Proboscidea and Primates) from the NOW database [22] and calculated the mean ordinated crown height for each locality (Appendix A) following Fortelius et al. [23] for lists with at least two species with a hypsodonty value. All NOW localities between 7 Ma to recent from North and South America, Eurasia and Africa were included in the study and divided into four different age groups: 7–4 Ma; 4–2.5 Ma; 2.5–1.5 Ma; 1.5 to the recent. Mean ordinated crown height is a robust proxy for humidity and productivity at the regional scale [23,24,25,26]. We plotted the results onto present-day maps and interpolated between the localities using Quantum GIS 3.14.16 Pi. For the interpolations, thematic mapping and grid interpolation were used with the following settings: 20 km grid size; 800 km search radius; 800 grid borders. The interpolation method employed an inverse distance-weighted algorithm (IDW). 

Finally, we discuss the most recent phylogenetic outcomes on the origin of the genus *Equus*. We compared the morphological-based cladistic results of Barrón-Ortiz et al. [9] and Cirilli et al. [10] with the genomic-based analyses of Orlando et al. [27], Jónsson et al. [28] and Heintzman et al. [29] in order to identify similarities between the two different cladistic approaches.

## 3. Systematics of the Equinae since 5.3 Ma in North, Central and South America

### 3.1. North and Central America

Horses have been commonly found in numerous terrestrial North and Central American vertebrate faunas. From the Middle Miocene through the Early Pleistocene, the diversity of horses encompassed the genera *Astrohippus*, *Boreohippidion*, *Calippus*, *Cormohipparion*, *Dinohippus*, *Nannipus*, *Neohipparion*, *Pliohippus*, *Protohippus*, *Pseudohipparion* and an Hipparionini of indeterminate genus or species. The genus *Equus* is commonly interpreted to have first appeared during the Blancan North American Land Mammal Age (NALMA), though recent analyses propose an earlier origin of crown-group *Equus* that extends into the Middle to Late Pliocene [29]. Nevertheless, the genus peaked in diversity and widespread geographic distribution during the Pleistocene. In the particular case of North and Central America, we use *Equus* in the broad sense (i.e., sensu lato), as the generic taxonomy of this group of equids has not been resolved and is an area of ongoing study (e.g., [9,10,29]).

What follows is a summary of the species of equids present in North and Central America from the Late Miocene to the Late Pleistocene (Hemphillian to Rancholabrean NALMAs) based on a review of the literature. In the particular case of *Equus* sensu lato, we recognized potentially valid species based on a meta-analysis of relevant studies (published between 1901 and 2021; n = 68) discussing fossil specimens of this group of equids; details of this analysis are provided in the Appendix A.

1. *Calippus elachistus*, Hulbert, 1988 [30] (Right Mandibular Fragment with m2–m3, UF342139). This species seems to be restricted to Florida (USA) from the Late Miocene to the Late Hemphillian. The occlusal dimensions of its cheeck teeth are much smaller than any other species of *Calippus*, except *Ca. regulus*, with slightly smaller occlusal dimensions in the early to middle wear stages and significantly smaller basal crown lengths than *Ca. regulus*.2. *Calippus hondurensis*, Olson and McGrew, 1941 [31] (Partial Skull Containing Left P2–M3 and Right P2–4, WM 1769). This species has been reported in Puntarenas (Costa Rica), Gracias (Honduras), Mexico (Guanajuato, Hidalgo, Jalisco and Zacatecas) and in the USA (Florida). It may be distinguished by its small size and relatively small protocone.3. *Dinohippus leardi*, Drescher, 1941 [32] (M1, CIT 2645). This species has been recorded from the Late Miocene in California (USA). The size of the molars is similar to that of *Pliohippus nobilis* or larger in unworn teeth.4. *Dinohippus spectans*, Cope, 1880 [33] (Left M2 with Associated or Referred P2, AMNH 8183). This species has been recorded in Oregon, Nevada, California, Texas and Idaho (USA) dating to the Late Miocene, with molar teeth of larger size than those of any of the extinct American horses, except *Equus excelsus*, approximately equal to those of *Hippidion principale*.5. *Astrohippus ansae*, Matthew and Stirton, 1930 [34] (Partial Left Maxilla with P2-M3, UC30225). This species is a Hemphilian–Blancan species recorded in Zacatecas (Mexico), New Mexico, Oklahoma and Texas (USA).6. *Astrohippus stockii*, Lance, 1950 [35] (Palate with P2-M2, Front Portion of M3 on the Right Maxilla and P2-M2 on the Left One, CIT3576). This species has been recorded in Chihuahua, Guanajuato and Jalisco (Mexico) and Florida, New Mexico and Texas (USA), from the latest Hemphilian to Blancan NALMAs. *Astrohippus stockii* is smaller than *A. ansae*, but it possesses higher-crowned cheek teeth [35]. In recent phylogenetic analyses, *A. stockii* was recovered as the sister group to the clade composed of “*Dinohippus*” *mexicanus* plus *Equus* sensu lato [8,36] or the sister group to the clade composed of successive species of “*Dinohippus*” (i.e., “*Dinohippus*” *leardi*, “*Dinohippus*” *interpolatus*, *Dinohippus leidyanus* and “*Dinohippus*” *mexicanus*) plus *Equus* sensu lato [37].7. *Dinohippus interpolatus*, Cope, 1893 [38] (First and Second Upper Molars, Plate XII, Figures 3 and 4). This is a late Hemphillian-Blancan species that has been recorded in Hidalgo and Zacatecas (Mexico) and in California, Kansas, New Mexico and Texas (USA).8. *Dinohippus leiydianus*, Osborn, 1918 [3] (Skull, Jaws, Vertebrae, Fore and Hind Limbs, Considerable Portions of the Ribs and Other Parts of the Skeleton of One Individual, AMNH 17224). This species comes from the late Hemphillian–Blancan with records in Alberta (Canada) and in Arizona, California, Kansas, Nebraska and Oklahoma (USA).9. *Dinohippus mexicanus*, Lance, 1950 [35] (Partial Left Maxilla with P2-M3 and Part of the Zygomatic Arch, LACM-CIT 3697). This species has been found in Chihuahua, Guanajuato, Jalisco, Hidalgo, Nayarit and Zacatecas (Mexico) and in California, Florida, New Mexico and Texas (USA) from Hemphillian to Blancan NALMAs. It is a medium-sized monodactyl equine horse.10. *Cormohipparion occidentale*, Leidy, 1856 [39] (Four Left and One Right Upper Cheek Teeth, ANSP 11287). This species is a Hemphillian–Blancan species recorded in California, Florida, Nebraska, New Mexico and Oklahoma (USA). It is a large and hypsodont North American hipparion.11. *Nannippus aztecus*, Mooser, 1968 [40] (Fragmented Right Maxillary with P3–M3, FO 873). This species has been recorded in Mexico (i.e., Chihuahua, Guanajuato and Jalisco) and in the USA (i.e., Alabama, Florida, Louisiana, Mississippi, Oklahoma and Texas) from the latest Hemphillian to Blancan NALMAs. It is a small-sized horse.12. *Nannippus lenticularis*, Cope, 1893 [3] (Two Upper Cheek Teeth). This species has been recorded from Hemphillian to Blancan NALMAs in Alberta (Canada) and in Alabama, Kansas, Nebraska, Oklahoma and Texas (USA).13. *Nannippus peninsulatus*, Cope, 1885 [41] (M2, AMNH8345). This is a Hemphillian–Blancan species with records in Guanajuato, Hidalgo, Jalisco and Michoacan (Mexico) and in Arizona, Florida, Kansas, Nebraska, New Mexico and Texas (USA). *Nannippus peninsulatus* was a highly cursorial equid that appears to have been functionally monodactyl [42,43]. It had an estimated body mass of 59.6 kg [44].14. *Neohipparion eurystyle*, Cope, 1893 [38] (TMM 40289-1). This species has been recorded in Alberta (Canada); Guanajuato, Hidalgo, Jalisco and Zacatecas (Mexico); Alabama, California, Florida, Kansas, Nebraska, Oklahoma and Texas (USA) from the Hemphillian to Blancan NALMAs. It is a very hypsodont medium-sized hipparion.15. *Neohipparion leptode*, Merriam, 1915 [45] (Lower Molar, UCMP 19414). This species is a Hemphillian–Blancan species recorded in California, Kansas, Nebraska, Nevada, Oklahoma and Oregon (USA). It is a large hipparion.16. *Hipparionini genus* and Species Indeterminate. Hulbert and Harington [46] reported a remarkable specimen of a Hipparionini equid from the Canadian Arctic, which represents the northernmost fossil record of an equid reported to date. It was found in an Early Pliocene deposit (~3.5–4 Ma) from the Strathcona Fiord, Beaver Pond locality, Ellesmere Island, Canada [46]. The specimen consists of associated maxillae and premaxillae with the right dI1 and dP2–dP4 and the left dP1–dP4 of a young foal (approximately 6–10 months of age) [46]. It is a relatively large hipparionine equid (estimated adult tooth row length of 150 mm), with deciduous premolars that have low crowns; complex enamel plications; oval, isolated protocones; a facial region that shows a reduced preorbital fossa located posterior to the infraorbital foramen [46]. This combination of traits is not known in any contemporaneous North American hipparionines, but it is found in some Asiatic hipparionines, particularly *Plesiohipparion*, indicating possible affinities with this group and suggesting a previously unrecognized dispersal event from Asia into North America [8,46]. Alternatively, the Ellesmere Island hipparionine could represent a previously unknown autochthonous lineage of high-latitude North American hipparionines that potentially evolved from the mid-Miocene North American *Cormohipparion* [46].17. *Neohipparion gidley*, Merriam, 1915 [45] (Left M3, UCMP 21382). This species is a Hemphillian species recorded in California and Oklahoma (USA). It is the largest of the North American hipparions.18. *Boreohippidion galushai*, MacFadden and Skinner, 1979 [47] (Partial Skull with Well-Preserved Dentition, AMNH 100077). This is a late Hemphillian horse from Arizona (USA).19. *Cormohipparion emsliei*, Hulbert, 1987 [48] (partial skull with most of the right maxilla including dP1, P2-M3; right and left premaxillae with I1–I3; edentulous fragment of the left maxilla with alveoli for dP1 and P2). UF 94700. All elements possibly belong to the same individual as they present similar stages of tooth wear and preservation. It is a species recorded in the latest Hemphillian to Blancan NALMAs in Alabama, Florida and Louisiana (USA). It is a medium-sized species of *Cormohipparion*. 20. *Pseudohipparion simpsoni*, Webb and Hulbert, 1986 [49] (Associated P3-M1, UF 12943). This is a latest Hemphillian species recorded in Florida, Kansas, Oklahoma and Texas (USA).21. *Pliohippus coalingensis*, Merriam, 1914 [50] (UCMVP 21341). This species is a Pliocene horse from California (USA).22. *Nannippus beckensis*, Dalquest and Donovan, 1973 [51] (Partial Skull with Right and Left P2–M3, TMM41452-1). This is a Blancan species from Texas (USA). This species is a medium-sized and moderately hypsodont hipparion.23. *Equus simplicidens*, Cope, 1892 [52] (Left M1, TMM 40282-6). This species is interpreted to have been a medium- to large-sized equid with primitive dentition [53], recorded in Baja California (Mexico) and Arizona, California, Idaho, Kansas, Nebraska and Texas (USA) from Blancan to Irvingtonian. The species was initially based upon fragmented molars, with sizes comparable to *E. occidentalis* and *E. caballus* [52]. According to Skinner [54], *E. simplicidens* shows great similarities in the skull and dentition with the modern *Equus grevyi*, and the differences in the skull are small and expected in temporal and geographic separation. Gidley [55] described the Hagerman horses based upon characters common to all zebrine horses, with taxonomically significant differences from non-zebrines, but the characters used to distinguish it from other zebrines are of doubtful validity [56]. Comparing *E*. *simplicidens* with the East African Grevy’s zebra, *E. grevyi*, Skinner [54] included both in the subgenus *Dolichohippus* [57]. This proposal was questioned by Forsten and Eisenmann [57], as the cranial similarities found by Skinner [54] might be allometrically related to the large skull size [57]. Furthermore, Skinner [54] did not compare the basicranium, missing the comparison of Franck’s Index (i.e., the distance from the staphylioin to the hormion and from the hormion to the basion). The index was considered phylogenetically important, as the lengthening of the hormion to the basion distance seems to have led to a decrease in the index during *Equus* evolution, with a high index being related to a more primitive character than a lower derived index [57]. In Forsten and Eisenmann’s [57] analysis, both *E. simplicidens* and *Pliohippus* (*Dinohippus*), considered the generic ancestor of *Equus*, presented a high index, while *E. grevyi* and the other extant species presented lower indices. Following Matthew [4], Forsten and Eisenmann [57] also suggested *E. simplicidens* should be included in the subgenus *Plesippus* [4,58]. *Equus simplicidens* has long been considered the earliest common ancestor of *Equus* [57], but a recent analysis of the genus *Equus* suggested that *Plesippus* and *Allohippus* should be elevated to a generic rank, indicating *Allohippus stenonis* as the sister taxon to *Equus* and *Plesippus simplicidens* and *P. idahoensis* as the sister taxa to the *Allohippus* plus *Equus* clades [9]. On the other hand, recent cladistic analyses combined with morphological and morphometrical comparisons of skulls suggest *E. simplicidens* as the ancestor of *Equus*, not endorsing *Plesippus* and *Allohippus* at the genus or subgenus level [10].24. *Equus idahoensis*, Merriam, 1918 [59] (Upper Left Premolar, UCMP 22348). This is a Blancan and early Irvingtonian species recorded in Arizona, California, Idaho and Nevada (USA). The type locality is Locality 3036C, in the beds of the Idaho locality, near Froman Ferry on the Snake River, 8 mi SW Caldwell, Idaho. According to Winans [56], none of the traits from the original description of *E. idahoensis* are unique to this species. However, large samples of specimens (e.g., Grandview, Idaho; 111 Ranch, Arizona) have been referred to as *E. idahoensis*, which have distinctive morphological features [9,60] that indicate that this is a potentially valid species. The cheek teeth are large and heavily cemented.25. *Equus enormis*, Downs and Miller, 1994 [61] (The holotype, IVCM 32, is a partial skull and right and left mandibles, with the right distal humerus, right radius-ulna, MCIII, unciform, magnum, trapezoid and MCIII, phalanges 1, 2, and 3 of the manus; partial pelvis, right femur, MTIII with MTII and MTIV and phalanx 3 of the pes from Vallecito Creek, Anza-Borrego Desert State Park, San Diego County, California, USA). This species is known primarily from the late Blancan – Irvingtonian of California (USA). *Equus enormis* is a large-sized monodactyl horse with an estimated height at the withers of 1.5 m.26. *Equus cumminsii*, Cope, 1893 [38] (Fragmentary Upper Molar, TMM 40287-14). This small species has been recorded in Kansas and Texas (USA) from Blancan to early Irvingtonian (NALMAs). Although it is poorly represented by fossils and the type of specimen is a single damaged tooth, some authors consider this species as an early ass based on dental morphology [60,62,63].27. *Equus calobatus*, Troxell, 1915 [64] (Left MTIII, YPM 13470). This species is a large stilt-legged horse reported from the late Blancan to early Rancholabrean NALMAs, with records in Alberta (Canada); Aguascalientes (Mexico); Colorado, Kansas, Nebraska, New Mexico, Oklahoma and Texas (USA). The original discovery consisted of “unusually long and slender” limb bones [64] from Rock Creek, Texas, but no single holotype specimen was designated. Hibbard [65], therefore, selected YPM 13460 as the lectotype. Because the lectotype and cotypes are limb bones with no distinctive characters other than the large size and relative slenderness of the metapodials, there are few morphological criteria available for evaluating this species. Multiple studies [29,56] have synonymized *E. calobatus* with *Equus* (or *Haringtonhippus*) *francisci*, but other studies consider it a valid species [66,67].28. *Equus scotti*, Gidley, 1900 [68] (Associated Skeleton with Skull, Mandible, Complete Feet and Forelimb Bones, One Complete Foot and Hindlimb and All the Cervical, Several Dorsal and Lumbar Vertebrae, AMNH 10606). This species is recorded from the late Blancan to Rancholabrean NALMAs in Alberta, Ontario, Saskatchewan and Yukon (Canada) and in California, Florida, Idaho, Kansas, Nebraska, New Mexico, Oklahoma and Texas (USA). Winans [54] also interpreted *E. scotti* to be on average slightly smaller than *E. simplicidens*, but the measurements provided in that study actually indicate the opposite, and subsequent review confirms that *E. scotti* was of a larger form than *E. simplicidens*.29. *Equus stenonis anguinus*, Azzaroli and Voorhies, 1993 [69] (Complete Skull and Jaw, USNM 23903). This is a late Blancan species recorded in Arizona and Idaho (USA). It is described as similar to *E. stenonis* from the Early Pleistocene of Italy, with skull dimensions falling within the size range of this latter species [69]; however, the limb bones are, on average, more elongated. *Equus stenonis anguinus* possesses a preorbital pit and a deep narial notch as do the European *E. stenonis* samples.30. *Equus conversidens*, Owen, 1869 [70] (Fragmentary Right and Left Maxilla with All Cheek Teeth, IGM4008, Old Catalog Number MNM-403). This is a widespread species reported to have ranged from the Irvingtonian to Rancholabrean with a geographic distribution encompassing North and Central America: Alberta (Canada); Aguacaliente de Cartago (Costa Rica); Apopa Municipality (El Salvador); Aguascalientes, Chiapas, Hidalgo, Jalisco, Michoacán, Nuevo Leon, Puebla, Oaxaca, San Luis Potosi, Sonora, Estado de Mexico, Tlaxcala, Yucatán and Zacatecas (Mexico); Azuero Peninsula and El Hatillo (Panama); Arizona, California, Florida, Kansas, Nebraska, New Mexico, Oklahoma, Texas and Wyoming (USA). The holotype specimen from the Tepeyac Mountain was described by Owen [70] based upon photos [71]. Owen considered the species to be almost identical to *E. curvidens* (South American *E. neogeus*) but with cheek tooth rows converging towards their anterior ends. Cope [72] interpreted the anterior convergence of the cheek tooth rows to be an artifact of the restoration compounded by the photography and assigned the specimen to *E. tau*, albeit without any stated justification. Gidley [69] interpreted the two sides of the maxilla to be from different individuals, since they were found separately and with missing broken edges. Hibbard [65] provided a reconsideration of the specimen and confirmed that Cope [68] was correct regarding the distortion of the palate and that Gidley [73] was incorrect regarding the two sides deriving from different individuals. Azzaroli [74] described a fragmentary skull (LACM 308/123900) from Barranca del Muerto near Tequixquiac, Mexico, in which the “two tooth rows converge rostrally, giving evidence that the palate of the holotype (of *E. conversidens*) was correctly mounted and that Owen’s name is after all appropriate”.31. *Equus lambei*, Hay 1915 [75] (~200 ka–~10 ka) (nearly complete skull from a female, USNM8426, collected from Gold Run Creek in the Klondike Region, Yukon Territory, Canada). This species inhabited the steppe–tundra grasslands of Beringia, with remains having been recovered from Siberia, Alaska, and the Yukon (extending slightly into the adjacent Northwest Territories). Recent genomic evidence suggests that *E. lambei* and *E. ferus* may represent a single species [29,76,77,78], although further research is required before this phylogeny can be resolved (see Appendix A supplementary text for a discussion on *E. ferus* in North America).32. *Equus* (or *Haringtonhippus*) *francisci*, Hay, 1915 [76] (Complete Cranium, Mandible and MTIII, TMM 34–2518). This species has been recorded from Irvingtonian to Rancholabrean localities in the Yukon (Canada); Aguascalientes, Estado de Mexico, Jalisco, Puebla, Sonora, and Zacatecas (Mexico); Alaska, Arizona, Florida, Kansas, Nebraska, New Mexico, Texas and Wyoming (USA). It is the oldest name assigned to the stilt-legged group and was first described as being similar to *E. tau* but with different P3-M1 proportions, which is expected in teeth at different stages of wear and, therefore, probably not a significant taxonomical difference (56). Eisenmann et al. [67] reassigned *E. francisci* to the genus *Amerhippus*, as *Amerhippus francisci*. Heintzman et al. [29] assigned stilt-legged, non-caballine specimens from Gypsum Cave, Natural Trap Cave, the Yukon and elsewhere to their new genus *Haringtonhippus* under the species *Ha. francisci*, based upon complete mitochondrial and partial nuclear genomes as well as morphological data and a crown group definition of the genus *Equus.* Barrón-Ortiz et al. [9] considered *Haringtonhippus* to be a synonym of *Equus* and regarded both *E. francisci* and *E. conversidens* as distinct taxa based upon morphological criteria.33. *Equus fraternus*, Leidy, 1860 [79] (Upper Left P2, AMNH 9200). This species has been recorded in Alberta (Canada) and in Florida, Illinois, Mississippi, Nebraska, Pennsylvania, South Carolina and Texas (USA) from Irvintonian to Rancholabrean. Winans [56] considered the dental characteristics used to identify the species to vary with wear and that the specimens used in its diagnosis represented more than one individual, making it uncertain whether to attribute it to one species. Azzaroli [74,80] referred several complete skulls, mandibles and other bones from the southeastern USA to this species.34. *Equus pseudaltidens*, Hulbert, 1995 [67] (right maxillary with worn DP2, DP3, DP4, M1 and M2 and unerupted P2, P3 and P4 (BEG 31186-35); right and left mandibles with worn i1, di2, dp2, dp3, dp4, m1 and m2 and unerupted p2, p3 and p4 (BEG 31186-36); cranium lacking occiput (BEG 31186-37); right third metacarpal (BEG 31186-3); right and left femora (BEG 31186-2, 34); right and left tibiae (BEG 31186-1, 10); right and left third metatarsals (BEG 31186-4, 7); first phalanx (BEG 31186-24), all thought to belong to the same individual and estimated to have been approximately 3 years old [81]). This species was originally described as *Onager altidens* by Quinn [81]. The use of *Onager* instead of *Hemionus* by Quinn [81] was invalid [67]. Referral of this species to the genus *Equus* makes it a homonym of *Equus altidens*, von Reichenau, 1915 [53,67,82]. Therefore, Hulbert [67] proposed the replacement name *E. pseudaltidens* for *E. altidens* (Quinn). Also referred to this species are a pair of maxillae (BEG 31186-23) and a right mandible (BEG 31186-22) of an animal approximately 1 year old, and 24 deciduous and permanent upper and lower teeth recovered from the type locality [81]. *Equus pseudaltidens* is known from the Irvingtonian–Rancholabrean and it has been reported from the Gulf Coastal Plain of Texas [67,81] and possibly from Coleman, Florida [67]. It is a stilt-legged equid with metapodial dimensions that are similar to extant hemionines [67,76]. Compared to other stilt-legged equids discussed here, *Equus pseudaltidens* is smaller than *E. calobatus* but larger than both *E. francisci* and *E. cedralensis* [67,76,83]. Kurtén and Anderson [84] synonymized *E.* (*Hemionus*) *pseudaltidens* with *Equus* (*Hemionus*) *hemionus*. Winans [85] assigned it to her *E. francisci* species group. Hulbert [67] considered that *E. pseudaltidens* and *E. francisci* were distinct species and hypothesized that they are sister taxa. Azzaroli [74] synonymized *E. pseudaltidens* (as *Onager altidens*) with *E. semiplicatus*. Eisenmann et al. [67] considered *E. pseudaltidens* distinct from *E. semiplicatus* and *E. francisci* and assigned it along with the latter species to *Amerhippus*. Heintzman et al. [29] considered *E. altidens* (=*E. pseudaltidens*) a junior synonym of *Haringtonhippus francisci*.35. *Equus verae*, Sher, 1971 [86] (Holotype Mandible with a Full Row of Teeth, GIN 835-123/21, River Bolshaja Chukochya Exp. 21, Kolyma Lowland, Northeast Yakutia). This species is a large-bodied, stout-legged Early Pleistocene (Olyorian)–Rancholabrean species recorded in Northeastern Siberia (Russia) and the Yukon (Canada). *E. verae* is much larger than the *E. stenonis* species and similarities in the teeth and the size of limb bones with *E. suessenbornensis*, suggesting a subspecies position for *E. verae* as well as for *E. coliemensis* (see below). Eisenmann [87,88] suggested that *E. verae* may belong to the subgenus *Sussemionus*, but this has not been substantiated by other authors.36. *Equus occidentalis*, Leidy, 1865 [89] (Lectotype Left P3, VPM 9129). It is a Rancholabrean NALMA horse with records in Mexico (i.e., Baja California and Sonora) and the USA (i.e., Arizona, California, Nevada, New Mexico and Oregon). Leidy [89] named *E. occidentalis* from two upper premolars and one lower molar from two widely separated geographic localities but did not designate a holotype. Gidley [73] selected a left P3 from Tuolumne County, California, as the lectotype. Merriam [90] referred to *E. occidentalis* thousands of bones of a large and stout-limbed equid recovered from Rancho La Brea. Savage [91] and Miller [92] believed that the equid from Rancho La Brea, identified as *E. occidentalis*, did not conform to the lectotype designated by Gidley [73], but neither of these authors proposed a new name. Azzaroli [74] decided to retain the name *E. occidentalis sensu* Merriam [90] and selected the skull figured by Merriam [90] as his lectotype. However, since Gidley [73] had already designated a lectotype for the species, according to ICZN Article 74, no subsequent lectotype designations can be made. Brown et al. [93] concluded that some of Leidy’s original fossils of *E. occidentalis* (exclusive of the Tuolumne County tooth) most likely come from the McKittrick asphalt deposits; this locality was not named by Leidy [89], because the town of McKittrick, California, was not named until 1900. These authors also confirmed that many specimens from the original type series of *E. occidentalis* closely resemble the large Pleistocene horses from McKittrick and Rancho La Brea [93]. Barrón-Ortiz et al. [94] recognized the presence of this species outside of the North American Western Interior during the Late Pleistocene. Barrón-Ortiz et al. [9] recognized it as a valid species closely related to *E. neogeus*.37. *Equus cedralensis*, Alberdi et al., 2014 [83] (fragment of a mandibular ramus formed by two specimens: one p2-m3 right row (DP-2675 I-2 15) and a second fragment of the symphysis with the anterior dentition (DP-2674 I-2 8), articulated together from Rancho La Amapola, Cedral, San Luis Potosí, Mexico, and stored at the Paleontological Collection (DP-INAH) of the Laboratorio de Arqueozoología “M. en C. Ticul Álvarez Solórzano” Subdirección de Laboratorios y Apoyo Académico, INAH in Mexico City). This species is primarily known from the Rancholabrean of Mexico (i.e., Aguascalientes, Chihuahua, Estado de Mexico, Michoacán, Puebla and San Luis Potosí). *Equus cedralensis* is an equid with a small body mass (estimated mean mass of 138 kg) [83,95]. *Equus cedralensis* was diagnosed as stout-legged, but a recent analysis placed it within stilt-legged horses and with its dental morphology being similar to *Ha. francisci*. Jimenez-Hidalgo and Diaz-Sibaja [96] considered it a junior synonym of *Ha*. *francisci*. *Equus cedralensis* differs from the holotype of *Ha*. *francisci*, as it is smaller in size and the lower first and second incisors possess enamel cups.38. *Equus mexicanus*, Hibbard, 1955 [65] (cranium lacking the LM3 (No. 48 (HV-3)) from Tajo de Tequixquiac, Estado de México, Mexico, and stored at the Museo Nacional de Historia Natural; the specimen is cataloged as IGM4009). This species is known from the Rancholabrean of Mexico (i.e., Aguascalientes, Chiapas, Estado de Mexico, Jalisco, Michoacán, Oaxaca, Puebla, San Luis Potosi and Zacatecas) and the USA (i.e., California, Oregon and Texas). *Equus mexicanus* is a large body sized species (estimated mean mass of 458 kg) [83,95]. Winans [85] placed *E*. *mexicanus* in her *Equus laurentius* species group, but as with other species groups in this study, this was not a strict synonymy. Azzaroli [74] recognized *E*. *mexicanus* as a valid taxon, noting that previous investigations had proposed synonymy with and tentatively identified as *E*. *pacificus* but rejecting this, since the latter species was initially based upon a single tooth. Barrón-Ortiz [97] assigned specimens identified as *E*. *mexicanus* to *E*. *ferus scotti*. Barrón-Ortiz et al. [94] assigned specimens identified as *E*. *mexicanus* to *E*. *ferus*. Barrón-Ortiz et al. [9] recognized *E*. *mexicanus* as a valid taxon distinct from *E*. *ferus*.

### 3.2. South America

Two genera, *Equus* and *Hippidion*, inhabited the South American continent with records from the Late Pliocene to the Late Pleistocene [98,99,100]. *Hippidion* is an endemic genus of South American horses characterized mostly by the retraction of the nasal notch, a particular tooth morphology (considered more primitive than Equus and comparable with Pliohippus) and the robustness and shortness of its limb bones [98]. The genus is, at present, represented by three species: *Hippidion saldiasi*, *Hippidion devillei* and *Hippidion principale* [98]. On the other hand, the South American *Equus* is represented by a single species, *E. neogeus*, with caballine affinities and metapodial variation corresponding to an intraspecific characteristic representing a smooth cline [9,99,100].

1. *Equus neogeus*, Lund, 1840 [101] (MTIII, 866 Zoologisk Museum). This is a Middle–Late Pleistocene species (Ensenandan and Lujanian SALMA) and the only representative of the genus in the South American continent [9,98,99,100]. Most records are from the Late Pleistocene, but its earliest appearance is recorded in the Middle Pleistocene in Tarija, Bolivia, dated at approximately 1.0–0.8 Ma [102,103]. It has a wide geographic range distribution, encompassing all of South America, except for the Amazon basin and latitudes below 40° [100]. The species probably became extinct sometime during the Late Pleistocene–Holocene transition as suggested by the youngest direct radiocarbon date of 11,700 BP (Río Quequén Salado, Argentina [104]).2. *Hippidion saldiasi*, Roth, 1899 [105] (p2, Museo Nacional de La Plata). This is a Late Pleistocene species, dated between 12,000 and 10,000 years BP, mostly known from Argentinian and Chilean Patagonia, with records in Central Chile and the Atacama Desert [98,106]. The last records for the species were radiocarbon dated between 12,110 and 9870 BP in Southern Patagonia (Cerro Bombero, Argentina [107]) and Cueva Lago Sofía, Chile [108].3. *Hippidion principale*, Lund, 1846 [109] (M2, Peter W. Lund Collection, ZMK). This is a Late Pleistocene species (Lujanian SALMA), with records in Argentina, Bolivia, Brazil and Uruguay [98]. This species represents the largest *Hippidion*. There are few radiocarbon-dated records for this species, with the youngest situated at approximately 16,130 BP (Arroyo La Carolina, Argentina) [104]; however, remains found in archaeological contexts dating close to 13,200 BP in Tagua, Central Chile [110], suggest a later presence for this taxon.4. *Hippidion devillei*, Gervais, 1855 [111] (P2–P3 Row and Fragmented Astragalus, IPMNHN). This species has been reported in Uquia (Argentina, Late Pliocene–Early Pleistocene), Tarija (Bolivia) and Buenos Aires (Argentina) from the Middle Pleistocene (Ensenadan SALMA) to the Late Pleistocene (Lujanian SALMA) and in Brazil [98]. This taxon has been directly radiocarbon dated only from cave contexts in the high Andes of Peru, with the youngest record of 12,860 BP [112,113].

## 4. Systematics of the Equinae since 5.3 Ma in Eurasia and Africa

### 4.1. Eastern and Central Asia (China, Mongolia, Russia, Uzbekistan, Kazakhstan and Tajikistan)

The fossil record of the three-toed horses from Eastern Asia includes four different genera (i.e., *Plesiohipparion*, *Cremohipparion*, *Proboscidipparion* and *Baryhipparion*) with seven identified species. As for the Indian Subcontinent, Europe and Africa (see below), the Miocene–Pliocene boundary marks the extinction of the genera *Hippotherium*, *Hipparion* s.s., *Sivalhippus* and *Shanxihippus* [8]. On the other hand, Sun and Deng [114] argued that the *Equus* Datum in China is represented by the simultaneous appearance of five stenonine *Equus* species: *E. eisenmannae*, *E. sanmeniensis*, *E. huanghoensis*, *E. qyingingensis* and *E. yunnanensis*. Subsequently, Sun et al. [115] indicated the *E. qyingingensis* FAD at 2.1 Ma. 

1. *Plesiohipparion houfenense*, Teilhard de Chardin and Young, 1931 [116] (MN13–MN15; 6–3.55Ma). The lectotype RV 31031 includes the right p3–m3 from Jingle, Shanxi. The earliest *P. houfenense* first occurs in the Late Miocene Khunuk Formation, Kholobolchi Nor, Mongolia [8,117,118,119] and the Late Miocene/Early Pliocene Goazhuan Formation of the Yushe Basin (5.8–4.2 Ma) [8,120]. It also occurs into the Pliocene of China. 2. *Proboscidipparion pater*, Matsumoto, 1927 [121] (MN14–MN15; 5–3.5 Ma). The lectotype THP14321 is a skull with a mandible estimated to be 4–3.55 Ma [122]. This species is reported from the Yushe Basin (China), and it may be the original source for the evolution of the Pliocene European species *Proboscidipparion crassum* and *Proboscidipparion heintzi*. 3. *Plesiohipparion huangheense*, Qiu et al., 1987 [122] (MN15; 5.0–3.55). The lectotype THP 10097 is a lower jaw fragment, including the cheek teeth, from the Yushe Basin [122]. It is a Chinese species reported from Inner Mongolia at 3.9 Ma [7,119,122] and more broadly from the MN15 of China and India [8]. Ultimately, *Pl*. aff. *huangheense* has been reported from the Early Pleistocene in Gulyazi, Turkey [123]. 4. *Cremohipparion licenti*, Qiu et al., 1987 [122] (MN15, circa 4.0 Ma). The holotype is THP20764, an incomplete cranium from the Yushe Basin [122]. This distinctly Chinese species is the latest occurring member of the genus *Cremohipparion* and is reported from the Yushe Basin [8]. 5. *Baryhipparion insperatum*, Qiu et al., 1987 [122] (MN16–MNQ17; 3.55–1.8 Ma). The holotype is THP19009, an incomplete cranium with the mandible from the Yushe Basin [122]. It is reported that this species is from the Pliocene in China. 6. *Plesiohipparion shanxiense*, Bernor et al., 2015 [124] (MNQ17; 2.5–1.8 Ma). The holotype is F:AM111820, a complete skull with the mandible [124]. This species, previously recognized as *Plesiohipparion* cf. *P. houfenense* [117], is the largest and, at the same, the time youngest member of the genus in Eastern Eurasia. It is believed to be 2.0 Ma in age [124]. It may represent the last evolutionary stage of the genus *Plesiohipparion* in China. The absence of the POF suggest an evolutionary relationship with *Pl. houfenense.*7. *Proboscidipparion sinense*, Sefve, 1927 [125] (MN17-MQ1; 2.5–1.0 Ma). The holotype is PMU M3925, a complete cranium from Henan Province, China. *Proboscidipparion sinense* occurs later in the record and is approximately one-seventh larger than *P. pater. Proboscidipparion sinense* is the latest occurring hipparion in China extending its range up to 1.0 Ma [126]. 8. *Equus eisenmannae*, Qiu et al., 2004 [127] (2.55–1.86 Ma). The holotype is IVPP V13552, a complete cranium with and the mandible from Longdan [127]. It is a large-sized horse, mostly known from the Early Pleistocene locality of Longdan (China), similar in size to *E. livenzovensis* (see below), and the primitive features of the cranial morphology suggest a close evolutionary relationship with *E. simplicidens* [10,13]. At the present time, its evolutionary linkage with other Chinese *Equus* is not known. 9. *Equus sanmeniensis*, Teilhard de Chardin and Piveteau, 1930 [128] (2.5–0.8 Ma). The lectotype is NIH 002 (Paris), a complete cranium with the mandible from Nihewan, Hebei Province. It is a large-sized, Early Pleistocene species from North and northwest China, Siberia (Aldan River, Bajakal Lake area), Kazakhstan and Tajikistan [114,129]. Sun and Deng [114] suggested a morphological similarity between *E. sanmeniensis* and *E. simplicidens*, although diversified from *E. stenonis*. This evolutionary hypothesis has also been supported by Cirilli et al. [10,13] through morphometric studies on crania. *Equus sanmeniensis* has been reported from the Early to Middle Pleistocene [114,130]. 10. *Equus huanghoensis*, Chow and Liu, 1959 [131] (2.5–1.7 Ma). The holotype is IVPP V2385–2389, with three upper premolars and two molars from Huanghe, Shanxi (Sun and Deng, 2019). It is a large-sized, Early Pleistocene species from the localities of Nihewan (Hebei), Linyi (Shanxi), Sanmenxia Pinglu (Shanxi), Xunyi (Shaanxi) and Nanjing (Jiangsu). Sun and Deng [114] supported the hypothesis provided by Deng and Xue [130] that *E. huanghoensis* is a stenonid horse, considering the Nihewan sample as one of the *Equus* species with the largest palatal length with *E. eisenmannae*. The morphometric analyses of Cirilli et al. [10,13] show a primitive morphology of the cranium, similar to *E. simplicidens* and distinct from *E. stenonis*. Ao et al. [132] indicated the age of 1.7 Ma as the youngest record of this species in China. 11. *Equus yunnanensis*, Colbert, 1940 [133] (2.5–0.01 Ma). The lectotype is IVPP V 4250.1, an almost complete but deformed cranium. It is a medium-sized, Early Pleistocene species from the Chinese localities of Yuanmou (Yunnan), Liucheng (Guangxi), Jianshi and Enshi (Hubei), Hanzhong (Shaanxi) and Huili (Sichuan) and from Irrawaddy in Myanmar. The species was initially described by Colbert [133] on isolated cheek teeth, while better knowledge of this species came with the new discoveries from the Yuanmou locality. Deng and Xue [130] proposed a close evolutionary relationship with *E. wangi*, whereas Sun and Deng [114] suggested a close evolutionary relationship with *E. teilhardi*, suggesting that these species are distinct from all other Chinese stenonid horses [114]. 12. *Equus teilhardi*, Eisenmann, 1975 [134] (2.0–1.0 Ma). The holotype is NIH001, an incomplete mandible. It is a medium-sized, Early Pleistocene species from northwestern and North China. Sun et al. [135] proposed a close evolutionary relationship between *E. teilhardi* and *E. yunnanensis*, later supported by the cladistic analysis of Sun and Deng [114]. 13. *Equus qyingyangensis*, Deng and Xue, 1999 [130] (2.1–1.2 Ma). The holotype is NWUV 1128, an incomplete cranium. It is a medium-sized, Early Pleistocene species from northwestern and North China. Eisenmann and Deng [136] recognized some close anatomical features between *E. simplicidens* and *E. qyingyangensis*, suggesting a close evolutionary relationship between these two species. The latest phylogenetic results of Sun and Deng [114] and Cirilli et al. [10] support this last hypothesis. A new *E. qyingyangensis* sample has recently been described from Jinyuan Cave [115], with an FAD of 2.1 Ma. 14. *Equus wangi*, Deng and Xue, 1999 [130] (2.0–ca. 1.0 Ma). The holotype is NWUV 1170, a complete upper and lower cheek teeth rows from Gansu Province, Early Pleistocene [130]. Sun and Deng [114] reported a large size for *E. wangi*, similar to *E. eisenmanne*, *E. sanmeniensis* and *E. huanghoensis*. The phylogenetic position of *E. wangi* is not well defined, although Sun and Deng [114] highlighted a possible closer relationship with *E. eisenmannae* than any other stenonine *Equus*. 15. *Equus pamirensis*, Sharapov, 1986 [137] (Early Pleistocene). The holotype is IZIP 1-438 (Institute of Zoology and Parasitology, Uzbekistan), a complete upper tooth row from Kuruksai [137], approximately 2 Ma (MN17), possibly earlier. It is a large species of stenonine horse described from the Kuruksai, 18 km NE of Baldzuan, Tajikistan, in the Kuruksai River valley in the Afghan–Tajik Depression [138]. The site has been correlated to the middle Villafranchian. The taxonomy of this horse is contentious [138], with it being referred to variously as *Equus* (*Allohippus*) aff. *sivalensis* [137] and *E. stenonis bactrianus* [139]. We currently regard this as a distinct species.16. Central Asian Small *Equus* sp. from Kuruksai [138] (Early Pleistocene). This is a small species of horse that co-occurs with the larger *E. pamirensis* at Kuruksai. Metrically, the metapodials fall within the range of variation of *E. stehlini* [138]. Similar small horse remains have also been found in the Early–Middle Pleistocene Lakhuti 1 locality in the Afghan–Tajik Depression [138].17. *Equus* (*Hemionus*) *nalaikhaensis*, Kuznetsova and Zhegallo, 1996 [140]. This species was found in the late Early Pleistocene and early Middle Pleistocene (approximately 1 my, Jaramillo paleomagnetic episode) in Mongolia. The lectotype PIN 3747/500 is represented by the incomplete skull of an old male from Nalajkha [141]. 18. *Equus coliemensis*, Lazarev, 1980 [142]. The holotype is a skull with very worn teeth (col. IA 1741). The type locality is the river Bolshaja Chukochya, Kolyma lowland, northeast Yakutia, Siberia, Russia. It is reported from the late Early Pleistocene in northeastern Siberia (Russia). Recently, Eisenmann [87] included *E. coliemensis* in the subgenus *Sussemonius*. 19. *Equus lenensis*, Rusanov, 1968 [143]. The holotype skull comes from the Lena River delta (GIN Yakutia col. 33). The type locality is the river Bolshaja Chukochya, Kolyma lowland, northeast Yakutia, Siberia (Russia). It is reported from the Middle Pleistocene from northeastern Siberia (Russia). Lazarev [144] considered *E. lenensis* to be close to the North American *E. lambei*, even larger and more heavily built than the latter. It is also known from the Middle Pleistocene in Yakutia *Equus orientalis (Equus caballus/ferus orientalis)* and *Equus nordostensis (Equus ferus/caballus nordostensis)* [143,144,145]. *Equus nordostensis* is characterized by a large size based on the skull, with low plication of the “marks” of the upper teeth and a long protocone [144]. According to Kuzmina [129], it is junior synonym of *E. mosbachensis*. *Equus orientalis* has a large skull and a long snout with an elongated teeth row, flat protocone and rare plication of the upper teeth [144].20. *Equus beijingensis*, Liu, 1963 [146] (late Middle Pleistocene). The holotype is V2573-2574, a palate and jaw from Zhoukoudian, China [146,147]. It is a Chinese caballine horse recovered mostly form locality 21 of Zhoukoudian. Unfortunately, the species is not well represented, although Forsten [147] indicated a similar size with *E. sanmeniensis*. Liu [146] also indicated *E. sanmeniensis* as the possible ancestor for *E. beijingensis*; nevertheless, this hypothesis was discarded by Fortsen [147] and Deng and Xue [130], who identified *E. beijingensis* as a caballine horse, characterized by a U-shaped linguaflexid. Its evolutionary position is not well defined, although Forsten [147] and Deng and Xue [130] proposed that *E. beijingensis* is a relative of European *E. ferus* (their *E. mosbachensis*) or of a North Americam caballine horse [130]. 21. *Equus valeriani*, Gromova, 1946 [148] (Late Middle–Late Pleistocene). The hypodigm includes upper and lower cheek teeth, figured in Eisenmann et al. [149]. It is an enigmatic taxon, described by Gromova [148], from Samarkand, Uzbekistan. According to Gromova [148] and Eisenmann et al. [149], it shows a stenonine metaconid-metastylid in the lower cheek teeth but with a long protocone in the upper cheek teeth. Its possible occurrence has also been proposed in Syrie (Kéberien Géométrique d’Umm el Tlel), although this identification still remains uncertain [149].22. *Equus dalianensis*, Zhow et al., 1985 [150] (Late Pleistocene). The holotype is V821966, an incomplete mandible preserving two lower cheek teeth rows. It is a Chinese caballine horse described from Gulongshan Cave, Liaoning. Forsten [147] demonstrated close morphological and morphometrical similarities with *E. ferus gemanicus*, *E. ferus orientalis* and *E. ferus chosaricus*, suggesting that they all represent individual populations of a single widespread species, *E. ferus*. Deng and Xue [130] suggested a common origin for *E. dalianensis* and *E. przewalskii*, with no ancestor-descendant relationships between them. Nevertheless, a recent genomic analysis by Yuan et al. [151] revealed that *E. dalianensis* is a separate clade of caballine horses, distinct from *E. przewalskii.*23. *Equus ovodovi*, Eisenmann and Vasiliev, 2011 [152] (0.04–0.01 Ma). The holotype is IAES 21, a fragmentary palate from Proskuriakova Cave [152]. It was described in the Late Pleistocene site of Proskuriakova Cave (Khakassia, southwestern Siberia, Russia). It was first considered a species related to *E. hydruntinus* and modern hemiones, although the genomic analyses by Orlando et al. [153] suggest a relationship with wild asses, representing a new separated fossil clade with no extant relatives [150]. For this reason, Orlando et al. [153] and Eisenmann and Vasiliev [152] included this species in the subgenus *Sussemionus*. Molecular studies suggest that *E. ovodovi* is a sister to extant zebras and is nested within the clade that includes both extant zebra and asses [154]. *Equus ovodovi* has been recognized in the Late Pleistocene of southern and eastern Russia and more recently in China [152,154,155].24. *Equus hemionus*, Pallas, 1774 [156] (0.0 Ma). Pallas [156] did not refer to any holotype or lectotype but gave a detailed description of the anatomical features of the species, associated with an illustration of an animal located near Lake Torej-Nur, Transbaikal area [156] (V.19, pp. 394–417, Pl. VII). *Equus hemionus* is known as the Asiatic wild ass, distributed in China, India, Iran, Mongolia and Turkmenistan. Historically, it has also been reported in Afghanistan, Armenia, Azerbaijan, Georgia, Iraq, Jordan, Kuwait, Kyrgyzstan, Russia, Saudi Arabia, Syria, Tajikistan, Turkey and Ukraine. Four different subspecies are identified, mostly describing the present areal distribution: *E. hemionus hemionus* (Mongolia), *E. hemionus khur* (India), *E. hemionus kulan* (Turkmenistan) and *E. hemionus onager* (Iran). Another extinct subspecies was recognized in Syria, *E. hemionus hemippus*. The fossil record is not well studied, but crania and mandibles of the species have been reported from Narmada Valley in Central India [157], the Indian state of Gujarat [158], and the Son Valley in Northern India [159]. This species is distinguished from *E. namadicus* by its smaller size and smaller protocones on the premolars. Radiocarbon dates from these deposits suggest that this species entered South Asia during the last glacial period, most likely from West Asia [160,161]. The most recent genetic analyses suggest that the onager and kiang populations diverged evolutionarily ca. 0.4–0.2 Ma [28].25. *Equus kiang*, Moorcroft, 1841 [162] (0.0 Ma). Moorcroft [162] did not designate any holotype or lectotype but provided a general description of the species [162]. *Equus kiang* is known as the Tibetan ass, with a distribution in China, Pakistan, India, Nepal and, possibly, Bhutan. Three subspecies have been identified, *E. kiang kiang*, *E. kiang holdereri* and *E. kiang polyodont*, with different authors pointing out their subspecific statuses [162,163,164,165]. At the present time, no paleontological information is available for *E. kiang*. Jonsson et al. [28] distinguished *E. kiang* from *E. hemionus* as a valid taxon. The most recent morphological cladistic analysis found *E. kiang* and *E. hemionus* to be stenonine horses [10].26. *Equus przewalskii*, Poliakov, 1881 [166] (~0.1–0.0 Ma). The holotype, a skull (number 512) and skin (number 1523), are found in the collection of the Laboratory of Evolutionary Morphology, Moscow (museum exposition), originally obtained by N. M. Polyakov in Central Asia, southern Dzhungaria, in 1878 [129]. For a complete description of the species see Groves [167] and Grubb and Groves [168]. It is an extant species of caballine horse that is found in small geographic areas of Central Asia, although, historically, it once ranged from Eastern Europe to eastern Russia [169]. In China, *E. przewalskii* is common in the Late Pleistocene (~0.1–0.012 Ma) sites in the northern and central regions of the country, but it is absent in the Holocene, except in northwestern China [170]. It shows close morphological similarities with *Equus ferus* (see below), and both are members of caballine horses [10,28,153]. While genomic analyses have shown that the Przewalski’s horses are the descendants of the first domesticated horses from the Botai culture in Central Asia (Kazakhstan) around 5.5 ka [171,172], subsequent morphological studies have shown that Botai horses are not domestic horses but harvested wild Prezwalski’s horses [173].

### 4.2. Indian Subcontinent

The Siwalik Group and co-eval sediments of the Himalayan Foreland Basin preserve an exceptional record of hipparionine and equine horses. The earliest lineage, *Cormohipparion*, appears in the record at approximately 10.8 Ma. The diverse indigenous *Sivalhippus* lineage ranges from 10.4 to 6.8 Ma, “*Hipparion*” from 10 to 9.6 Ma and *Cremohipparion* from 8.8 to 7.2 Ma [174]. Across the Mio–Pliocene boundary, a turnover in hipparion taxa seems to have taken place, with the older lineages being replaced in the Late Pliocene by *Plesiohipparion* and *Eurygnathohippus* along with a distinct but poorly known species “*Hippotherium*” *antelopinum* [119,175]. These Late Pliocene hipparionines are replaced by stenonine equids represented by *Equus sivalensis* and a small species of ass-like *Equus* in the Early Pleistocene [176]. By the Middle Pleistocene, a third species of large stenonine horse, *Equus namadicus*, is common in peninsular India deposits; this species went extinct in the Late Pleistocene [177]. *Equus hemionus* doesn’t appear in the record until ~0.03 Ma [177], and *Equus caballus* is found in Holocene archaeological sites [178].

1. “*Hippotherium*” *antelopinum*, Falconer and Cautley, 1849 [179] (3.6–2.6 Ma). The lectotype is NHMUK PV M.2647, a subadult right maxilla fragment with P2-M3. It is a species of hipparionine horse from the late Pliocene age deposits between the rivers Yamuna and Sutlej in India. This taxon has been the subject of much nomenclatural confusion. Lydekker named the lectotype and along with the hypodigm, placed it within the genus *Hipparion*. Later authors [180,181,182] have referred Miocene hipparionine material collected on the Potwar Plateau in Pakistan to this táxon and reassigned the species to the genus *Cremohipparion*. However, given that the hypodigm of “*Hippotherium*” *antelopinum* comes from the Late Pliocene and does not preserve any apomorphies of *Cremohipparion*; we refer to the late Pliocene specimens as “*Hippotherium*” *antelopinum*, separate from the Potwar Plateau specimens from the Dhok Pathan Formation, which can still be taxonomically referred to as *Cremohipparion*, but a formal description of the species with a new type of specimen is required. A more comprehensive study currently in preparation will attempt to resolve this issue2. *Plesiohipparion huangheense*, Qiu et al., 1987 [122] (3.6–2.6 Ma). Jukar, et al. [119] identified NHMUK PV OR 15790, a mandibular fragment with p4-m1, originally classified as “*Hippotherium*” *antelopinum*, as *P. huangheense* from the Late Pliocene of the Siwalik Hills.3. *Eurygnathohippus* sp. (3.6–2.6 Ma). Four mandibular cheek teeth from the Late Pliocene of the Siwaliks from the Potwar Plateau in Pakistan and the Siwalik Hills in India were identified by Jukar et al. [175] as *Eurygnathohippus* sp. These specimens all bear the characteristic single pli-caballinid and ectostylid.4. *Equus sivalensis*, Falconer and Cautley, 1849 [179] (2.6–0.6 Ma). The lectotype is NHMUK PV M.16160, an incomplete cranium from the Siwaliks [183]. It is a large species of stenonine horse found in the Siwaliks of the Indian Subcontinent, ranging from the Potwar Plateau in the west to the Nepal Siwaliks in the east. The exact temporal distribution is unknown; however, based on paleomagnetic dating of the Pinjor Formation where the species was found, it likely ranges in age from 2.6 to 0.6 Ma [176,184]. However, some potentially older occurrences from just below the Gauss–Matuyama boundary (>2.6 Ma) have also been reported [185,186].5. *Equus* sp. (~2.2–1.2 Ma). This is small species of *Equus* with smaller slender metapodials has been reported from the Pinjor Formation of the Upper Siwaliks. This species has been referred to as *Equus sivalensis minor* [187], *Equus* sp. A [188] or *Equus* cf. *E.*
*sivalensis* [189]. A set of postcranial remains, including metapodials, astragali and phalanges, which were formerly tentatively referred to as “*Hippotherium*” *antelopinum*, are now believed to belong to this small species of *Equus* [138,184]. Based on specimens collected from the Mangla–Samwal Anticline and the Pabbi Hills in Pakistan, this species likely ranges in age from ~2.2 to 1.2 Ma [176]. Geographically, it ranges from the Pabbi Hills to the river Yamuna in the east.6. *Equus namadicus*, Falconer and Cautley, 1849 [179] (~0.5–0.015 Ma). The lectotype is NHMUK PV M.2683, an incomplete cranium from the Siwaliks [183]. It is a large-sized stenonine horse from the Middle and Late Pleistocene of the Indian Subcontinent. The stratigraphic range includes the Middle Pleistocene Surajkund Formation in Central India [190] and Late Pleistocene deposits throughout peninsular India [177]. However, Lydekker [191] reported some specimens from the uppermost Upper Siwaliks, which might suggest that the species extends back to the early Middle Pleistocene.

### 4.3. Europe

As in Eastern and Central Asia, the Mio–Pliocene boundary represents a relevant turnover of three-toed horses in Europe, with the extinction of the genera *Hippotherium* and *Hipparion* s.s. The Pliocene and Pleistocene are characterized by the persistence of *Cremohipparion*, and the dispersion of *Proboscidipparion* and *Plesiohipparion* [7], represented by five species. The *Equus* Datum is represented by the oldest species, *Equus livenzovensis*, at ca. 2.6 Ma in Russia, Italy, France and Spain [176], which led to the *Equus stenonis’* evolution and to the radiation of the African fossil species [10,13,184,192]. At the present time, we recognize 13 species in the genus *Equus* during the Pleistocene.

1. *Plesiohipparion longipes*, Gromova, 1952 [193] (7–3.0 Ma). The holotype is PIN2413/5030, a complete mt3 from Pavlodar [193]. It has been identified from Pavlodar (Kazhakstan), Akkasdagi and Calta (Turkey) [194] and Baynunah (UAE) [8,195]. In all cases, *Pl. longipes* was recognized by its extreme length dimensions of mc3s and mt3s. None of these attributions of *Plesiohipparion* display the characteristics of extremely angled and pointed metaconids and metastylids of *Pl. houfenense*, *Pl. huangheense*, *Pl. rocinantis* or *Pl. shanxiense.* If all these taxa are referable to *Plesiohipparion*, the chronologic range would be 7 Ma to the Early Pleistocene and the geographic range being from China to Spain.2. “*Cremohipparion*” *fissurae*, Crusafont and Sondaar, 1971 [196] (MN14-15). The holotype is an mt3 from Layna, Spain, figured in Crusafont and Sondaar [196] (Pl. 1). It was originally described as “*Hipparion*” *fissurae*. The species is recorded from the MN15 Pliocene localities in Spain [197] and more recently from the MN14 of Puerto de la Cadena [198]. The most recent analyses provided by Cirilli et al. [192] suggest an attribution to the genus *Cremohipparion*, yielding this species as the possible last representative of the genus in Europe. Its evolutionary framework is not yet defined.3. *Proboscidipparion crassum*, Gervais, 1859 [199] (ca. 4.0–2.7 Ma; MN14-16). Deperet [200] described and figured the sample from Roussillon, France, without assigning a holotype or lectotype. The sample figured by Deperet [200] (Pl. V, Figures 6–10; Pl. VI) represents the hypodigm for the species. The sample is a small-sized equid with remarkable similarities with *Pr. heintzi* [192,194]. No complete crania are known from this species, whereas it is well documented by isolated upper and lower cheek teeth and postcranial elements. Bernor and Sen [194] showed that *Pr. crassum* also has a very short mc3, whereas Cirilli et al. [192] gave substantial indications in cranial and postcranial elements for its attribution to the genus *Proboscidipparion*. This species is mostly known from the Pliocene of France (Perpignan, Montpellier) but also from Dorkovo (Bulgaria) and Reg Crag (England) [197,201,202].4. *Proboscidipparion heintzi*, Eisenmann and Sondaar, 1998 [203] (MN15). The holotype is MNHN.F.ACA49A, a complete mc3 from Calta, Turkey [203]. It was originally identified and described “*Hipparion*” *heintzi* [203], whereas Bernor and Sen [194] restudied and allocated this taxon to *Pr. heintzi*, recognizing the similarity of the Calta juvenile skull MNHN.F.ACA336 to Chinese *Pr. pater* in the retracted and anteriorly broadly open nasal aperture accompanied by very elongate anterostyle of dP2 [190]. It has only been reported from the locality of Calta.5. *Plesiohipparion rocinantis*, Hernández-Pacheco, 1921 [204] (3.0–2.58 Ma). The lectotype is a p3/4 figured in Alberdi [205] (Pl. 6, Figure 4). It is the largest three-toed horse from Europe. Qiu et al. [122], followed by Bernor et al. [124,206] and Bernor and Sun [207], recognized this species as being a member of the *Plesiohipparion* clade by cranial and postcranial morphological features. *Plesiohipparion rocinantis* is reported between 3.0 and 2.6 Ma [208,209] from La Puebla de Almoradier, Las Higuerelas and Villaroya (Spain); Roca-Neyra (France); Red Crag (England); Kvabebi (Georgia). This species may include “*Hipparion*” *moriturum* from Ercsi (Hungary) and the sample from Sèsklo (Greece) previously ascribed to *Plesiohipparion* cf. *Pl*. *shanxiense* [210]. It represents the last occurrence of *Plesiohipparion* in Europe at the Plio–Pleistocene transition [8,192].

Rook et al. [211] reported the occurrence of “*Hipparion*” sp. from the Early Pleistocene locality of Montopoli, Italy (2.6 Ma). The “*Hipparion*” sp. from Montopoli is represented by a single incomplete upper cheek tooth that, however, shows some morphological features distinct from the first *Equus*, *E. livenzovensis* occurring in the same locality [211]. Together with Villarroya, Roca-Neyra, Sèsklo, Guliazy and Kvabebi, the Montopoli specimen represents one of the last occurrences of three-toed horses in Europe and may, in fact, be a small species of *Cremohipparion*.

6. *Equus livenzovensis*, Bajgusheva, 1978 [212] (2.6–2.0 Ma). The holotype is POMK L-4, a fragmentary skull from Liventsovka [212] in the Early Pleistocene. The species represents the *Equus* Datum in Western Eurasia, Early Pleistocene localities dated at the Plio–Pleistocene boundary (2.58 Ma) such as Liventsovka (Russia), Montopoli (Italy) and El Rincón–1 (Spain) [184,212,213,214,215,216,217,218,219]. Recent research [214,216,220] suggests the occurrence of *E. livenzovensis* in Eastern European localities, dated between 2.58 and 2.0 Ma. The species shows the typical stenonine morphology, even though it is a large-sized horse.7. *Equus major*, Delafond and Depéret, 1893 [221] (?2.6–1.9 Ma). The hypodigm is represented by a P2-M1 and a 1ph3 figured by Delafond and Depéret [221] from Chagny, France. It is poorly represented in the Early Pleistocene of Europe, being represented by few remains and localities. It was described by an incomplete upper cheek tooth row and postcranial elements from the Early Pleistocene locality of Chagny (Central France). Following the ICZN guidelines, Alberdi et al. [216] established that *E. major* has priority over *Equus robustus* Pomel, 1853 [222]; *Equus stenonis* race *major* Boule, 1891 [223]; *Equus bressanus* Viret, 1954 [224]; *Equus major euxinicus* Samson, 1975 [225]. *Equus major* has been reported from the European sites of Senèze, Chagny, Pardines and Le Coupet (France); Tegelen (the Netherlands); East Reunion and Norfolk (England) [216], and, as reported by Forsten [214], it may also be present at Liventsovka (Russia). It is a large-sized monodactyl horse, the largest species of the European Early Pleistocene.8. *Equus stenonis*, Cocchi, 1867 [226] (2.45–ca. 1.6 Ma). The holotype is IGF560, a complete cranium from the Upper Valdarno Basin, Italy [13]. It was the most widespread *Equus* species during the Early Pleistocene (MNQ17 and MNQ18). In the last century, *E. stenonis* samples were identified under several subspecies as *E. stenonis vireti*, *E. stenonis senezensis*, *E. stenonis stenonis*, *E. stenonis granatensis*, *E. stenonis guthi*, *E. stenonis mygdoniensis*, *E. stenonis anguinus*, *E. stenonis pueblensis* and *E. stenonis olivolanus*. Recently, Cirilli et al. [13] reevaluated these subspecies, considering most of them to be ecomorphotypes of the same species, resulting in recognizing that *E. stenonis* is a monotypic, polymorphic species. At the present time, the species is reported as having occurred from Georgia to Spain in the circum-Mediterranean area as well as the Levant.9. *Equus senezensis*, Prat, 1964 [227] (2.2–2.0 Ma). The lectotype includes a left P2-M3 and two mc3s, figured in Prat [227] (Pl.1 Figure C; Pl.2 Figure D,E). It is a medium-sized horse, distributed in France and Italy. Earlier referred to as *E. stenonis senezensis* [227], it has been recognized as a different species by Alberdi et al. [216] and Cirilli et al. [13]. It is morphologically similar to the European *E. stenonis* but with a reduced size. It is reported from the type locality of Senèze and possibly from Italy between 2.2 and 2.1 [228,229].10. *Equus stehlini*, Azzaroli, 1964 [230] (1.9–1.78 Ma). The holotype is IGF563, an incomplete cranium from the Upper Valdarno Basin. It is the smallest Early Pleistocene *Equus* species. Over the last decades, it was considered both either a species or a subspecies of *E. senezensis* Azzaroli [216,230]. Cirilli [229] found that it can be considered a different Early Pleistocene species that probably evolved from *E. senezensis*. At the present time, *E. stehlini* is known only from Italy [228,229].11. *Equus altidens*, von Reichenau, 1915 [82] (1.8–0.78 Ma). The lectotype is a right p2 figured in von Reichenau [82] (Pl. 6, Figure 17) from Sussenborn, Germany. It is a medium-sized horse, intermediate in size between *E. stehlini* and *E. stenonis*, occurring in Western Eurasia in the late Early to early Middle Pleistocene. Originally described from the Middle Pleistocene in Süssenborn [82], over the last decades its chronologic range was extended to the late Early Pleistocene, representing the most widespread species after 1.8 Ma until the Middle Pleistocene [231,232]. Recently, Bernor et al. [14] reported its first occurrence in Dmanisi (Georgia, 1.85–1.76 Ma), supporting the hypothesis of a dispersion of this species from east to west, being part of a faunal turnover that included several other mammalian species during this time frame [14,233,234]. It was also identified in Moldova (Tiraspol, layer 5) [129]. Several populations of *E. altidens* have been identified as different subspecies (i.e., *E. altidens altidens* and *E. altidens granatensis*). Nevertheless, within the nomen, *E. altidens* may also be included in other European taxa such as *Equus marxi* von Reichenau, 1915 [82]; *Equus hipparionoides* Vekua, 1962 [235]; *E. stenonis mygdoniensis* Koufos, 1992 [236]; *Equus granatensis* Eisenmann, 1995 [13,216,232,233,237,238]. Its origin remains controversial. Guerrero Alba and Palmqvist [239] proposed a possible African origin, claiming it to be part of the *E. numidicus–E. tabeti* evolutionary lineage. This latter hypothesis was also reported by Belmaker [240]. Eisenmann [87] included *E. altidens* in the new subgenus *Sussemionus*, with other Early and Middle Pleistocene species. More recently, Bernor et al. [14] proposed a new evolutionary hypothesis, considering that *E. altidens* originated in Western Asia and is potentially related to living *E. hemionus* and *E. grevyi*.12. *Equus suessenbornensis*, Wüst, 1901 [241] (1.5–0.6 Ma). The lectotype is IQW1964/1177, a P2-M3 from Süssenborn, Germany. It is a large horse, larger than *E. stenonis* and *E. livenzovensis* but smaller than *E. major*. As for *E. altidens*, the species has been described from the Middle Pleistocene in Süssenborn [241], even if its best-known sample comes from the Georgian locality of Akhalkalaki (ca. 1.0 Ma) [242]. In addition to Akhalkalaki and Süssenborn, it has been reported in Central European localities such as Stránská Skála (Czech Republic) and Ceyssaguet and Solilhac (France) [216,243,244]. Over the last decades, its biochronologic range was extended to the late Early Pleistocene, with its earliest occurrence in the Italian localities of Farneta and Pirro Nord [216,231] and in the Spanish sites of Barranco León 5 and Fuente Nueva 3 [245].13. *Equus apolloniensis*, Koufos et al., 1997 [246] (1.2–0.9 Ma). The holotype is LGPUT-APL-148, a nearly complete cranium from Apollonia, Greece [246]. It is a peculiar species of the late Early Pleistocene, mostly recorded from the locality of Apollonia–1 (Mygdonia Basin, Greece) and, possibly, from other localities of the Balkans and Anatolia [244,246,247]. As reported by Gkeme et al. [247], this species differs from *E. stenonis* and other European Early Pleistocene *Equus*, with a distinct cranial morphology, and its size is intermediate between *E. stenonis* and *E. suessenbornensis*. Koufos et al. [246] interpreted *E. apolloniensis* as an intermediate species between *E. stenonis* and *E. suessenbornensis*, whereas Eisenmann and Boulbes [248] considered *E. apolloniensis* as “a step within the lineage of asses”.14. *Equus wuesti*, Musil, 2001 [249] (1.1–0.9 Ma). The holotypes are IQW1980/17067 and IQW1981/17619, two fragmentary mandibles with p2-m3 from Untermassfeld, Germany [249]. It has been established from the Epivillafranchian locality of Untermassfeld (Germany). Musil [249] reported isolated teeth, mandibles and long bones with primitive and derivate characteristics, and a larger size when compared with the widespread *E. altidens*. This evidence was also reported by Palombo and Alberdi [232], highlighting a more robust morphology of the postcranial elements. However, scholars disagree on its possible origin. Forsten [250] considered *E. wuesti* close to and derived from *E. altidens*, whereas others [249,251] recognized *E. wuesti* as the possible source for *E. altidens*. Nevertheless, considering the latest *E. altidens* discoveries [14,231,232,238], this last hypothesis seems to not be well supported. Palombo and Alberdi [232] suggest also that it can represent an ecomorphotype of *E. altidens*.15. *Equus petralonensis*, Tsoukala, 1989 [252] (ca. 0.4 Ma). The holotype is PEC-500, an mc3 from Petralona Cave, Greece [252]. It is a slender and gracile horse from Greece (Petralona Cave). However, the taxonomic status of this species has been actively debated. Forsten [250] considered *E. petralonensis* as being a member of the *E. altidens* group, together with the Early Pleistocene equids from Libakos, Krimini and Gerakaou. Eisenmann et al. [67] synonymized *E. petralonensis* with *Equus hydruntinus*, within the subspecies *E. hydruntinus petralonensis*. Despite the taxonomic controversy surrounding this species, *E. petralonensis* may be considered a stenonine horse because of its mandibular cheek tooth morphology [246].16. *Equus graziosii*, Azzaroli, 1969 [253] (MIS 6). No access number is available for the holotype, which is figured in Azzaroli [254] (Pl. XLV, Figure 1a,b) as a complete cranium from Val di Chiana, Arezzo. It is an enigmatic species from the late Middle–Late Pleistocene. It was described as a different species by Azzaroli [254] based on a partial cranium, mandibles and postcranial elements. According to Azzaroli [254], the cranium and maxillary and mandibular cheek teeth have typical asinine features, although some other authors have highlighted the morphological similarities with *E. hydruntinus* [244,255,256]. The evolutionary and phylogenetic position of *E. graziosii* is still questionable.17. *Equus hydruntinus*, Regalia, 1907 [257] (0.6–0.01 Ma). There is no holotype. The hypodigm includes isolated upper and lower molars and the fragments of radius and tibia from Grotta Castello (Sicily, Italy). It is known also as the European wild ass, which is a small-sized and slender horse from the Middle and Late Pleistocene in Europe. Its geographic range spans across Europe and is documented in numerous localities [244]. Apparently, the first specimens of this species were also found in Central Asia in several Uzbekistan Late Weichselian localities associated with the Late Paleolithic [129,258]. Its evolutionary history has been debated by many authors, who have proposed different scenarios such as a direct origin from *E. altidens* [150,259,260], *Equus tabeti* [237,239] or, more recently, as a new taxon arrival from Asia [228,238]. Nevertheless, the most recent DNA analyses relate *E. hydruntinus* as a morphotype of the modern *E. hemionus*, proposing the subspecies *E. hemionus hydruntinus* [261]. However, several morphological features distinguish *E. hydruntinus* from *E. hemionus* (for a detailed discussion, see [244]), allowing for the consideration of *E. hydruntinus* as a still valid name for fossil identification. The oldest remains of *E. hydruntinus* were reported from the Middle Pleistocene levels of Vallparadìs, ca. 0.6 Ma [262,263], although the species has been documented in Western Eurasia until the Holocene [244]. Different subspecies have been suggested including *E. hydruntinus minor* (Lunel Viel, France), *E. hydruntinus danubiensis* (Romania), *E. hydruntinus petralonensis* (Petralona Cave, Greece) and *E. hydruntinus davidi* (Saint-Agneau, France). As reported by Boulbed and Van Asperen [244], *E. hydruntinus* adapted to semiarid, steppe conditions with a preference for temperate climates, although it could tolerate limited cold conditions.18. *Equus ferus*, Boddaert, 1758 [264] (ca. 0.7–0.6 Ma). Boddaert [264] did not refer to any holotype or lectotype but gave a detailed description of the anatomical features of the species [264] (p. 159). It first appears in Western Eurasia in the early Middle Pleistocene, although a precise age is not available at the present time. The species has been questioned mostly from a taxonomic viewpoint, wherein many subspecies or different species have been erected to identify the Middle and Late Pleistocene fossil samples of the caballine horses as *Equus mosbachensis*, *E. mosbachensis tautavelensis*, *E. mosbachensis campdepeyri*, *E. mosbachensis micoquii*, *E. mosbachensis palustris*, *Equus steinheimensis*, *Equus torralbae*, *Equus achenheimensis*, *E. ferus taubachensis*, *E. ferus piveteaui*, *E. ferus germanicus*, *E. ferus antunesi*, *E. ferus gallicus*, *E. ferus latipes*, *E. ferus arcelini* and *Equus caballus*. These taxa are based on the size or morphological differences among the different fossil samples, representing an interesting case of morphological variability within the same lineage. These different subspecies have been considered to be chrono species by Eisenmann and Kuznetsova [265]. Nevertheless, van Asperen [266] noted that differences in the size and morphology of Middle and Late Pleistocene caballine horses can be observed, although they are not more variable than modern ponies or highly homogeneous groups such as Arabian horses or *E. przewalskii* [244]. Moreover, no unidirectional or evolutionary trend in size and shape can be identified, whereas morphology and size fluctuate over time [19,266]. For this reason, the proposal of Boulbes and van Asperen [244] to consider these species/subspecies as ecomorphological variants of the same species, *E. ferus*, seems the most parsimonious position. Moreover, as reported by van Asperen [267] following the ICZN, the correct species to indicate the wild caballine horses should be *E. ferus* and not *E. caballus*, which refers to domesticated forms. In addition, the genomic studies of Weinstock et al. [77] and Orlando et al. [153] support the genetic variation of the Middle and Late Pleistocene caballine horses.

### 4.4. Africa

The 5.3 Ma to 10 ka record of Equidae in Africa include two groups of Equinae: Hipparionini and Equini. Churcher and Richardson [268] provided a comprehensive review of African Equidae that was updated by Bernor et al. [269] for its taxonomic content, biogeography and paleoecology with some consideration of the molecular evolution. Churcher and Richardson’s review [268] of the literature that led to their revision was extensive, and the reader is referred to their article for a complete rendering of the record. We documented 10 (+4 not well defined) species of hipparions with three genera *Cremohipparion*, *Sivalhippus* and *Eurygnathohippus*, and 13 species of *Equus* in the African record, but there certainly could be more or less, although significant synonymies were cited by Bernor et al. [269] and pending new studies, especially on the *Equus* samples. The *Equus* Datum in Africa has been a matter of debate over the last years. Bernor et al. [269] and Rook et al. [176] reported the first known occurrence of the genus *Equus* in in East Africa at 2.33 Ma in the Omo Shungura Formation, member G (*Equus* sp.). Materials from these earliest occurring *Equus* are not well represented across the skull, mandible, dentition and postcranial elements. Nevertheless, recent research in the North African sequence of Oued Boucherit (Algeria) have recalibrated the localities of Aïn Boucherit, El Kherba and Aïn Hanech. The lowermost stratigraphic level of Ain Boucherit has been dated at 2.44 Ma [270], where *Equus numidicus*, *Equus* cf. *E.*
*numidicus* and *Equus tabeti* have been reported [270,271]. Therefore, this new North African age for the *Equus* Datum anticipates the earliest occurrence of *Equus* in the north rather than in East Africa.

1. *Cremohipparion periafricanum*, Villalta and Crusafont, 1957 [272] (6.8–4.0 Ma). The lectotype is a P2-M3 figured in Alberdi [205] (Pl.3, Figure 3) from Vadecebro II, Spain. It is a small (dwarf) hipparion that is a close relative (or senior synonym) of *Cr. nikosi* from the Quarry 5 levels of Samos, Greece, dated 6.8 Ma [206]. Fragmentary remains of *Cr.* aff. *periafricanum* have been reported from Tizi N’Tadderth, Morocco [273], and Sahabi, Libya [274].2. *Eurygnathohippus feibeli*, Bernor and Harris, 2003 [275] (6.8–4.0 Ma). The holotype is KNM-LT139, a partial right forelimb including a fragmentary radius, mt3, a1ph3, a2ph3, partial mc2, a1ph2, a2ph2, a2ph3 and a partial mc4 [275]. There were additional dental and postcranial elements from Lower and Upper Nawata that were referred to this species. Bernor and Harris [275] suggested that the Ekora 4 cranium, ca. 4.0 Ma, was a late surviving member of *Eu. feibeli.* Whereas Churcher and Richardson [268] recognized “*Hipparion*” *sitifense* in the North African Late Miocene–Pliocene horizons, Bernor and Harris [275] and Bernor and Scott [276] noted that the type material described by Pomel [277] could not be located and could potentially be confused either with *Cr. periafricanum* or *Eu. feibeli.*3. *Sivalhippus turkanensis*, Hooijer and Maglio, 1973 [278] (6.5–4.0 Ma). The holotype is KNM-LT136, an adult female cranium. Bernor and Harris [275] assigned this as being a species of *Eurygnathohippus*, *Eu. turkanense.* Subsequent studies of the *Sivalhippus* clade [174] and by Sun et al. [279] demonstrate the close identity of cranial, dental and, in particular, postcranial anatomy of *Si. turkanensis* and *Si. perimensis* and the extension of this genus into China and Africa in the Late Miocene.4. *Eurygnathohippus hooijeri*, Bernor and Kaiser, 2006 [280] (5.0 Ma). The holotype is SAMPQ-L22187, a complete adult female skull with associated dentition, mandible and dentition and postcranial characteristics from the earliest Pliocene Langebaanweg E Quarry, South Africa [280]. Hooijer [281] originally described the specimen under the nomen “*Hipparion*” cf. *H*. *baardi.*
5. *Eurygnathohippus woldegabrieli*, Bernor et al., 2013 [282] (4.4–4.2 Ma). The holotype is ARA-VP-3/21, an incomplete mandible [282]. The hypodigm include also 156 dental and postcranial specimens from 14 localities at Aramis (Middle Awash), Ethiopia. The type specimen is ARA-VP-3/21 a mandible including symphysis, right partial ramus with p2 and p3, left ramus with p2 and 3 preserved and pr-m3 poorly preserved. Mandibular incisor teeth and canines, if originally present are lacking.6. *Eurygnathohippus afarense* Eisenmann, 1976 [283] (KH3, Hadar, ca. 3.0 Ma). The holotype is AL363-18, a partial cranium from the Kada Hadar Member [283]. “*Hipparion*” *afarense* was nominated for skeletal material originating from the Kada Hadar 3 horizon, Hadar, Ethiopia. Eisenmann [283] also referred a mandible, AL177-2, to *E. afarense.*
7. *Eurygnathohippus hasumense*, Eisenmann, 1983 [284] (3.8–3.2 Ma). The holotype is KNM-ER 2776, a p4-m2 from zones B and C of the Kubi Algi Formation [284]. She included cheek teeth of common morphology from the Chemeron Formation, Kenya and the Denen Dora Member of the Hadar Formation, Ethiopia. A cranium with associated mandible was included in this hypodigm, AL340-8 [269] (Figure 13 in Bernor et al. [269]), and a partial skeleton including cheek teeth and complete postcranial elements, AL155-6 from DD2, ca. 3.2 Ma. Bernor et al. [281,285] analyzed a series of Ethiopian *Eurygnathohippus* establishing a phyletic relationship that currently includes *Eu. feibeli*, *Eu. woldegabrieli*, *Eu.* “*afarense*”, *Eu. hasumense* and *Eu. cornelianus.*
8. *Eurygnathohippus pomeli* Eisenmann and Geraads, 2006, [286] (ca. 2.5 Ma). The holotype is AaO-3647, an almost complete, but transversely crushed skull [286]. It was originally described as “*Hipparion*” *pomeli.* Reference [286] reported a well-preserved assemblage of hipparionini Ahl al Oughlam near Casablanca. Eisenmann and Geraads [286] argued that the sample is homogeneous and biochronologically correlative with eastern African faunas that are ca. 2.5 Ma, roughly contemporaneous with Omo Shungura D. Bernor and Harris [275] recognized this as a species of *Eurygnathohippus*. The northward extension of Plio–Pleistocene *Eurygnathohippus* into North Africa is remarkable as was its extension into India at this time [175].9. *Eurygnathohippus cornelianus*, van Hoepen, 1930 [287] (ca. 2.6–1.0 Ma). The hypodigm includes a mandibular dentition from Cornelia, Orange Free State, with hypertrophied i1s and i2s and atrophied i3s placed immediately posterior to the i2s [287] (plates 20–22). Leakey [288] (pl. 20, 4 figures) reported the occurrence of “*Stylohipparion albertense*” (=*Eu. cornelianus*) from Bed II, Olduvai Gorge, Tanzania based on premaxillae and mandibular symphyses with identical incisor morphology. Hooijer [281] reported an adult skull from Olduvai BKII which he referred to *Hipparion* cf. *H*. *ethiopicum* which is likely a member of the *Eu. cornelianus* lineage. Eisenmann [284] did not recognize the existence of *Eu. cornelianus* at Olduvai, but referred an immature cranium, KNM-ER3539 to *Eu. cornelianus.* Armour-Chelu et al. [289] and Armour-Chelu and Bernor [290] argued that the first evidence of this clade may be from the Upper Ndolanya Beds, Tanzania, circa 2.6 Ma. It is also likely present in the Omo Shungura F, dated 2.36. Bernor et al. [8,269] advanced the hypothesis that *Eu. cornelianus* is a member of an evolving lineage that occurred in East and South Africa between ca. 2.4 and less than 1 Ma. The specimen/locality content of *Eurygnathohippus* is currently under investigation.10. *Hipparion* (*Eurygnathohippus*) *steytleri*, van Hoepen, 1930 [287]. Van der Made et al. [291] argued that the author named *Hipparion steytleri* based on a right M1/1, left M3 and left m1-2 that formed a type series. Van Hoepen [287] also named *Eu. cornelianus* on the basis of a mandibular symphysis. Van der Made et al. [291] considered the nomen *H. stytleri* to have priority over *Eu. cornelianus*, but the cheek teeth of *H. stytleri* are of insufficient diagnostic value in themselves to define a valid species as stipulated by Article 75.5 of the ICZN. On the other hand, Van Hoepen [287] exercised considerable foresight in recognizing the highly derived state of the type specimen mandible of *Eu. cornelianus* because of its very wide mandibular symphysis with hugely hypertrophied i1 and i2 with very reduced, peg-like i3 situated immediately posterior along the mid-line of i2. Moreover, Leakey [288] illustrated a series of *Eu. cornelianus* mandibular symphyses and premaxillae from Olduvai Bed 1. Bernor et al. [292] described a juvenile skull of *Eu. cornelianus* (RMNH67) from Olduvai Gorge [293] which has a long preorbital bar, faint preorbital fossa and a dP2 with an extended anterostyle. *Eurygnathohippus cornelianus* sensu strictu has a known chronologic range of 2–1 Ma, but the lineage apparent extends lower in time.11. *Hipparion* (*Eurygnathohippus*) *libycum* Pomel, 1897 [273]. The hypodigm includes a left p3/4 figured by Pomel ([273], pl. 1, Figures 5–7), two lower cheek teeth from “carriers des gres ouvertes a la campagne Brunie” in Oran (pl. 1, Figures 1–7) and a distal epiphysis of a third metatarsal from “carriers de grès du quartier”, St-Pierre, Oran. Hopwood [289] assigned the left p3/4 figured by Pomel ([273], pl. 1, Figures 5–7) as a lectotype [287]. It shows gracile metapodials. Van der Made et al. [287] reported that the original type specimens from Oran are in the Central Faculty of Algiers (MGFCA) but have provided no accession numbers for the holotype.12. *Hipparion* (*Eurygnathohippus*) *ambiguum*, Pomel, 1897 [277]. The hypodigm includes a right P2 from Beni Fouda (Ain Boucherit) [277] (pl. 2, Figures 2–4). Previously the repository was unknown. Van der Made et al. [291] (Figures 2a,b and 3), redrafted an image by Pomel [277] and reported that the specimen is currently maintained in the Central Faculty of Algiers (MGFCA) but offered no photographic images or measurements of this specimen. Van der Made et al. [291] have provided no accession number for the holotype. The type locality of Aïn Boucherit has a magnetostratigraphic date of 2.44 [291] whereas the East African localities referred to by Van der Made et al. [291] have an age range of 3.8–1.2 Ma [269].13. *Hipparion* (*Eurygnatohippus*) *massoesylium*, Pomel [277]. The hypodigm includes five teeth from “puits Carouby” and “aux portes d’Oran” (also Puits Karoubi), left P4, M1-3 and right M3 (pl. 1, Figures 8–10). Van der Made et al. [291] considered these specimens to be the holotype of the species. Van der Made et al. [291] have reported that the holotype is kept in the Central Faculty of Algiers (MGFCA) but have provided no accession numbers for the associated cheek teeth.14. “*Hipparion*” *sitifense*, Pomel, 1897 [277]. The hypodigm includes four specimens including two teeth from St. Arnaud, a calcaneum from a nearby locality and a tooth figured by Thomas [294]. Pomel figured an M1/2 (Pl. 1, Figures 13–16), a right P4 (Pl. 1, Figures 11 and 12) and a calcaneum (Pl.2, Figures 9 and 10). The M1/2 figured by Pomel represents the lectotype [291]. The latter authors re-figured the original illustrations from Pomel [277], but this figure does not have a scale bar, and the specimens do not have formal institutional accession numbers. Eisenmann [295] stated that it is not known where the “type specimens” from Saint Arnaud et al. are, whereas Van der Made et al. [291] report that the original material is in the Central Faculty of Algiers (MGFCA; p. 44). Van der Made et al. [291] nominated a lectotype citing a specimen figured by Pomel [277] (pl. 1, Figures 13–15) without specifically designating a specimen accession number, institution, element and providing a redrafted original figure of the specimen without the benefit of a scale. It cannot be known if “*H. sitifense*” is referable to the genus *Hipparion* or other small hipparionins *Cremohipparion* or *Eurygnathohippus* (for which mandibular cheek teeth are needed). Arambourg [296,297] reported a specimen “from the type locality (for which we cannot be certain)” described and figured a mandibular specimen “with rounded metaconid-metastylid and no ectostylid”. It should be noted that this material is not a legitimate sample of “the original type series of Pomel, 1897”. Arambourg [296,297] assigned material from Ain el Hadj Baba, Mascara, Saint Donat and Aïn el Bey to “*H. sitifense*”. These are all small sized.

The *Equus* Datum in Africa has been a matter of debate over the last years. Bernor et al. [269] and Rook et al. [176] reported the first known occurrence of the genus *Equus* in East Africa at 2.33 Ma in the Omo Shungura Formation, member G (*Equus* sp.). Material of these earliest occurring *Equus* are not well represented across the skull, mandible, dentition and postcranial elements. During this interval of time, the most common European species is *E. stenonis* and possibly some late surviving populations of *E. livenzovensis* [13], *E. eisenmannae*, and *E. sanmeniensis* in China. Nevertheless, recent research in the North African sequence of Oued Boucherit (Algeria) have recalibrated the localities of Aïn Boucherit, El Kherba and Aïn Hanech. The lowermost stratigraphic level of Aïn Boucherit has been dated at 2.44 Ma [270], where *Equus numidicus*, *Equus* cf. *E. numidicus* and *Equus tabeti* have been reported [270,271]. Therefore, this new North African age for the *Equus* Datum anticipates the earliest occurrence of Equus in the North rather than in East Africa.

15. *Equus numidicus*, Pomel, 1897 [277] (2.44–1.2 Ma). The hypodigm includes a right P2 [273] (Pl. 2, Figure 2). Arambourg reported cranial and postcranial elements from Aïn Boucherit and Aïn Jourdel (Pl. 18, Figures 6 and 7; Pl. 19–20; Figures 58 and 62). It is a medium-sized horse approximately the size of a large zebra that originated from Aïn Boucherit and Aïn Hanech, Algeria, ca. 2.44–2.0 Ma [270,271,295]. No complete crania are known, although incomplete cranial remains associated with postcranial elements have been reported [296]. The evolutionary relationships of *E. numidicus* are not well defined. Azzaroli [298] pointed out a possible relationship to *E. stenonis*, although it was no longer investigated. However, some anatomical features of the upper and lower cheek teeth and postcranial elements resemble those of *E. stenonis* [297,298].16. *Equus tabeti*, Arambourg, 1970 [297] (2.44–1.2 Ma). The holotype is a partial palate (1949.2:773) figured in Arambourg [297] (Pl. 21, Figure 3). It is a medium-small sized species of *Equus* with “asinine” maxillary cheek teeth, stenonine mandibular cheek teeth and slender third metapodials and phalanges [284,297]. No complete crania are known, even if some incomplete crania and mandibles have been reported by Arambourg [297]. The type material originates from Aïn Hanech, Algeria [297]. Geraads [299] estimated the age of Aïn Hanech to be 1.2 Ma. Eisenmann [284] reported the possible presence of *Equus* cf. *tabeti* at Koobi Fora, Kenya believing that it is a primitive ass and may have been derived from *E. numidicus.* The recent analyses of Duval et al. [270] suggest an earlier occurrence of *E. tabeti* and *E. numidicus* at 2.44 Ma in the lowermost levels of Aïn Boucherit. Beside North Africa, *E. tabeti* is reported from Early Pleistocene sites of “Ubeidiya”, Bizat Ruhama and Qafzed (Levantine corridor [300,301,302]) and from East Africa [183].17. *Equus koobiforensis*, Eisenmann, 1983 [284] (2.1–1.0 Ma). The holotype specimen is KNM-ER 1484, a complete skull recovered from the *Notochoerus scotti* Zone, Area 130, just below the KBS Tuff, ca. 1.9 Ma. Eisenmann [284] reported a number of close dental similarities shared by *E. koobiforensis* and European *E. stenonis* but did not suggest a direct phylogenetic relationship between these taxa. Azzaroli [258] stated that *E. koobiforensis* was essentially a Grevy’s zebra. Bernor et al. [183] cited the likely evolutionary relationship between North American Pliocene *E. simplicidens*, European *E. stenonis* and *E. koobiforensis.* Cirilli et al. [10] demonstrated the explicit cladistic relationships between the *E. simplicidens* -*E. stenonis*-*E. koobiforensis*-*E. grevyi* clade, whereas Cirilli et al. [13] reinforced this result on robust statistical grounds. No precise age is available for its last occurrence in the fossil record.18. *Equus oldowayensis*, Hopwood, 1937 [293] (1.9–1.0 Ma). The holotype is a lower jaw from an animal approximately 2 years old [293] (Figures 1 and 2; Catalogue Number VIII, 353, in the Bayerische Paläontologische Staatssammlung, Munich). Hopwood [293] also designated a lower incisive region with the left incisors and right first incisor (BMNH M14199) as the paratype. The original Olduvai collection deposited in Munich, which included the type of *E. oldowayensis*, was destroyed together with its catalogue, during WW II (K. Heissig personal communication with Churcher and Hooijer; [269]). *Equus oldowayensis* is usually reported from the type locality of Olduvai (1.8 Ma, Tanzania), but no precise ages are available for its chronologic range. Recently, Bernor et al. [183] reported an incomplete cranium from Olorgesailie (ca. 1.0 Ma, Kenya).19. *Equus capensis*, Broom, 1909 [303] (ca. 2.0?–? Ma). It was a large-bodied horse estimated to be 150 cm at the withers, with a body mass of approximately 400 kg [304]. It originated from South Africa. Churcher [305] synonymized *E. helmei*, *E. cawoodi*, *E. kubmi*, *E. zietsmani* and specimens of *E. harrisi* and *E. plicatus* into *E. capensis. Equus capensis* was widely distributed in the Plio–Pleistocene of South Africa, although no information is known about its first and last occurrence in the fossil record.20. *Equus mauritanicus*, Pomel, 1897 [277] (1.0 Ma). The hypodigm include isolated teeth and postcranial elements, figured in Pomel [277] (Pl.3–8). It is reported from a large sample from Tighenif (Algeria). Churcher and Richardson [268] referred *E. mauritanicus* to a subspecies of *E. burchellii* (=*E. quagga mauritanicus*). Eisenmann [284,295] recognized *E. mauritanicus* as a distinct species of *Hippotigris* and claimed similarities to *E. stenonis* in the dentition. Eisenmann [284] further introduced the notion of cross-breeding between *E. mauritanicus* and Quaggas. Churcher and Richardson [268] reported an extensive distribution of *E.* (*Hippotigris*) “*burchelli*” (=*quagga*) from North, East and South Africa. The possible relationships of *E. mauritanicus* with plain zebras have been suggested by Eisenmann [295] and more recently from Bernor et al. [14] by morphological analyses. Beside the type locality of Tighenif, *E. mauritanicus* is reported also in Oumm Qatafa (Juden Desert, Egypt, Middle Pleistocene [306]).21. *Equus melkiensis* Bagtache et al. 1984 [307] (Late Pleistocene). The holotype is a short mc3 (I.P.H. Allo. 61–1314) recovered from Allobroges, Algeria and of latest Pleistocene age. Eisenmann [236] reported *E. melkiensis* also from Morocco. This species has been identified also at Gesher Benot Ya’akov and Nahal Hesi (Israel; [87,301]) and Oumm Qatafa (Egypt, Middle Pleistocene, [306]).22. *Equus algericus*, Bagtache et al. 1984 [307] (Late Pleistocene). The holotype is IPH61-103, a m2 from Allobroges, Algeria [307] (Figure 1). It is reported to be a caballine species with a withers height of approximately 1.44 m. *Equus algericus* was also reported from Morocco [308], which are purported to have the characteristic caballine metaconid-metastylid (=double knot) morphology.23. *Equus grevyi*, Oustalet, 1882 [309] (0.5–0.0 Ma). Oustalet [309] (v.10, pp. 12–14) described the anatomical features of the species, with an associated illustration (Figures 1 and 2). He also reported that the living animal was donated to the Museum National d’Histoire Naturelle in Paris, where the holotype should be kept. *Equus grevyi* is the largest living wild equid with a withers height of 140–160 cm. Azzaroli [259], Bernor et al. [14,183] and Cirilli et al. [10,13] all have cited the close relationship between European *E. stenonis*, *E. koobiforensis* and *E. grevyi. Equus grevyi* is currently distributed in the arid regions of Ethiopia and Northern Kenya and has recently vanished from Somalia, Djibouti and Eritrea [269]. Eisenmann [284] recognized *Equus* cf. *E*. *grevyi* from the *Metridiochoerus compactus* zone, the Guomde Formation and Galana Boi beds of Kenya based on both cheek tooth and postcranial remains. Bernor et al. [183] and Cirilli et al. [10,13] have recognized *E. grevyi* as a terminal member of the *E. simplicidens–E. stenonis–E. koobiforensis–E. grevyi* clade. Recently, O’Brien et al. [310] reported an incomplete cranium ascribed to *E. grevyi* from the Kapthurin Formation (Kenya) dated between 547 and 392.6 ka. This age represents the best dated *E. grevyi* FAD, at the present time.24. *Equus quagga*, Boddaert, 1785 [264] (?1.0–0.0 Ma). Boddaert [264] did not identify a holotype or lectotype for *E. quagga* but gave a description of the anatomical features of the species (p. 160). It has a shoulder height ranging from a mean of 128 cm in males and 123 cm in females. *Equus quagga* is one of the most widely distributed African ungulates ranging from southern Sudan and southern Ethiopia to northern Nambia and northern South Africa. Several subspecies have been recognized including *E. quagga crawshaii*, *E. quagga borensis*, *E. quagga boehmi*, *E. quagga chapmani*, *E. quagga burchellii* and *E. quagga quagga.* Fossil remains have been reported from North to South Africa. Leonard et al. [311] suggest that the various subspecies of *E. quagga* differentiated between 120 and 290 ka. Fossil remains of quagga have been reported from South African Plio–Pleistocene karst deposits, but their certain identity in North and East Africa are somewhat elusive [269]. Pedersen et al. [312] have identified a South African region as the likely source for the origin of the plain zebras from which all extant populations expanded from at approximately 370 ka. Moreover, the genetic analyses of Pedersen et al. [312] have reported a remarkable gene flow in the extant *E. quagga* subspecies, highlighting the challenge of identifying the subspecific designation only by morphology, identifying at least four genetic clusters for *E. quagga boehmi* and *E. quagga crawshayi*.25. *Equus zebra*, Linnaeus, 1758 [313] (?0.5–0.0 Ma). Linnaeus [313] does not refer any holotype or lectotype but gives a description of the anatomical features of the species (p. 101). It is a medium-sized, long-legged zebra with a mean shoulder height ranging from 124–127 cm. There are two recognized sub-species, *E. zebra zebra* (Cape Mountain Zebra) and *E. zebra hartmannae* (Hartmann’s Mountain Zebra). Churcher and Richardson [264] report a relatively small sample from the Middle Pleistocene to recent fossil remains of *E. zebra* in South Africa.26. *Equus (Asinus) africanus*, Heuglin and Fitzinger, 1866 [314]. The lectotype is designated as a skull of an adult female collected by von Heuglin near Atbarah River, Sudan, and in Stuttgart (SMNS32026). *Equus* (*Asinus*) *africanus* are equines of small size and stocky build. Churcher [315] reported the earliest occurrence of this taxon from the middle of Bed II, Olduvai Gorge (>1.2 Ma). This identification was based on a single third metatarsal, which was short (231 mm) and slender, although not to the extent of *E. tabeti*. Two subspecies are recognized, *E.* (*A*.) *africanus africanus* von Heuglin and Fitzinger, 1866 [314] (Nubian wild ass) and *E.* (*A*.) *africanus somaliensis* Noak, 1884 [316] (Somali wild ass) [317,318,319]. The African wild ass *E. africanus* is widely believed to have been the ancestor of the domestic donkey [269] and, more recently, believed to be the descendant of the Early Pleistocene species *E. tabeti* [320]. Nevertheless, this last evidence has only been suggested and not yet proven.

## 5. Biochronology and Biogeography

Rook et al. [176] recently provided a comprehensive biochronology and biogeography for latest Neogene–Pleistocene *Equus*-bearing horizons of Eurasia, Africa and North and South America. Following this recent summary, we report here a synthesis of major Equinae evolutionary events for the last 5.3 million years across these continents. Berggren and Van Couvering [321] suggest the application of the term biochron for units of geologic time that are based on paleontological data without reference to lithostratigraphy or rock units. Mammal biochronologic scales have been developed for Europe (ELMA), Asia (ALMA), North America (NALMA) and South America (SALMA) and, most recently, for Africa (AFLMA). These timescales are variously expressed in terms of conventional mammal biostratigraphic zones or as land mammal ages (LMAs). Each timescale based on land mammals in different continental landmasses has its own history of development reflecting the uniqueness of the records and the extent to which faunal succession has been resolved. 

### 5.1. NALMA Timescale and Equinae Evolution

The Hemphillian NALMA ranges from approximately 8 Ma to about 4.9 Ma, with four faunal stages (Hh1-Hh4), and correlates with the Late Miocene to Early Pliocene in age, the most recent age (Hh4) extending over the Mio–Pliocene boundary (5.3–4.9 to 4.6 Ma). Equids recorded from this interval include the genera *Dinohippus* (i.e., *D. leidyanus*, *D. interpolatus*, *D. leardi*, *D. spectans* and *D. mexicanus*,), the hipparions *Cormohipparion* (i.e., *Co. occidentale* and *Co. emsliei*), *Nannippus* (i.e., *Na. aztecus*, *Na. beckensis*, *Na. lenticularis* and *Na. peninsulatus*), *Neohipparion* (i.e., *Ne. eurystyle*, *Ne. gidley* and *Ne. leptode*), *Calippus* (i.e., *Ca. elachistus*, Ca. *hondurensis*), *Astrohippus* (i.e., *A. stocki* and *A. ansae*) and *Boreohippidion* (i.e., *B. galushai*). These last are taxa that hold-over from the Late Miocene.

Hemphillian Hh4 ranges into the Early Pliocene until 4.6 Ma, and a range of 4.9–4.6 Ma for the earliest Blancan has been proposed [322]. In fact, the succeeding Blancan NALMA has been defined by the first appearance in North America of arvicoline rodents circa 4.8 Ma. The Blancan has recently been subdivided into five intervals: Blancan I (4.9–4.62 Ma.), Blancan II (4.62–4.1 Ma.), Blancan III (4.1–3.0 Ma.), Blancan IV (3.0–2.5 Ma.) and Blancan V (2.5–1.9 Ma.) [322]. The equid *Dinohippus* is known to persist into Blancan I and II intervals, while *E. simplicidens*, *E. idahoensis* and *E. cumminsii* are known from the Blancan III and later Blancan assemblages. The diminutive hipparionine horse *Na. peninsulatus* is reported from the Blancan V interval but does not survive into the Irvingtonian. The Blancan IV/V boundary corresponds closely to the base of the Pleistocene. 

The Irvingtonian NALMA is subdivided into three units: Irvingtonian I (~1.9–0.85 Ma), Irvingtonian II (0.85–0.4 Ma) and Irvingtonian III (0.4–0.195 Ma). Early Irvingtonian assemblages includes the equids *E. scotti*, *E. conversidens* [60], and *Equus* (or *Haringtonhippus*) *francisci* (possibly including *E. calobatus*). 

Finally, the Rancholabrean NALMA extends from 0.195 to about 0.11 Ma with the onset of the Holocene. Common Rancholabrean equid species include *E. scotti*, *E. conversidens* and *E*. (or *Haringtonhippus*) *francisci*. *Equus occidentalis* is also abundant in the American southwest during this period. Fossils resembling *E. ferus* (e.g., *E. lambei*) have also been documented from Rancholabrean faunas.

### 5.2. SALMA Timescale and Equinae Evolution

Two lineages of Equidae occurred in South America during the Pleistocene, *Hippidion* and *Equus*. Although there are no records of *Hippidion* in Central or North America, most evidence suggests that both lineages originated in Holarctica and then migrated independently to South America during the important biogeographical event known as the Great American Biotic Interchange (GABI) [103,323,324]. However, there is no record for Equidae in South America until the beginning of Pleistocene, following the formation of the Isthmus of Panama from the early Pliocene onward (approximately 3 Ma) [103,324]. The earliest record of Equidae in South America is *Hippidion principale* from Early Pleistocene deposits (Uquian) of Argentina [323,324]. However, the age of the first record of *Equus* in South America is controversial. Traditionally, its earliest record is from middle Pleistocene (Ensenadan SALMA) of Tarija outcrops in Southern Bolivia, based on a biostratigraphic sequence at Tolomosa Formation and independently calibrated to occur between ~0.99 and ˂0.76 Ma [103]. Nevertheless, there is no consensus regarding the age of these deposits and some researchers consider the deposition in Tarija to have occurred only during the Late Pleistocene [325] (and references therein). Recently, it was proposed that only one species of *Equus* lived in South America during the Pleistocene, *E. neogeus* [99]. This species is considered a fossil-index for deposits of Lujanian SALMA (late Pleistocene-earliest Holocene; 0.8 to 0.011 Ma). Although *E. neogeus* was widely distributed in South America, only few localities are calibrated by independent chronostratigraphic data, indicating a Lujanian SALMA [98]. Therefore, the dispersal of *Equus* into South America occurred during the GABI, but if it is considered that *Equus’* earliest record is in the Late Pleistocene, it thus followed the fourth and latest phase of the GABI or *Equus* migrated to South America during GABI 3, considering its early record to be in the Middle Pleistocene [103]. All equids that occurred in South America during the Pleistocene (*Hippidion* and *Equus*) became extinct in the early Holocene [99,324]. 

### 5.3. ALMA Timescale and Equinae evolution

Equids first appear in Asia in the Miocene, between 11.4 and 11.1 Ma with the dispersal of the three-toed horse *Cormohipparion* [8,274,326]. Thereafter, a diverse assemblage of hipparionines is seen between ~10.2 and 6.0 Ma. Hipparionines are found across the Mio-Pliocene boundary which lies in the Dhok Pathan Formation (ca. 5.3 Ma) on the Potwar Plateau in Pakistan [327]. Hipparionines are rare in South Asia between 6.0 and 2.6 Ma, but recently, three taxa have been reported between 3.6 Ma and 2.6 Ma: “*Hippotherium*”, *Plesiohipparion*, and *Eurygnathohippus* [119,174]. The youngest indeterminate hipparionine records are dated paleomagnetically ~2.6–2.5 Ma, around the same time the first *Equus* occurs in South Asia [328,329]. *Equus* makes its appearance just above the Gauss-Matuyama boundary, which coincides with the Plio-Pleistocene transition. 

In China *E. eisenmannae*, a large yet primitive stenonine horse from the Longdan loessic section in Linxia Basin, Gansu Province, was magnetically dated to 2.55 Ma for its lower fossil-producing horizon, which is the earliest record of *Equus* in China. Although represented by poorly preserved fossils, *Equus* sp. from Zanda Basin in southern Tibet [118] was dated to 2.48 Ma, suggesting fast dispersion of *Equus* even in higher elevations. 

In the Siwaliks, remains of *Equus* have been found in sediments ranging from 2.6 to 0.6 Ma, a period termed the Pinjor Faunal Zone [328]. As noted in Bernor et al. [182], two morphotypes of *Equus* have been recorded—a large taxon called *Equus sivalensis* and a smaller taxon sometimes called *Equus sivalensis minor* for specimens from the the Upper Pinjor Formation near the town of Mirzapur [330], *Equus* cf. *E*. *sivalensis* from the Pabbi Hills [192] and *Equus* sp. (small) for specimens from the Mangla–Samwal anticline [190]. *Equus sivalensis* has been recorded from the entire temporal range of the Pinjor Faunal Zone [329]; however, the temporal range of the smaller horse appears to be restricted to ~2.2–1.2 Ma. 

*Equus sanmeniensis* is magnetically dated 1.7–1.6 Ma from the Shangshazui stone artifact site in the classical Nihewan Basin and to 1.66 Ma from the nearby Majuangou III hominin tool site. This species was also recorded from the *Homo erectus* site at Gongwangling, Lantian, Shaanxi Province, and magnetically dated to a slightly younger ~1.54–1.65 Ma. *Equus yunnanensis* from the *Homo erectus* site at Niujianbao in Yuanmou Basin was magnetically dated to 1.7 Ma. [331] In the Late Pleistocene, *Equus namadicus* and *Equus hemionus* are known from the Indian peninsula [332].

### 5.4. ELMA Timescale and Equinae Evolution

The Miocene fossil record (Vallesian and Turiolian Land Mammal Ages) as well as the Pliocene one (Ruscinian and early Villafranchian Land Mammal Ages) in Europe do not have monodactyl horses. These times are characterized by different hipparionine horses’ evolutionary lines and, during the Pliocene, three lineages of hipparions persisted in Europe: *Cremohipparion* (*C. fissurae*), *Plesiohipparion* (*Pl. longipes*, *Pl. rocinantis*) and *Proboscidipparion* (*Pr. heintzi*, *Pr. crassum*) [8,206,270]. It is during the early to middle Villafranchian transition, that *E. livenzovensis* first occurs in Southwest Russia and Italy at around 2.6 Ma (beginning of middle Villafranchian; Early Pleistocene) and constitutes the regional *Equus* First Appearance Datum [10,13,184,192]. In Europe, the earliest representatives of the genus *Equus* co-existed with the last hipparionin horses (the genera *Plesiohipparion*, *Proboscidipparion* and *Cremohipparion*) in the Early Pleistocene [10], although at the present time the effective co-existance of *Equus* and hipparion is found in the localities of Montopoli (Italy) and Roca-Neyra (France).

*Equus livenzovensis* appears to be at the base of the radiation of the later lineage of fossils horses, the European Pleistocene *Equus stenonis* group (=stenonine horses). The European stenonine horses have been recently revised [10,13]. In addition to *E. livenzovensis*, the species (and their chronological ranges) included in this group are *E. stenonis* (end of middle Villafranchian to early late Villafranchian; Early Pleistocene, 2.4–1.7 Ma; [13]), *E. stehlini* (late Villafranchian; Early Pleistocene, 1.8–1.6 Ma; [229]), *E. altidens*, and *E. suessenbornensis* (end of the late Villafranchian to Early Galerian, Early Pleistocene to Early Middle Pleistocene, 1.6–0.6 Ma; [184]). The most relevant turnover for the *Equus* species occurs in the Middle Pleistocene (ca 0.6 Ma) with the first occurrence of *E. ferus* (or *E. ferus mosbachensis*) in Mosbach (Germany) and *E. hydruntinus* from Vallparadìs (Spain) [240]. The arrival of these two species marks the extinction of the Early and early Middle Pleistocene *Equus* species. 

### 5.5. AFLMA Timescale and Equinae Evolution

The Miocene to Pleistocene mammal record of Africa is overall less complete than the fossil record on other continents with no established land mammal age scheme for Africa at a continental scale was established until very recent times [333]. 

The completeness of the mammal fossil record across the continent is extremely variable with regions in which the Neogene record is totally missing and others (such as Kenya or Ethiopia) with a relatively continuous and well documented record [334]. 

Without an established continental-scale biochronology, Africa’s “biochronology” is based on stratigraphic ordering in different sedimentary basins and is largely dependent on radiochronology with limited use of magnetostratigraphy. As an example, Pickford [335] subdivided Miocene faunas from Kenyan sites into Faunal Sets I to VII; suggesting age spans for late Miocene sets are 12.0–10.5 Ma (V), 10.5–7.5 (VI), and 7.5–5.5 (VII). 

The Late Miocene–Early Pliocene boundary (Sugutan and Baringian Land Mammal Ages) is poorly represented in Africa. Hipparionine horses are first found in North and East Africa circa 10.5 Ma [8] the richest locality being the Algerian site of Bou Hanifia and the Ethiopian site of Chorora [269]. At the end of Late Miocene (Baringian), diversification of the hipparionine genus *Eurygnathohippus* (exhibiting evolutionary relationships to Siwalik hipparionines; [175]) was well underway, as was significant the branching by endemic elephants and marked successes by new bovid tribes and suines arriving from Eurasia. 

The Plio–Pleistocene time period has been rigorously studied biochronologically in Africa by temporal distributions of elephants and suids, often in conjunction with the dating of hominin finds. *Equus* first occurs during the Early Pleistocene, even if this occurrence event in Africa is delayed relative to Eurasia, where it is at ca 2.6 Ma. Indeed, the most recent results identify the presence of the genus *Equus* in Nord Africa at ca. 2.44 Ma (*E. numidicus*, [269]) and in East Africa in lower Member G of the Omo Shungura Formation ca. 2.33 Ma (Shunguran Land Mammal Age; [176]). The first occurring African *Equus* is apparently related to European *E. stenonis* [10,13]. Representatives of the genus *Equus* are two times as abundant as *Eurygnathohippus* during the Early Pleistocene (Natronian Land Mammal Age) like in the Ethiopian locality of Daka [336], with *Eurygnathohippus* sharply declining in its numbers in East and South Africa after 1 Ma (Naivashan Land Mammal Age). Unfortunately, little is known of the first occurrence of the living species. Recently, a report by O’Brien et al. [310] of an incomplete *E. grevyi* cranium from the Kapthurin Formation (Kenya, 547–392.6 ka) represents the best dated *E. grevyi* FAD.

### 5.6. General Remarks about Equinae Biochronology

Unlike high-resolution biostratigraphic tools available in the marine realm, mammalian biochronology is not permissive of recognizing strictly synchronous events at global scale. Nevertheless, a review of the currently available evidence of the Land Mammal Ages, defined and calibrated across different continents (either in North and South America, Eurasia and Africa), allows us to recognize major faunal change (corresponding to the limit between subsequent Land Mammal Ages) always correlatable within the magnetochronostratigraphic scale and to place in this framework the main evolutionary events occurring around the world along the Equinae evolutionary history.

## 6. Paleoecology

In this chapter, we present some new paleoecological insights in diet and body size on the subfamily equinae, with special remarks for the genus *Equus*.

### 6.1. Relationships of Diet, Habitats and Body Size in Equines, with Pleistocene Equus from Eurasia and Africa as a Particular Example

In general, equines of the genus *Equus* tend to have mesowear values indicating grazing diets, but there is considerable variation in diets and body size (Appendix A), which likely reflect differences in habitat preference and social strategies, as well as the effect of available vegetation in different paleoenvironments. Our new analyses based on the most extensive compilation of body mass and palaeodiet data of particularly in Eurasian equines largely confirm previous hypotheses of the relationship between body size, diet, behavior and environments of equines during the Neogene, and during the Quaternary in particular. A few main hypotheses have been presented for the main factors that affected body size evolution and body size variation (also within species) of equine horses during the Neogene and the Quaternary. First, resource availability and quality (mainly regulated by annual primary productivity as well as nutritional properties and chemical defenses of available plants) have been suggested to be a limiting factor for some of the small-sized equine species or populations [13,19,244,267]. Conversely, the positive effect of seasonally high productivity and high resource quality due to low plant defense mechanisms has been suggested to explain the particularly large body size of Pleistocene herbivorous mammals in general [337,338,339,340]. Third, the effects of differential habitat heterogeneity, social structures and population density (intraspecific competition) on limiting the body size of some *Equus* species/paleopopulations, especially purely grazing ones that were abundant in open environments, has been discussed [19,20,340]. Observations of modern and Pleistocene *Equus* populations indicate that large-sized species tend to have smaller group sizes and population densities and more mixed or browse-dominated diets than small-sized species which are graze dedicated [19].

Ordinary least squares multiple regression models indicate that both diet and productivity of environments (estimated NPP values) are related to the variation in body size, both between and within the species of *Equus* during the Pleistocene-present (Figure 1). The “All *Equus*” model includes a wide range of extant and Pleistocene *Equus* populations from Eurasia and Africa, and it shows a significant negative effect of mesowear score and a significant positive effect of NPP on *Equus* body mass (although there is remarkable scatter especially in the residuals of the NPP estimate effect on body mass). The pattern is similar in the most abundant Pleistocene species/lineages of European *Equus*, including *E. stenonis*, *E. altidens* and the Middle–Late Pleistocene caballines (*E. ferus* + *E. mosbachensis*), although the patterns were statistically less robust (but note the small sample size especially in the case of *E. stenonis* and *E. altidens*). The connection between diet and body size was particularly robust, especially for the all-*Equus* model, indicating that very large species of *Equus* had more browse-dominated diets than small and medium-sized species/populations (Figure 1). Furthermore, the association of large size and more browse-dominated diets is shown for Africa as well as Eurasia (Figure 2).

These results indicate that both primary production and differences in the dietary niche had an effect on the body size of *Equus*. Large size in *Equus* is associated with more browse-dominated diets and more productive paleoenvironments, while small and medium-sized horses typically occupied less productive environments and had more purely grazing diets. The association of diet and body size appears stronger than the association of estimated NPP and diet, except perhaps in *E. ferus/mosbachensis*. A possible explanation for this is that the diet includes a signal of niche partitioning between sympatric species of *Equus*. As Saarinen et al. [19] discuss, when a small and a large species of *Equus* occurred sympatrically, the larger species was typically less abundant in the fossil assemblage and (in all such cases, including the new ones added in the present study) had more browse-dominated diet. Thus, the effect of larger group sizes of grazing, gregarious populations of *Equus* on limiting their body size via the effect of larger population densities and more intense intraspecific competition appears to have been a significant mechanism limiting their body size, although there was also a more general positive effect of primary production on body size.

This model of the relationship of body size with diet (affected by available vegetation and dietary niche partitioning), population density (associated with dietary preferences and social strategies of different *Equus* species) and habitat (vegetation openness, heterogeneity and resource availability) based on data from Eurasian and African *Equus* is by and large supported by more general observations from earlier equines and from *Equus* from North America. In Eurasian and African hipparionines, large body size is typically associated with browse-dominated diets, such as in *Hippotherium* from the early late Miocene of Europe, while smallest body sizes tend to occur in grazing taxa, such as the small species of *Cremohipparion* from the circum-Mediterranean environments during the late late Miocene (Appendix A) [8].

Tracking the abrasion incurred on molars in deep time (hypsodonty index), the level of abrasion incurred by individuals cumulatively in their lifetimes (mesowear) and the short-term acquisition of microwear scars from exogenous grit and/or food items has shed light on dietary and environmental shifts through time for North American Equini. By using three dietary proxies with different temporal resolution capabilities, the amounts of different levels of dietary abrasion as well as the possible causes of this abrasion was elucidated. Highly hypsodont members of the subfamily Equinae appear in North America about 17.5 Ma (late early Miocene-Hemingfordian) [341]. It is at this time that high degrees of large pitting are found in their dental enamel, and they begin to show scratch textures regardless of dietary classification indicating heavy exposure to exogenous grit. Also, highly hypsodont equines first appeared in the late Middle Miocene (ca. 14 million years ago), well after the projected availability of pervasive open grasslands (earliest Miocene) [342,343] or even the latest Oligocene [344]. Thus, the appearance of hypsodonty in Miocene equines was not synchronous with the appearance of grasslands in North America, and exogenous grit appears to have been a contributing factor to increased exposure to abrasion and subsequent increases in crown height in fossil horses rather than grazing alone [21,345,346]. Mesowear patterns closely mirror hypsodonty trends (i.e., higher mesowear scores when hypsodonty increases.) The species of *Equus* shown in Appendix A have abrasive mesowear consistent with grazing. Thus, grass was an important dietary item as it is today. Also, as these taxa, while grazing, were feeding low to the ground, they would have encountered grit encroaching on their food items which would have contributed further to their abrasive mesowear. Even so, microwear has shown that roughly 80% of these taxa (Appendix A) seasonally or regionally engaged in mixed feeding, or in one case, browsing. This demonstrates that hypsodonty, although advantageous for consuming grass, does not preclude consuming leaves or other browse at times. The other and earlier Equini shown in Appendix A (i.e., *Cormohipparion*, *Pseudhipparion*, *Calippus*, *Hipparion*, *Dinohippus*, *Neohipparion* and *Nannipus*) exhibited a less grazing type mesowear and/or microwear indicative of a tendency to either consume grass or a mixture of grass and browse. These results are consistent with the fact that even after open grasslands became pervasive in North America, forest vegetation was apparently also available until the late Miocene [21,342,347].

Analyses of body mass and mesowear of Pleistocene *Equus* from Mexico and the USA (Alaska) show that the body sizes of most of the species were relatively small compared to the larger species from the Pleistocene of Europe, and all of these species had grass-dominated diets, although some variation occurred at a smaller scale than in the Pleistocene of Europe (Appendix A). The relatively small body size and lesser variation in size and diet in North American *Equus* paleopopulations compared to Europe could reflect less productive paleoenvironments in general, with the estimated NPP values for nearly all of the North American and Mexican localities being comparatively low (between ca. 299 and 730 g(C)/m^2^/a) (Appendix A). The highest estimated NPP (of the sites analyzed here) is at Rancho la Brea (ca. 839 g(C)/m^2^/a), which is also the only one of those localities that has a very large-sized species, *E. occidentalis*. Pérez-Crespo et al. [348] noted that the sympatric *E. conversidens* and *Equus* (or *Haringtonhippus*) *francisci* from Valsequillo, Mexico, had differences in diet and body size that corresponded with the “Eurasian and African *Equus* model”, with the larger species *E. conversidens* having more browse-based and the smaller *Equus* (or *Haringtonhippus*) *francisci* having more grazing dietary signal. The observation that equines in general tend to have heavily grazing diets in North America since the late Miocene makes sense (and more so than in the Eurasia and Africa, where more browsing and mixed-feeding forms occurred, even among Pleistocene *Equus*), as it explains the evolution of the prominent grazing adaptations of this group in North America. Interestingly, it seems that body sizes of North American equines were on average smaller than European equines, mostly lacking the very largest body size category (above ca. 550 kg in *Equus*), with *E. occidentalis* being the only exception. Species of this largest size category such as *E. major*, *E. suessenbornensis* and *E. mosbachensis* in Eurasia and *E. capensis* in Africa, were the ones with most browse-dominated diets during the Pleistocene. It is possible that the evolution of these giant species of *Equus* is associated with changes in the dietary niche and population densities in the more wooded, comparatively high-productivity paleoenvironments of Pleistocene Europe in particular, while the more open-adapted, grazing horses in North America and much of Africa and Central Asia attained more modest sizes due either to less productive environments or grazing, gregarious ecological strategies that limited individual body size.

### 6.2. North America

The major evolution and diversification of equids occurred in North America even though a number of successive dispersals took place to Eurasia. Equidae apparently evolved in isolation from Eurasia in North America from the middle Eocene to the late Oligocene [44]. During the Tertiary, equids were very widespread in North America. In fact, at most fossil localities, they are the most common medium- to large-sized mammals recovered [44]. Equids achieved their maximum diversity in the late Miocene [44,349] resulting in the evolution of the subfamily Equinae [350].

The Miocene was also a time of craniodental reorganization of the equid skull. [351,352]. These trends began in parahippine and merychippine equids and eventually led to the genus *Equus*, which is thought to have evolved from *Dinohippus* in the Pliocene [346,353]. The dramatic changes in equid skulls, teeth and limbs have long been thought to reflect evolutionary adaptations to a changing environment, often thought of as evidence that grasslands expanded during this time [354,355,356,357,358,359,360,361,362]. Gross changes in dental morphology reveal a true shift toward more abrasive diets including grass in the derived Equinae from the middle Miocene onward reflecting their adaptation to grazing in more open environments [346,363]. While grazing was clearly an important long-term dietary strategy for derived Equinae once they appear in North America, some also engaged periodically in short-term mixed feeding regionally or seasonally. 

Interestingly, it appears that there was less variation in the dietary ecology of North American derived equines than the species that encountered a wide range of environments from wooded to open in Eurasia. Middle Miocene equines such as *Co. quinni* and *Co. goorisi* were small sized and had mixed to grazing diets (Appendix A). The species of *Equus* from the Pleistocene of Mexico and North America mostly have relatively small body sizes and mostly grazing mesowear (Appendix A). The relatively small body sizes could be related to the grazing, gregarious lifestyle or the relatively low-productivity, open paleoenvironments of the North American paleopopulations summarized in this study. In Mexico and Southern USA, the small species *E. cedralensis* and *E. conversidens* had grass-dominated mesowear signals and occupied low-productivity, sometimes nearly desert-like environments as suggested by low estimated NPP values of their localities (Appendix A). *Equus mexicanus* and *E. scotti* were large (mean body mass between 450 and 480 kg), but not as large as the “very large”, predominantly mixed-feeding or browse-dominated “woodland” species in the Eurasia, such as *E. major* and *E. suessenbornensis*, both with average body masses around 550–600 kg (Appendix A). The only species in the “very large” size category from the North American sites is *E. occidentalis*. This species occupied a relatively high-productivity paleoenvironment in Rancho la Brea (Appendix A), where its mesowear signal indicates predominantly grazing diet [363]. However, the proportion of sharp cusps is also relatively high in the Rancho la Brea population of *E. occidentalis*, and microwear texture analyses indicate that it consumed a significant proportion of woody browse during the pre-LGM cool stages at Rancho la Brea, so at least periodically significant inclusion of browse is indicated for the diet of also this very large equine [364]. In Alaska, the small to medium-sized *E. lambei* occupied a cold mammoth steppe paleoenvironment with relatively modest estimated NPP, and its mesowear signal indicates a grazing or at least heavily grass-dominated diet (Appendix A).

### 6.3. Eurasia

The earliest hipparionines that dispersed from North America to Eurasia at the beginning of the late Miocene were medium-sized (around 160 kg in body mass) species of the genus *Cormohipparion*, such as *Co. sinapensis* from Sinap, Turkey. These were relatively slender and modest-sized and probably occupied relatively open environments from East Asia to Turkey [8]. Early on, however, the larger, more robust hipparionines of the genus *Hippotherium* emerged and were widespread across Eurasia. *Hippotherium primigenium* was a relatively large species (body mass 200–250 kg) with predominantly browsing diets that occupied primarily forest and woodland environments in Central Europe during the early late Miocene [8]. Considerable dietary variation occurred in *Hi. primigenium*, being purely browsing in the forested paleoenvironment of Höwenegg and more grass-dominated in the locally more open floodplain environment in Eppelsheim, in Germany [365,366]. Two species of *Hippotherium*, *H. primigenium* and *H. kammerschmittae*, from the later late Miocene had browse-dominated diets in Dorn Dürkheim, Germany [366]. During the later late Miocene (Turolian), hipparionines diversified in Eurasia, and included several species and lineages of different body size and dietary ecology. In the Mediterranean realm and Western Asia for example, medium-sized (ca. 100–200 kg) species of *Hipparion* and *Cremohipparion* species were mixed-feeders, whereas larger species of the genus *Hippotherium* (with body masses more than 200 kg) retained browse-dominated or mixed diets, while the small species of *Cremohipparion* (body mass less than 100 kg) had grazing diets (Appendix A) [8]. Hipparionines thus seem to by and large reflect the model of larger sizes being related to more browse-dominated diets and smaller sizes to grazing diets, as in the example of *Equus* during the Pleistocene in Eurasia. Similarly high diversity of hipparionines occurred in East Asia during the latest Miocene.

During the Pliocene, the diversity of hipparionines in Eurasia dropped drastically and there was a turnover in the species composition, with a few, in general large-sized (200–350 kg) species in the genera *Plesiohipparion*, *Baryhipparion* and *Proboscidipparion* surviving [8]. All these genera had mostly mixed-feeding diets, with *Plesiohipparion* and *Proboscidipparion* having wide geographic ranges from East Asia to Europe [8]. Considerable ecological flexibility seems typical, especially for *Proboscidipparion*. While *Pr. sinense* occupied relatively open environments in East Asia and had a mixed but relatively abrasion-rich mesowear signal [8], *Proboscidipparion* sp. from Red Crag, England (latest Pliocene, ca. 2.7 Ma) had a browse-dominated diet [8,367] and lived in a warm-temperate forest environment [368] (Appendix A).

The earliest species of *Equus* to disperse from North America to Eurasia were relatively large sized but ecologically quite generalized, grazing, open-adapted species such as *E. eisenmannae* in East Asia and *E. livenzovensis in* Western Asia and Europe. These taxa had average body masses around 500 kg and at least *E. eisenmannae* had mesowear values indicating typical grazing diet for the genus (Appendix A). At the beginning of the Pleistocene, the first of the specialized, very large-sized and robust woodland horses with mixed and even browse-dominated diets, *E. major*, emerged in Western Europe. This species typically occurs in Early Pleistocene sites in Europe where palaeoenvironmental proxies such as pollen records and large mammal ecometrics indicate relatively wooded and productive paleoenvironments such as in Red Crag (UK) and Tegelen (Netherlands) [19,369]. *Equus major* was one of the largest species of equid, with mean body mass around 600 kg, and maximum body mass of ca. 800 kg.

The common Early Pleistocene species of European *Equus*, *E. stenonis*, occurred in a wide range of localities suggesting broad tolerance of environmental conditions. It was a medium-large species of *Equus*, with mean body mass between 400 and 500 kg (Appendix A). Available paleodietary evidence indicates mostly grazing diets for *E. stenonis* [370,371] (Appendix A), but the small sample from East Runton, UK, had a more mixed dietary signal (Appendix A). Analysis of body mass, mesowear and the NPP of *E. stenonis* paleopopulations indicates a strong inverse relationship of the amount of grass in diet and body size (Figure 1). There was also a geographic pattern of body size in *E. stenonis*, with the Western European populations associated with more high-productivity environments having on average larger body sizes than Eastern European populations, which occurred in less productive environments [13]. *Equus senezensis* was a smaller species with mean body mass around 350 kg and a grazing diet and occupied mostly open landscape [372,373] (Appendix A).

Later during the Early Pleistocene, the large-sized *E. suessenbornensis* and the small-sized *E. altidens* became prominent species in Eurasia, being the dominant species there during the late Early and early Middle Pleistocene. *Equus suessenbornensis* was a very large-sized and robust species, comparable in size to the earlier Pleistocene Western European *Equus major* (with mean body mass within paleopopulations ranging from over 500 kg to slightly over 600 kg). Similar to other very large species of *Equus*, *E. suessenbornensis* typically had mixed to even browsing diets [19,374] (Appendix A), and although being widespread in Europe, it was typically less abundant than the small *E. altidens*, also where these two co-occurred. *Equus altidens* was the earliest identified hemionine and it shares interesting similarities in mesowear signal to extant hemionines. As in modern hemionines, most paleopopulations of *E. altidens* show heavily grazing mesowear signals [375] (Appendix A), but also relatively abundant association of low occlusal relief and sharp cusps in some localities [375]. This kind of mesowear signal suggests diet based mostly on grasses but also including a significant component of dry, open environment browse such as aridity-resistant shrubs [15]. Some populations also display microwear patterns compatible with a mixed diet suggesting a certain degree of dietary plasticity for this species [369,375] (Appendix A). Mean body mass estimates of *E. altidens* vary around 350 kg (Appendix A). In general, *E. altidens* tends to be associated with paleoenvironments where dental ecometrics of large herbivorous mammal communities indicate relatively modest primary production estimates (between ca. 700 and 900). In Guadix-Baza Basin, Andalucia, Spain, the paleoenvironments of *E. altidens* have been suggested to have been similar to present Mediterranean woodlands and forest-steppes [374,376,377]. The diet of *E. altidens* reflects differences in paleoenvironments, being purely grazing in Venta Micena and Vallparadìs (EVT12 layer; MIS 31) but more mixed in Barranco León and Fuente Nueva 3 in Guadix-Baza Basin, Andalucia, where the paleoenvironment was more Mediterranean forest or woodland type and in layer EVT7 (MIS 21) of Vallparadìs where environmental conditions became more humid, and seasonality might have increased following the “0.9 Ma event” [374,375]. In Süssenborn, Germany, this species occurred in a paleoenvironment which has been interpreted periodically cool and relatively open, but not periglacial, based on the faunal association [378].

The Middle Pleistocene marks the arrival of caballine horses in Eurasia, a significant turnover event. *Equus mosbachensis* (=*E. ferus mosbachensis*/*E. ferus*), the typical caballine during the early Middle Pleistocene in Europe, was a very large and robust form (mean body mass from over 500 kg to nearly 600 kg), and it displayed more diverse dietary adaptations including grazing, mixed or even browse-dominated diets [19,373,379] (Appendix A). Large, browse-dominated forms of this taxon are associated with relatively wooded paleoenvironments such as Boxgrove in the UK and Schöningen in Germany [19,380,381]. Even grazing populations can be found in habitats dominated by wooded landscapes (e.g., open woodlands), feeding also in closed environments such as in Fontana Ranuccio (0.4 Ma) [374,382] (Appendix A). The cold-stage paleopopulation of *E. mosbachensis* from Caune d’Arago, France, had somewhat smaller average body size and more grazing diet (Appendix A). The wild horse (*E. ferus*) was abundant and widespread in Eurasia during the late Middle and Late Pleistocene, with small-sized forms having more grazing dietary signals and occurring in sites with smaller estimated NPP than larger forms of the species (Figure 1, Appendix A). The smallest forms of *E. ferus* with most grazing dietary signals come from sites where associated paleobotanical evidence indicates very open and grass-dominated “mammoth steppe” environments, such as Brighton (MIS 6 glacial) and Gough’s Cave (MIS2 glacial) in UK (Appendix A) [383,384]. Conversely, large forms of *E. ferus* typically occurred in more wooded paleoenvironments, such as Grays Thurrock (MIS 9 interglacial) and Brundon and Ilford (MIS 7 interglacial) in the UK, and Taubach (last interglacial, MIS 5) in Germany, and had less purely grazing diets (Appendix A) [385,386]. Further east, a medium-large sized form of *E. ferus* (“*E. ferus latipes*”) had a grass-dominated diet at the locality of Kostenki 14 in western Russia (Appendix A), where it occupied a cool and arid steppe environment [387]. The northernmost populations from Taimyr and Yakutia, Northern Siberia, “*E. lenensis*”, mostly lived in cold, low-productivity steppe-tundra environments, although the northern edge of boreal forest advanced in these areas during warmer stages [388]. They are characterized by small average body size and grass-dominated mesowear signal, although some individuals show sharper and more high-relief cusps indicating inclusion of browse or non-grass herbaceous vegetation in their diet (Appendix A). *Equus hydruntinus* had a more limited range in Eurasia during the late Pleistocene, and similarly to other hemionines, it seems to have been associated with relatively open habitats and it consistently had grass-dominated diets (Appendix A) [389].

The extant equids in Eurasia are currently limited to the Central Asian hemionines (*E. hemionus* and *E. kiang*) and the Przewalski horse (*E. przewalski*). These are all relatively small-sized members of the genus, they all have grazing diets, and they occupy the steppe environments of Central Asia (Appendix A). Similar to *E. altidens*, the hemionines today have a comparatively high proportion of low and sharp mesowear among *Equus*, indicating some inclusion of “dry browse” or non-grass herbs in diet, in addition to grass (Appendix A).

### 6.4. Africa

The earliest hipparionine with palaeodietary evidence from Africa is *Cormohipparion* sp. from the late Miocene of Chorora, Ethiopia (ca. 8.5 Ma), which has a browse-dominated mesowear signal and medium body size (ca. 160 kg) (Appendix A) [8,274,390,391,392,393,394]. Since this earliest record, most of the equines in Africa show mesowear and other paleodietary evidence suggesting grass-dominated to grazing diets. The hipparionines of the genus *Eurygnathohippus* were relatively large in size (over 200 kg in mean body mass) and yet they had remarkably grass-dominated diets (Appendix A), unlike the large-sized hipparionines in Eurasia, which tend to have more mixed or browse-dominated diets. This could reflect adaptation of the African derived hipparionines of the genus *Eurygnathohippus* to graze in relatively productive, but grass-dominated savanna environments.

After the arrival of *Equus* in Africa in the Pleistocene, most of the African equine species had grazing diets and were of small body size compared to a much wider range of sizes and diets in Eurasia (Figure 2), probably reflecting similarity in their adaptations to grazing in grass-dominated African savanna environments. The only clear exception to this pattern is the very large-sized South African species *E. capensis*, which had a more mixed or even browse-dominated dietary signal, paralleling the relationship between diet and body size observed for the Pleistocene of Europe (Figure 2; Appendix A).

The extant African zebras (*E. quagga*, *E. grevyi* and *E. zebra*) all have relatively small body size compared with the large Pleistocene species of *Equus* (particularly in Eurasia) and they typically have some of the most purely grazing diets among the equids (Appendix A). The Grevy’s zebra (*E. grevyi*) is the largest of these species, and the largest extant species of wild *Equus*, but it has a relatively tall and slender morphology, with elongate metapodials compared to the Quagga, and mean body mass estimates are relatively modest (around 360 kg on average) compared to many of the Pleistocene taxa, resembling however those of *E. koobiforensis* from the Pleistocene of East Africa. Our mesowear data suggest that the proportion of sharp but low-relief cusps is higher in *E. grevyi* than the rest of extant zebras, indicating perhaps a somewhat higher proportion of dry browse such as dry-adapted shrubs in its diet (Appendix A). The Africa wild ass (*E. africanus*) also has a relatively high proportion of low and sharp mesowear, despite mostly grazing dietary signal, which could also reflect inclusion of dry browse in the arid environments of this species (Appendix A) [395].

## 7. Climate and Evolution

Figure 3 provides the distribution of Equinae in North America, 7–4 Ma. As with the succeeding climate and evolution maps, the numbers on these maps are tied to Appendix A. During this time frame, 10 genera are recognized from North America (*Calippus*, *Dinohippus*, *Cormohipparion*, *Nannippus*, *Neohipparion*, *Astrohippus*, *Boreohippidion*, *Pliohippus*, *Pseudohipparion* and *Equus*), 3 from Eurasia (*Plesiohipparion*, *Proboscidipparion* and *Cremohipparion*) and 3 from Africa (*Cremohipparion*, *Eurygnathohippus* and *Sivalhippus*). The Equinae in North America include Ca. *elaschistus*, Ca. *hondurensis*, *D. interpolatus*, *D. leardi*, *D. leydianus*, *D. spectans*, *Co. occidentale*, *Na. aztecus*, *Na. lenticularis*, *Na. peninsulatus*, *Ne. leptode*, *A. ansae*, *A. lenticularis*, *Bo. galushai*, *D. mexicanus*, *Co. emsliei*, *Plio. coalingensis*, *Ne. eurystyle*, *Ne. gidleyi*, *Ps. simpsoni*, *Na. beckensis*, *Equus/Plesippus simplicidens*, *E. cumminsi*, *E. enormis* and *Equus/Plesippus idahoensis*. The Eurasian record includes *Pl. longipes*, *Pl. houfenense*, *Pr. pater*, *Pr. crassum*, *Cr. fissurae* and *Pl. huangheense*, wheres Africa has *Cr. periafricanum*, *Eu. feibeli*, *S. turkanensis*, *Eu. hooijeri* and *Eu. woldegabrieli* (taxa 1–36, Appendix A). *Calippus*, *Dinohippus*, *Cormohipparion*, *Nannippus*, *Neohipparion* and *Astrohippus* have records extending back to 10 Ma and amongst these *Cormohipparion*, *Nannippus*, and *Neohipparion* are hipparionine horses. *Dinohippus* has a chronology beginning at 10.3 Ma and *D. mexicanus* (5.3–4.6 Ma) is demonstrably related to *Equus.* Four species of *Equus*, *E. cumminsi*, *E. enormis*, *E. idahoensis* and *E. simplicidens* first occur in the Blancan (since 4.9 Ma) and in particular, *E. simplicidens* would appear to be related to first occurring Eurasian *Equus* [10,190]. Large mammal Mean HYP in North America ranges from 2.0–2.5, whereas Europe has the lowest mean HYP ranging from 1.5–2, with higher values (2.0–2.5) in Turkey, Greece and Spain. Africa has mostly 2.0 values with slightly lower values in the horn of Africa, whereas Asia shows mostly 2.0 values with localized areas ranging between 2.0 and slightly above 2.5 in China, Mongolia, Kazakhstan and Iran. 

Figure 4 presents the distribution of taxa between 4 and 2.6 Ma. During this time frame, 4 genera are recognized from North America (*Pliohippus*, *Nannippus*, *Equus* and *Plesiohipparion*), 6 from Eurasia (*Plesiohipparion*, *Proboscidipparion*, *Cremohipparion*, “*Hippotherium*”, *Eurygnathohippus*, *Baryhipparion*) and 2 from Africa (*Eurygnathohippus*, *Cremohipparion*). North American taxa carrying over into this interval include *Plio. coalingensis*, *Na. beckensis*, *Equus/Plesippus simplicidens*, *E. cumminsi* and *Equus/Plesippus idahoensis*. The persisting Eurasia taxa include *Pl. longipes*, *Pl. houfenense*, *Pr. pater*, *Pr. crassum*, *Cr. fissurae* and *Pl. huangheense*, whereas most African species disappeared, with the only survival of the hipparion genera *Eurygnathohippus* and *Cremohipparion.* First occurring taxa include *Plesiohipparion* sp. in Ellesmere Island (N. America), *Ba. insperatum*, *Cr. licenti*, “*Hippotherium*” *antelopinum*, *Eurygnathohippus* sp, *Pr. heintzi*, *Pl. rocinantis*, *E. afarensis*, *Eu. hasumense* and *Eu. cornelianus* (Africa). The African clade *Eurygnatohippus* is found there during this interval and is represented by a lower cheek tooth from India at the end of this temporal interval. “*Hippotherium*” *antelopinum* is a medium sized hipparionine whose type locality is in India. North America records the immigration of *Plesiohipparion* into Greenland. North America and South America have mostly Mean HYP between 2.0 and 2.5, with some areas in the west recording values of around 2.5 and others between 1.5–2.0. 

Figure 5 presents the distribution of taxa between 2.58 and 1.5 Ma, including the *Equus* Datum in Eurasia at the beginning of the Pleistocene. During this time frame, 2 genera are recognized from North America (*Nannippus* and *Equus*), 4 from Eurasia (*Plesiohipparion*, *Proboscidipparion*, *Baryhipparion* and *Equus*) and 2 from Africa (*Eurygnathohippus* and *Equus*). North America records *E. calobatus*, *E. scotti*, *E. stenonis anguinus*, *E. conversidens*, *E. ferus/lambei*, *E. francisci*, *E. fraternus* and *E. pseudaltidens* during this interval, in addiction to *Na. beckensis*, *Equus/Plesippus simplicidens*, *E. cumminsi* and *Equus/Plesippus idahoensis*. Europe records several species of *Equus* in this interval, including: *E. livenzovensis*, *E. major*, *E. stenonis*, *E. senezensis*, *E. stehlini* and *E. altidens*. The Indian subcontinent has *E. sivalensis* and *Equus* sp. in India. China records several *Equus* species during this interval including *E. eisenmannae*, *E. sanmeniensis E. huanghoensis*, *E. yunnanensis*, *E. qingyangensis*, *E. teihardi* and *E. wangi*, with *Ba. insperatum*, *Pl. shanxiense* and *Pr. sinense*. Central Asia includes *E. pamirensis*, whereas Africa includes the *Equus* species in North Africa (*E. numidicus*) and *Equus* sp. in East Africa. Overall, the African record includes *E. tabeti* in North Africa with *Eu. pomeli*, *E. koobiforensis*, *E. oldowayensis* in East Africa and *E. capensis* and *E. zebra* in South Africa. The earliest species of *Equus* are found in some localities at ca. 2.6 Ma in Europe, Siwalik Hills and China, which shows values between 2.0–2.5. These values are more diffused in Eurasia compared with Figure 4, although are still present values between 1.5–2.0 in China, Russia, Caucasus and Europe. Africa overall has values between 2.0 and 2.5 through most of the continent, with isolated areas between 2.5–3.0 in North and East Africa. During this interval, most continental mammal records are dry with mean hypsodonty mostly being 2.0 or higher with several higher incidences of 2.5 or higher. 

Figure 6 presents the distribution of taxa between 1.5 Ma to recent. During this time frame, 2 genera are potentially recognized from North America (*Equus* and *Haringtonhippus*), 2 from South America (*Equus* and *Hippidion*), 2 from Eurasia (*Equus* and *Proboscidipparion*) 2 from Africa (*Equus* and *Eurygnathohippus*). This time frame includes a number of taxa that carry over from the 2.58–1.5 Ma interval including *Equus/Plesippus simplicidens*, *E. cumminsi*, *Equus/Plesippus idahoensis*, *E. calobatus*, *E. scotti*, *E. conversidens*, *E. ferus/lambei*, *E. francisci*, *E. fraternus* and *E. pseudaltidens* (North America); *E. sameniensis*, *E. yunnanensis*, *E. qingyangensis*, *E. teilhardi*, *E. wangi* (China); *E. sivalensis* (India); *E. numidicus*, *E. tabeti* (North Africa); *E. koobiforensis*, *E. oldowayensis* (East Africa); *E. capensis*, *E. zebra* (South Africa). Taxa first occurring between 1.5 Ma to recent are *E. verae*, *E. cedralensis*, *E. mexicanus* and *E. occidentalis* for North America; *E. neogeus*, *Hippidion devillei*, *Hippidion saldiasi*, *Hippidion principale* for South America; *E. beijingensis*, *E. dalianensis*, *E. hemionus* (also India), *E. kiang* (also Nepal) and *E. przewalskii* in China; *E. nalaikensis* and *E. colimensis*, *E. lenensis* and *E. ovodovi* in Central and North Asia; *E. suessenbornensis*, *E. apolloniensis*, *E. wuesti*, *E. hipparionoides*, *E. marxi*, *E. ferus*, *E. hydruntinus*, *E. petraolnensis* and *E. graziosi* in Europe; *E. mauritanicus*, *E. melkiensis* and *E. algericus* in North Africa and *E. africanus*, *E. grevyi* and *E. quagga* in East and South Africa. This time frame record also records the last occurrence of the hipparionini horses in Asia with *Proboscidipparion* and Africa with *Eurygnathohippus* [8]. Mean HYP shows North and South America having more moderate climates around values of 2.0 and 2.5, with values lower than 2.0 in local areas in the East and the West. Europe likewise has Mean HYP values around 2.0 with values lower than 2.0 in Central and Eastern Europe, whereas Asia, the Indian Subcontinent and Africa have higher values hovering around Mean HYP of 2.5. Mean HYP values between 1.5–2.0 are also found in China.

The mean hypsodonty map patterns indicate that while most of Eurasia and Africa occupied by mild or humid values, arid environmental conditions were prominent in North America between 7 and 4 Ma (Figure 3). At the late Miocene–Early Pliocene transition, moist conditions occurred in Europe with an arid belt extending eastward into Asia and Africa. By the end of the Pliocene arid conditions remained in North America and aridity began to increase on the mid-latitudes of Eurasia and along the Rift Valley of East Africa and the western corners of North Africa (Figure 4). Mean ordinated crown height patterns of the large herbivorous mammal communities indicate that while Southeast Asia, Central and Western Europe, and Florida and California in North America occupied by semi-humid or humid values, arid environmental conditions persisted or increased drastically rest of the World and in particular in East and South Africa, 1.5 Ma to recent (Figure 6). 

Overall, these maps exhibit a general trend of increased drying over time. Most occurrences of *Equus* are with Mean HYP values of 2.0–2.5. Very few archaic Equinae taxa continue across the Pliocene–Quaternary boundary. North America retained the more primitive lineages of *Pliohippus* and *Nannippus* up into the 2.58–1.5 Ma interval. Hipparionines persisted up into the Pleistocene of Europe, as late as 1.0 Ma in China and slightly later than 1.0 in Africa. The extinction of older North American and Eurasian-African lineages would appear to be associated with the expansion of more open country dry conditions.

## 8. Phylogeny

As reported in the introduction, recently two morphological-based cladistic analyses have re-evaluated the origin of the genus *Equus*. Herein, we present the state of art of these cladistic hypotheses, with a separate section on the contribution from the molecular phylogenies. 

Barrón-Ortiz et al. [9] undertook a phylogenetic analysis using a matrix of 32 characters (22 cranial, 6 mandibular, 3 autopodial, and estimated body size). The authors included in the matrix 21 Equini species, two of which were considered as outgroups, *Acritohippus stylodontus*, and *Pliohippus pernix*. Barrón-Ortiz et al. [9] undertook the analysis using TNT 1.1 [396] with the implicit enumeration option (exhaustive search), using equal weighting for the characters, and without a collapsing rule. They treated all characters as unordered. 

Cirilli et al. [10] have combined a new matrix including 30 Operational Taxonomic Units (OTUs) and 129 characters (72 cranial, 40 mandibular and 17 on autopodia), 68 of which were new and the other extrapolated from the recent published matrices on perissodactyl phylogeny [9,114,397,398,399,400]. The characters were mainly coded by direct observations. The ingroup included a comprehensive sample of 26 equid species and the outgroup was represented by the Brazilian tapir *Tapirus terrestris*, the rhinocerotoid *Hyrachyus eximius*, the Rhinocerotidae *Trigonias osborni* and the early-diverging equid *Merychippus insignis*. The analysis was performed in PAUP 4.0b10, with Heuristic search, TBR and 1000 replications with additional random sequence, and gaps treated as missing. In this analysis, 24 characters have been ordered and 105 characters unordered. All characters were equally weighted. 

### 8.1. What Is Equus? Paradigms, Phylogeny, and Taxonomy

The primary objectives of the study conducted by Barrón-Ortiz et al. [9] were to review and discuss different paradigms for understanding generic-level taxonomy, particularly in regards to the delimitation of mammalian genera, and to evaluate how those different paradigms impact the concept and contents of *Equus* in a given phylogenetic tree. Barrón-Ortiz et al. [9] established a new phylogenetic tree of derived Equini for that analysis. The tree served as a model for the evaluation of how distinct paradigms impact our placement of generic names on any given tree.

#### 8.1.1. What Is a Genus?

Although several studies have discussed limitations of and provided alternatives to the Linnaean taxonomic system [401,402,403,404,405,406], Linnaean taxonomy continues to be widely used to study and communicate about past and present biodiversity [407]. This is especially true when it comes to the binomial nomen (genus and species). Because of the widespread use of binomial nomenclature within and outside of the life sciences, the genus is perhaps the most important higher-level taxonomic rank. Therefore, the question about how we define and delimit genera is not a trivial one as it affects how we view, study, and communicate about biological organisms. *Equus* is a model taxon for such discussions because of its complex generic and species-level history. At the core of the delimitation of *Equus* within any phylogenetic tree lie the philosophical and practical issues regarding the definition of genera and how best to reconcile taxonomy with evolutionary history.

Different paradigms exist for understanding and delimiting genera. In the case of mammals, Barrón-Ortiz et al. [9] identified four that are commonly used in combination with monophyly to delimit genera: phylogenetic gaps, uniqueness of adaptive zone, crown group definition, and divergence time [407,408,409,410,411]. One of the primary distinctions between the paradigms is the way genera are conceived. At one extreme, the uniqueness of adaptive zone paradigm conceives genera as having some level of biological reality beyond monophyly (i.e., a genus occupies a unique adaptive zone). An adaptive zone corresponds to a particular mode of life or a unique ecological situation [6,409,412,413]. At the other extreme, under the divergence time and crown group paradigms, genera are arbitrarily defined [168,410,411,414] and are not conceived as having biological reality other than monophyly. The phylogenetic gaps paradigm occupies an intermediate position. Under this paradigm, genera are not necessarily conceived as having some level of biological reality, but the gaps between monophyletic groups of species used to delimit genera arise from biological processes such as speciation, extinction, evolutionary and adaptive radiations, and unequal rates of evolution [409]. Because genera may be conceived under different paradigms, it is important for researchers to explicitly state their operational paradigm when considering questions of generic-level taxonomy.

#### 8.1.2. Morphological Phylogenetic Analysis of Derived Equini

For the second study objective, Barrón-Ortiz et al. [9] conducted a morphological phylogenetic analysis of derived Equini. The phylogenetic analysis produced three equally most parsimonious trees of 85 steps with consistency (CI) and retention (RI) indices of 0.57 and 0.80, respectively [9] Figure 1 for the strict consensus tree. 

Using the strict consensus tree obtained in their analysis, Barrón-Ortiz et al. [9] evaluated how the four paradigms commonly used to delimit mammalian genera impacted the concept and contents of *Equus*, although we emphasize that the same could be applied to any phylogenetic hypothesis. The results of this evaluation and taxonomic implications are summarized below.

#### 8.1.3. Phylogentic Gaps and the Concept of *Equus*

Under the phylogenetic gaps paradigm, a genus is comprised of a single species or a monophyletic group of species, separated from other single species or monophyletic groups of species of the same rank by a decided gap [409]. In the context of a phylogenetic analysis, the gaps between single species or monophyletic groups of species can be measured by the number of synapomorphic traits. Application of this paradigm to the strict consensus tree of Barrón-Ortiz et al. [9], suggested that *Equus* should be delimited to clade 6, as this clade shows the most synapomorphic traits, including the species *E. neogeus*, *E. occidentalis*, *E. ferus*, *E. mexicanus*, *E. hemionus*, *E. quagga*, *E. conversidens* and *E*. *francisci*. The Early Pleistocene *E. stenonis* and the North American *E. simplicidens* and *E. idahoensis* are not included in this clade. 

#### 8.1.4. Uniqueness of Adaptive Zone and the Concept of *Equus*

In the uniqueness of adaptive zone paradigm, a genus is comprised of a single species or a monophyletic group of species that occupies a different adaptive zone (i.e., a unique mode of life) from the one occupied by species of another genus [409,413]. Application of this criterion in the context of a phylogenetic analysis requires: (1) the identification of traits (i.e., character states) that allow or potentially allow a single species or a monophyletic group of species to occupy a unique adaptive zone and (2) identifying where those traits occur in the tree.

The unique mode of life of *Equus* could potentially be defined as “ungulate mammals that are adapted to live in generally open, arid habitats and that can thrive on low-quality, high-fiber foods such as grasses and other coarse and tough vegetation” [9,19]. Potential morphological adaptations for this mode of life include modifications to the locomotory [415] and digestive systems, particularly the dentition [9]. Based on the position of the majority of purported, adaptive zone-related traits, *Equus* is assigned to clade 6, or possibly clade 7, in the strict consensus tree of Barrón-Ortiz et al. [9], under the uniqueness of adaptive zone paradigm. The identification to the clade 7 would include *E. stenonis* in the genus *Equus*, but not *E. simplicidens* and *E. idahoensis*.

#### 8.1.5. Crown Group and the Concept of *Equus*

This paradigm follows a nominalist perspective to the definition of taxa. The nominalist perspective assumes that the limits of named taxa are arbitrary conventions, and then proceeds to spell out those conventions [414]. Under the crown group paradigm, a genus is defined as the clade that includes the most recent common ancestor of all extant species assigned to that genus, and all descendants of that ancestor. Therefore, under this paradigm, *Equus* is defined as the clade that includes the most recent common ancestor of all extant species assigned to *Equus*, and all descendants of that ancestor. *Equus* is constrained to clade 6 in the strict consensus tree of Barrón-Ortiz et al. [9] based on the crown group paradigm, which include the same species obtained under the phylogenetic gaps paradigm.

#### 8.1.6. Divergence Time and the Concept of *Equus*

The divergence time paradigm states that a species or a monophyletic group of species should be regarded as a distinct genus if it diverged well-before the Miocene-Pliocene boundary (4–7 Ma) [168,410,411]. Application of this paradigm in the context of a phylogenetic analysis requires the creation of a time-calibrated phylogeny. Based on the time-calibrated phylogeny of Equini of Barrón-Ortiz et al. [9], *Equus* is delimited to clade 9 under the divergence time paradigm. Here, *E. stenonis* and the North American *E. simplicidens* and *E. idahoensis* should be included in the *Equus* clade. 

#### 8.1.7. Taxonomic Implications

Barrón-Ortiz et al. [9] concluded that *Equus* should be delimited to clade 6 in their phylogenetic analysis, based on the fact that three out of the four paradigms used to define mammalian genera identified clade 6 as the most suitable position for *Equus*. This taxonomic arrangement excludes *E. stenonis*, *E. idahoensis*, *E. simplicidens*, and “*Dinohippus*” *mexicanus* from the genus *Equus* and it implies that *Haringtonhippus* is a junior synonym of *Equus*. Some researchers have assigned *E. simplicidens* and *E. idahoensis* to *Plesippus* at the generic or subgeneric rank [4,57,59,416,417], with *Plesippus simplicidens* selected as the type species [4,416]. Likewise, *E. stenonis* has been referred to *Allohippus* at the generic or subgeneric rank [416]. Based on the results of their analysis, Barrón-Ortiz et al. [9] suggested that *Plesippus* and *Allohippus* should be elevated to generic rank, “*Dinohippus*” *mexicanus* should be assigned to a new genus, and *Haringtonhippus* should be synonymized with *Equus*.

### 8.2. Cirilli et al. [10] Phylogeny: Equus Modeled as a Sigle Monophyletic Clade

The results from Cirilli et al. [10] differ from the previous phylogeny of Barròn-Ortiz et al. [9]. Cirilli et al. [10] obtained a single most parsimonious tree from the matrix used, and the genus *Equus* is modeled as a single clade with node 52 being supported by 18 unambiguous synapomorphies, and 13 of these have a CI ≥ 0.500 [10], (Figure 2), not allowing the endorsement of *Plesippus* or *Allohippus* at generic or subgeneic level. In particular, the genus is defined by a linear lateral outline of the skull, the absence or reduction of the buccinator fossa, the presence of a shallow depression on the lingual margin of the protocone, the squared shape of the protocone on P2, the presence of an elongated pli caballine on P3 and P4, the squared shape of the protocone on P3 and P4, a V-shaped morphology of the linguaflexid, part of the metaconid-metastylid complex, a squared morphology of the lingual side of the metastylid, a strong and broad 3rd phalanx, a reduced lateral second and fourth metapodials. Moreover, additional analyses as the bootstrap tree supports the *Equus* clade with 99/100 replications [10], (Appendix A). According to Brochu and Sumrall [418], clades within a cladogram are named if two criteria are met: the clade is stable and unlikely to collapse, and there is a need to discuss the group. In addition, Bryant [419], Cantino et al. [420], and Schulte et al. [421] provide some guidelines for the establishment of clade names, including the application of methods for measuring nodal support, careful consideration of those taxa that are likely to move around in different analyses, and use of multiple basal taxa as specifiers for node-based groups. A recently proposed phylogenetic nomenclatural system [422,423,424] specified that all supraspecific taxonomic nomina be explicitly defined on the basis of common ancestry. In the work by Cirilli et al. [10], *E. simplicidens* is considered as the common ancestor of all the *Equus* species, and place at the base of their radiation, separated from the genus *Dinohippus*. This view is supported by recent molecular analyses of the group, where all the extant equid taxa are grouped into a single genus, *Equus* [27,28,29,425]. A large *Equus* clade, including some fossil taxa, is also identified by Heintzman et al. [29], where a new clade composed by representatives of *Haringtonhippus* is supposed to diverge from *Equus* during the early Pliocene. It would appear that *Haringtonhippus* is convergent in cranial and postcranial features with Asian *E. hemionus* and perhaps Pleistocene *E. altidens*.

#### 8.2.1. Phylogenetic Gaps, Crown Group, Adaptive Zone, and Divergence Time Applied to the Phylogeny of Cirilli et al. [10] 

The phylogenetic gaps criterion identify a genus as a taxonomic category containing a single species, or a monophyletic group of species, which is separated from other taxa of the same rank by a relevant gap. The results from Cirilli et al. [10] support the definition of *Equus* as being a single monophyletic clade since *E. simplicidens*, grouped separately from the species included in the genus *Dinohippus*. 

As reported above, the concept of the adaptive zone implies that ecological factors contribute to the speciation process. In this regard, for the genus *Equus* may be taken in consideration the progressive shift to a diet mostly based on C4 grasses. The palaeoecological studies based on the North American record provide some insights between the last representative of the genus *Dinohippus, D. mexicanus*, and the first forms of *Equus*. As reported by MacFadden et al. [426] and Semprebon et al. [427], fossil and extant species of *Equus* have been almost grazers or mixed feeders (except for some large species) [19], whereas some populations of the late Hemphillian *D. mexicanus* from Florida show a browsing signal. However, other late Hemphillian *Dinohippus* samples were identified as mostly grazer, suggesting that the dietary transition from browser to mixed feeders and grazer may already have occurred in North America. Nevertheless, the presence of some individual with δ13C values of 24.7 and 21.5 per mil in the *D. mexicanus* sample studied by MacFadden et al. [426] suggests that some individuals of this sample were feeding on C4 grasses. This evidence indicates that some populations of *Dinohippus* shifted to a more grazing diet, which may have led to speciation process to new forms adapted to new environments. This would have affected not only the diet, but also the increase of the body mass, from ca. 300 kg in *D. mexicanus* to 300 and 400 kg in *E. simplicidens* [10,426]. Moreover, MacFadden et al. [426] reported that the dietary shift from browser to widespread grazing in *Equus* may have occurred during the early Pliocene in North America between, 4.8 and 4.5 Ma, a time frame coherent with the first occurrences of *E. simplicidens*.

The crown group as defined as being a collection of species composed of the living representatives of the collection, the most recent common ancestor of the collection, and all descendants of the most recent common ancestor. In this regard, the results from Cirilli et al. [10] include zebras, asses and the caballine horses in a single clade with the most recent common ancestor identified as being *E. simplicidens*. 

Moreover, the concept of the most recent common ancestor is directly linked also to divergence time, and to the estimations based on the genomic analyses from Orlando et al. [27], which estimated a time frame of 4.5–4.0 Ma for the origin of the most recent common ancestor for *Equus.* This estimated ages confirm some one of the oldest discoveries of *E. simplicidens* in North America [428,429,430] and therefore supporting the hypothesis of *E. simplicidens* as the first representative of the genus *Equus*. 

To summarize, phylogenetic gaps, crown group, adaptive zone and divergence time are congruent for identify of the *Equus* clade at node 52 of the phylogeny from Cirilli et al. [9], in agreement with the MPT, Bootstrap and UPGMA tree. 

#### 8.2.2. Living and Fossil Equids

The results by Cirilli et al. [10] support the taxonomic division of caballines (domestic horse and Przewalski’s wild horse) and noncaballines (zebras and African and Asiatic asses) proposed by morphological data [62] and other molecular and combined studies [431,432,433,434,435] ([8] UPGMA analyses, supplemental information). Similarly to previous studies, Cirilli et al. [10] reported a paraphyletic origin of the extant zebra species as proposed in the literature using cranial morphology [397], palaeogenetics [27,436,437], and nuclear data [438,439,440], but with a low levels of support of the nodes. Other molecular analyses instead suggested a monophyly of the zebra species with the mountain zebra placed as the sister taxon of Burchell’s and Grevy’s zebras [432,440,441,442]; a result affected, anyway, by the absence of fossil representatives of this group in the analyses.

### 8.3. The Contribution from the Molecular Phylogeny

In the last two decades, new perspectives on the evolution of the genus *Equus* have been reported with the contribution from the molecular phylogeny. Orlando et al. [27] coded the genome of a fossil horse dated ca. 780–560 ka, identifying that the most recent common ancestor for the genus *Equus* emerged at ca. 4.5–4.0 Ma in North America, which is now in agreement with oldest findings and occurrences of the North American *E. simplicidens*. Analogous results were obtained also by Vilstrup et al. [425], which however highlighted the distinction of the North American stilt legged horses from the living asses, supporting a different evolution which led to a similar morphology. Moreover, Vilstrup et al. [425] identified zebras and asses as distinct clades, proposing an estimated age for origin of the plain zebras at ca. 0.7 ± 0.1 Ma, and divergence from the Grevy’s zebas at ca. 1.5 Ma. 

More inputs came from Jónsson et al. [28], which identified all the living equids belonging to a single genus, *Equus*. Moreover, Jónsson et al. [28] estimated that living zebras and asses cluster into a single monophyletic clade originating at ca. 2 Ma, that the African and Asiatic asses diverged slightly later, at ca. 1.8 Ma, and that the living zebras already diverged at ca. 1 Ma. These estimated ages are in agreement with the first occurrence of fossil species related to living zebras (*E. koobiforensis*, *E. mauritanicus*) or asses (*E. altidens*, *E. tabeti*). *Equus hemionus* and *E. kiang* diverged later, between 356–233 ka. Jónsson et al. [28] estimated also that the gene flow between caballine and stenonine horses ceased between 3.4 and 2.1 Ma, which is in agreement with the dispersal of the stenonine horses in Eurasia at the beginning of the Pleistocene.

Heintzmann et al. [29] used the crown group definition for the genus *Equus* and focused their study on the North American species, especially on the stilt legged species, which were previously identified close to Asian asses [56,62,69]. Nevertheless, the genetic analyses of Orlando et al. [153] and Vilstrup et al. [425] separated these species from the Asian asses and placed them close to the caballine horses. The new phylogeny from Heintzmann et al. [29] has identified the North American stilt horses as a distinct branch from the living and fossil *Equus*, diverging between 5.7–4.1 Ma, during the late Hemphillian or early Blancan. This separation anticipates the origin of the most recent common ancestor identified by Orlando et al. [27]. Heintzmann et al. [29] proposed a new genus, *Harringtonhippus*, for the stilt legged horses from North America, represented by the species *Ha. Francisci*. However, this taxonomy is not accepted by other authors on philosophical grounds [9].

Vershinina et al. [78] identified two dispersal event for the caballine horses. The first occurred between 0.95–0.45 Ma, in east to west direction, consistent with the oldest findings of the caballine horses in Eurasia. The second occurred at 0.2–0.05 Ma, bidirectional but predominantly west to east, due the identification of metapopulations of Eurasian Late Pleistocene horses in Alaska and Northern Yukon, which provided the opportunity for a gene flow between the North American and Eurasia horses during the Late Pleistocene.

Another interesting perspective comes from the subgenus *Sussemionus*. This subgenus was proposed by Eisenmann [256] as an informal group of species from the Early and Middle Pleistocene of Eurasia. Later, Eisenmann [87] formalized the subgenus, characterized by a combination of some dental features [87]. Eisenmann [87] included in this subgenus the species *E. coliemensis*, *E. suessenbornensis*, *E. verae*, *E. granatensis*, *E. hipparionoides* and *E. altidens*. Following the description of the author, the genus includes some anatomical features observed in the Süssenborn sample and in the modern Asian asses. Nevertheless, recent molecular studies [29,151,153,427,443] identified this subgenus separated from the living species even included in the genus *Equus*, surviving until the late Holocene with the species *E. ovodovi*, a Late Pleistocene species from Siberia. However, it should be noted that this subgenus has never been tested with a morphological based cladistic analysis, which is needed to address its taxonomic status. 

## 9. Conclusions

Nineteen collaborating international scientists provide herein a detailed review and synthesis of fossil Equinae occurrences from the Plio-Plesitocene and recent of North, Central and South America, Eurasia and Africa including fossil and living species. At the present time, our review has identified valid 114 (+4) species of Equinae from 5.3 Ma to recent including 38 from North America, 4 from South America, 26 from East Asia, 6 from the Indian Subcontinent, 18 from Europe and 26 from Africa. In all continents other than South America, more primitive equine clades persisted after *Equus* appeared, and extinction of these more archaic clades were diachronous at the continental to inter-continental scale. While actively researched over the last several decades, Equinae taxonomy is not wholly settled and there are challenges to unifying them at the genus and higher taxonomic levels. That being said, the taxonomy of Equinae reviewed herein has allowed us to provide well resolved biochronology and biogeography, paleoecology and paleoclimatic context of the 5.3 Ma–recent Equinae records.

The paleoclimatic maps from 7 Ma to recent have shown a more suitable environment for the evolution of the modern Equini in North America rather Eurasia and Africa during the Pliocene, with more arid conditions which favored speciation of *Equus* and its dispersal into Eurasia and Africa at the beginning of the Pleistocene. This result is congruent with the hypothesis of several previous morphological, paleoecological and molecular studies cited herein which support the origin of *Equus* during the Pliocene in North America. 

Finally, we presented the most recent cladistic morphological based hypotheses on the origin of the genus *Equus*, combined with the results from the molecular phylogenies. Phylogenetic evidence suggests that the genus *Equus* is closely related with *Dinohippus*, from which evolved. The North American *E. simplicidens* represents the ancestral species for the origin of the stenonine horses in Eurasia and Africa, culminated with the evolution of modern zebras and asses. This last point is supported also by the molecular analyses, which have hypothesized that North American stilt legged horses diverged from the living asses and are not phylogenetically linked. However, more studies are needed to shed light on the evolution of the caballine horses, which at the present time remains unresolved. Lastly, we acknowledge the different interpretations of morphological and molecular based cladistic analyses and the need to better integrate these studies going forward.

## Figures and Tables

**Figure 1 biology-11-01258-f001:**
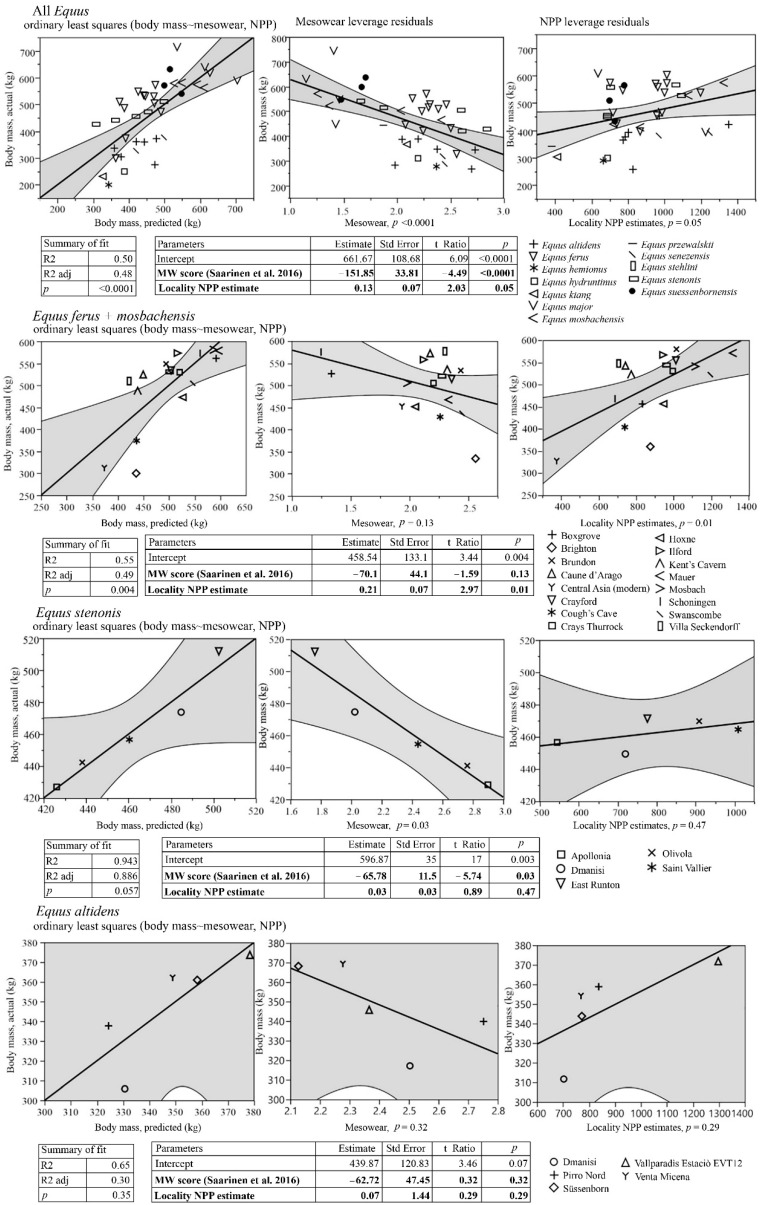
Ordinary least squares models of the effect of the mesowear score and the estimated NPP of the localities on the body mass estimates of the genus *Equus* from the Pleistocene of Eurasia.

**Figure 2 biology-11-01258-f002:**
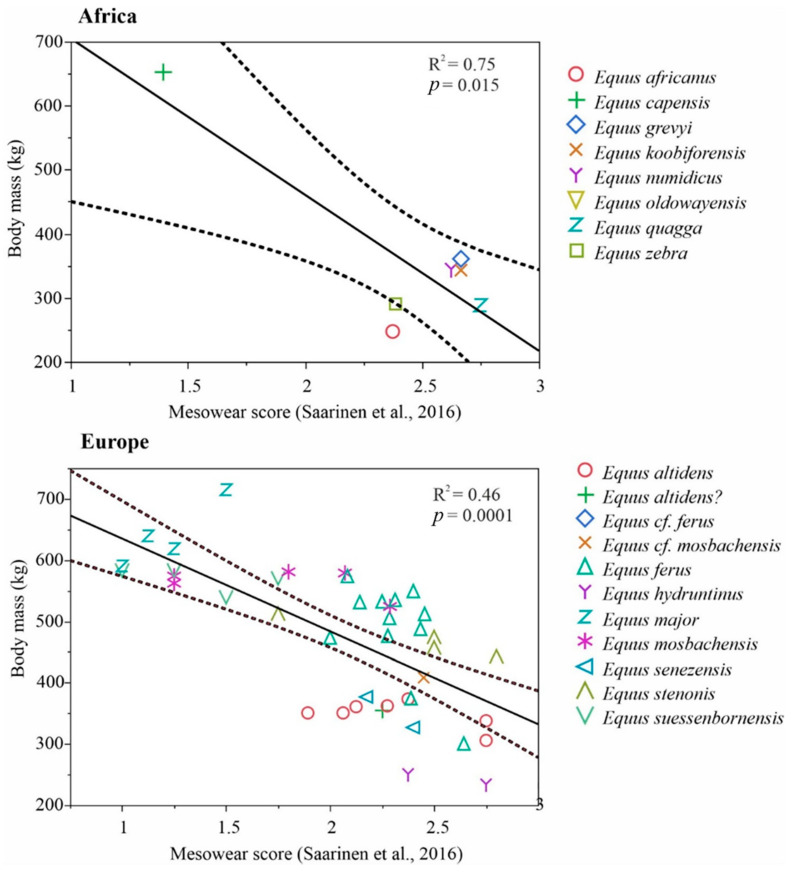
Linear regressions between body mass and the mesowear of *Equus* in Africa and Europe during the Quaternary. On both continents, the body mass of *Equus* was significantly negatively related to the mesowear score, although in Africa, this pattern was entirely driven by one very large-sized species, *E. capensis*. Adapted from [20].

**Figure 3 biology-11-01258-f003:**
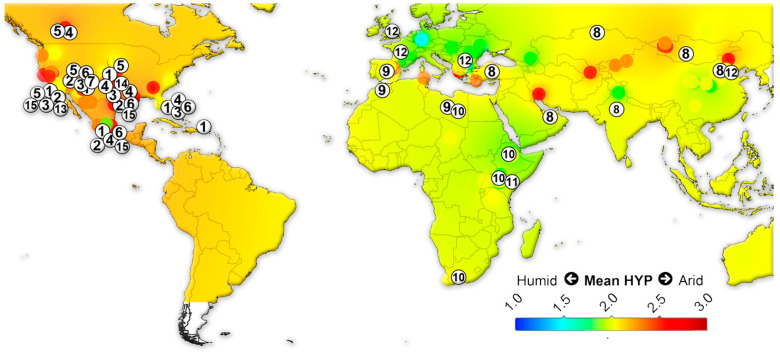
Spatial distribution of the large herbivorous genera mean ordinated crown height through time ranges 7 to 4 Ma in North, Central and South America, Eurasia and Africa. The mean ordinated hypsodonty map represents the paleoclimatological conditions grading from most humid (blue) to most arid (red). Numbers in white circles show coded number of each taxa given in Appendix A. The mean ordinated hypsodonty values are represented by the color-coded circles indicate the spatial position of the localities that mean hypsodonty scores calculated (Appendix A). IDW interpolation algorithm hypothetically interpolates no data (no locality) area based on the actual data. These areas should be ignored.

**Figure 4 biology-11-01258-f004:**
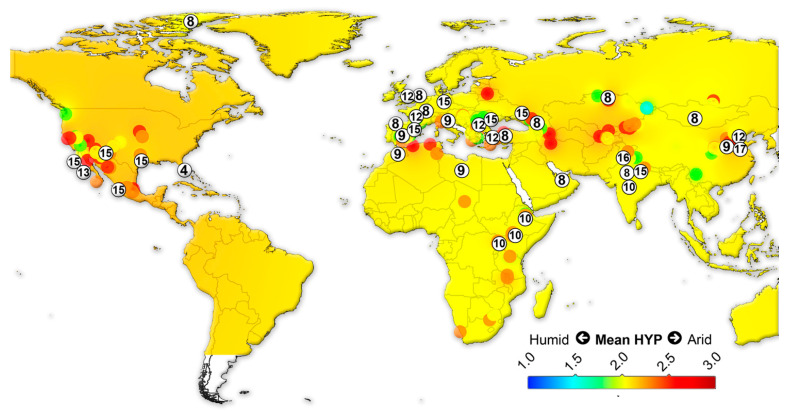
Spatial distribution of the large herbivorous genera mean ordinated crown height through time ranges 4 to 2.6 Ma in North, Central and South America, Eurasia and Africa. The mean ordinated hypsodonty map represents the paleoclimatological conditions grading from most humid (blue) to most arid (red). Numbers in white circles show coded number of each taxa given in Appendix A. The mean ordinated hypsodonty values are represented by the color-coded circles indicate the spatial position of the localities that mean hypsodonty scores calculated (Appendix A). IDW interpolation algorithm hypothetically interpolate no data (no locality) area based on the actual data. These areas should be ignored.

**Figure 5 biology-11-01258-f005:**
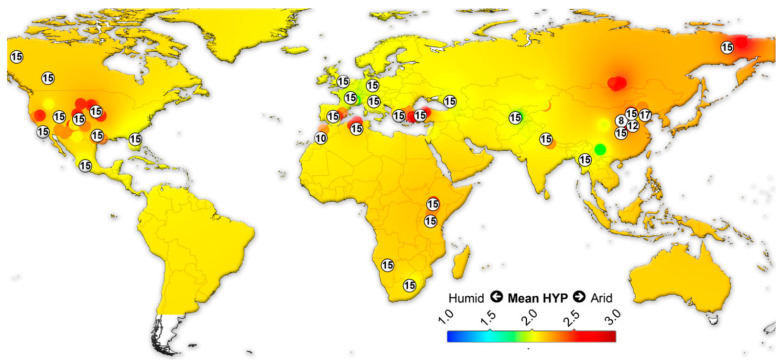
Spatial distribution of the large herbivorous genera mean ordinated crown height through time ranges 2.58 to 1.5 Ma in North, Central and South America, Eurasia and Africa. The mean ordinated hypsodonty map represents the paleoclimatological conditions grading from most humid (blue) to most arid (red). Numbers in white circles show coded number of each taxa given in Appendix A. The mean ordinated hypsodonty values are represented by the color-coded circles indicate the spatial position of the localities that mean hypsodonty scores calculated (Appendix A). IDW interpolation algorithm hypothetically interpolate no data (no locality) area based on the actual data. These areas should be ignored.

**Figure 6 biology-11-01258-f006:**
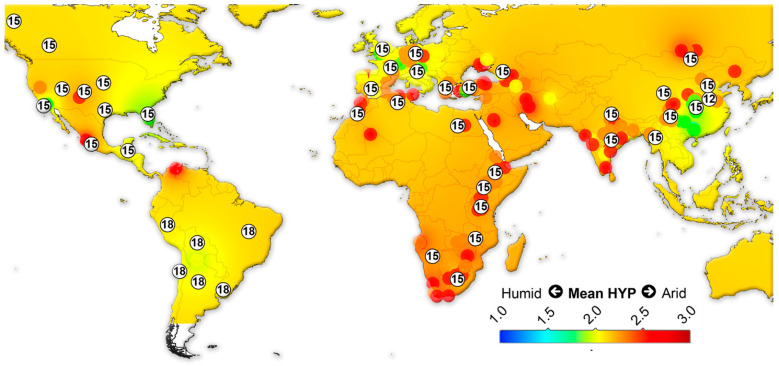
Spatial distribution of the large herbivorous genera mean ordinated crown height through time ranges 1.5 Ma to recent in North, Central and South America, Eurasia and Africa. The mean ordinated hypsodonty map represents the paleoclimatological conditions grading from most humid (blue) to most arid (red). Numbers in white circles show the coded numbers of each taxon given in Appendix A. The mean ordinated hypsodonty values are represented by the color-coded circles indicate the spatial position of the localities that mean hypsodonty scores calculated (Appendix A). IDW interpolation algorithm hypothetically interpolate no data (no locality) area based on the actual data. These areas should be ignored.

## Data Availability

All data generated by this study are available in this manuscript and the accompanying Appendix A.

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
