# Peer review of "Evolution of the Family Equidae, Subfamily Equinae, in North, Central and South America, Eurasia and Africa during the Plio-Pleistocene"

_biology, 2022, doi:10.3390/biology11091258_

Round 1

Reviewer 1 Report

This is a review work on the world fossil record of horses of great relevance to the scientific community. I have made a few small comments that I understand can help improve the quality of the manuscript. In particular there is an incorrect use of Geochronological/Chronostratigraphic units throughout the text. Particularly when referring to the Stage/Age, there is a mixture of adjectives such as "early", "middle" or "late" that in many cases do not constitute the formal name to which they refer. For example: "Early Pliocene" should be "early Pliocene" , with "early" in lowercase, since it is not the name of the earliest Pliocene but an adjective. In the case they want to use the formal Stage/Age should be in the same example "Zanclean", which corresponds to the name of the early Pliocene. An exception here could be "Upper Pleistocene" (Chronostratigraphic) or "Late Pleistocene" (Geochronologic) which are formal unit as the "Late Cretaceous".  I have marked in color throughout the text, most of these inconsistencies. 

Also I've marked  the use of "cf" which must be followed by a specific category in these case. The authors only name the specific epithet, which does not constitute a category. For example it should be: "genus" cf. "genus + specific epithet" not "genus" cf. "specific epithet".

None of these comments and suggestions opaque the great work of synthesis here done by the authors. 

Author Response

This is a review work on the world fossil record of horses of great relevance to the scientific community. I have made a few small comments that I understand can help improve the quality of the manuscript. In particular there is an incorrect use of Geochronological/Chronostratigraphic units throughout the text. Particularly when referring to the Stage/Age, there is a mixture of adjectives such as "early", "middle" or "late" that in many cases do not constitute the formal name to which they refer. For example: "Early Pliocene" should be "early Pliocene" , with "early" in lowercase, since it is not the name of the earliest Pliocene but an adjective. In the case they want to use the formal Stage/Age should be in the same example "Zanclean", which corresponds to the name of the early Pliocene. An exception here could be "Upper Pleistocene" (Chronostratigraphic) or "Late Pleistocene" (Geochronologic) which are formal unit as the "Late Cretaceous".  I have marked in color throughout the text, most of these inconsistencies. 

Reply: We thank the reviewer for this comment. We have followed this suggestion editing the substages of the Miocene, Pliocene and Pleistocene which are not defined in the latest Geologic Time Scale 2020 (Chapter 29 “The Neogene Period” – Chapter 30 “ The Quaternary Period”). Following the indications of the Raffi et al. 2020, the substages of the Miocene and of the Pliocene should not be capitalized (e.g. middle Miocene, late Miocene, early Pliocene, late Pliocene). On the other hand, Early, Middle and Late Pleistocene should be capitalized, considering their definition as formal Geochronologic units of the Pleistocene (Gibbard and Head, 2020).  

References:

  1. Raffi, B.S. Wade, H. Pälike, A.G. Beu, R. Cooper, M.P. Crundwell, W. Krijgsman, T. Moore, I. Raine, R. Sardella, Y.V. Vernyhorova. Chapter 29 - The Neogene Period. Editor(s): Felix M. Gradstein, James G. Ogg, Mark D. Schmitz, Gabi M. Ogg, Geologic Time Scale 2020, Elsevier, 2020, Pages 1141-1215, ISBN 9780128243602, https://doi.org/10.1016/B978-0-12-824360-2.00029-2

P.L. Gibbard, M.J. Head. Chapter 30 - The Quaternary Period. Editor(s): Felix M. Gradstein, James G. Ogg, Mark D. Schmitz, Gabi M. Ogg, Geologic Time Scale 2020, Elsevier, 2020, Pages 1217-1255, ISBN 9780128243602, https://doi.org/10.1016/B978-0-12-824360-2.00030-9

Also I've marked  the use of "cf" which must be followed by a specific category in these case. The authors only name the specific epithet, which does not constitute a category. For example it should be: "genus" cf. "genus + specific epithet" not "genus" cf. "specific epithet".

Reply: We thank the reviewer for this comment. This issue has been resolved as suggested.

 None of these comments and suggestions opaque the great work of synthesis here done by the authors. 

Reviewer 2 Report

This paper summarizes the evolution, biochronology, and biogeography of Equus sp. It is very for those interested in Equus evolution as it presents the most updated taxonomic revision of the genus. However, when reading the paper, I found it difficult to follow the geographic organization. I have been going back and forth in my mind if a chronological organization might be better or not. I suggest the authors explore this option.

I found the supplementary material and figures more readable than the long taxonomic list and suggest that these might be reversed or combined, i.e., sections 3 and 4 will include sections 3.7 – 4.4.26 from the supplementary text (I would note that the numbering is confusing and that there are numbers that are dropped like section 5 in the supplemental material).

The difficulty of the geographic organization is particularly evident in less robust descriptions of transitional Ares like the Southern Levant, which has both African taxa and European and Eurasian ones. I do not want to go through the complete list of species. Still, I wanted to point out several omissions of some species, which suggest that some identification of the geographic and chronological range of species needs to be amended. 

1.     Equus tabeti is found also in Bitzat Ruhama (1.0 Ma, Yeshurun et al 2011) and Qafzeh (120,000; Tchernov 2002). The 'Ubeidiya specimens were identified as Equus altidens in Gaudzinski 2004; indeed, multivariate analysis has suggested similarities between the equids from the E. numerousE  tabeti lineage and Equus cf. altidens (Guerreo-Alba and Palmqvist, 1997).  .  This lineage has been named "simplicidens" and includes E. numidicusE. tabeti, E. altidens, and Equus granatensis. This lineage has also been found in Spain (Venta Micena, Orce, Cúllar de Baza, Cueva Victoria, Huèscar-1), Italy (Pirro Nord, Selvella), France (Sainzelles), and Germany (Süssenborn) (Arribas and Palmqvist, 1999).  There has been a discussion with some suggesting an African origin for the species (Gibert et al., 2016), while others offer an Asian origin (Madurell-Malapeira et al., 2014).

2.     Equus melkiensis also is found in sites of Gesher Benot Ya'akov (Eisenmann 2006, 2012b), Nahal Hesi (Yeshurun et al. 2011), Oumm Qatafa (Middle Plesitocene, Marom, et al., 2022) as well as Yemen and Tajikistan (Eisenmann 2006)

3.     Equus mauritanicus is also found in Oumm Qatafa (Marom et al., 2022)

4.     E. valeriani is not discussed (Eisenmann, 2002)

The second issue I find is the analysis of paleoecology. While the study focused on using the NOW database to retrieve data, additional data could have been added to the research that is not available in NOW (e.g., Equus tabeti in 'Ubeidiya, Belmaker and O'Brien 2018). However, a more critical analysis of mesowear has been shown to have a phylogenetic signal, and all linear correlations should be analyzed using phylogenetic contrasts (Belmaker and O'Brien, 2018). This is particularly evident in figure 2, where the regression is pulled by one outlier and should not be viewed as linear regression. 

Author Response

This paper summarizes the evolution, biochronology, and biogeography of Equus sp. It is very for those interested in Equus evolution as it presents the most updated taxonomic revision of the genus.

Reply: We would like to acknowledge the reviewer to provide us his comments. We have addressed all the comments made by the reviewers to improve the quality of the present manuscript.

However, when reading the paper, I found it difficult to follow the geographic organization. I have been going back and forth in my mind if a chronological organization might be better or not. I suggest the authors explore this option.

Reply: We thank the reviewer for this comment. We have stated in the introduction (point 1) the organization of the taxa discussed in the manuscript. The list of taxa are divided by region as follows: North and Central America, South America, Eastern and Central Asia (China, Mongolia, Russia, Uzbekistan, Kazakhistan, Tajikistan), Indian Subcontinent, Europe, and Africa, with taxa organized chronologically from the oldest to the youngest. This description was already present in the original submitted manuscript, however we have strengthened this qualification.

I found the supplementary material and figures more readable than the long taxonomic list and suggest that these might be reversed or combined, i.e., sections 3 and 4 will include sections 3.7 – 4.4.26 from the supplementary text (I would note that the numbering is confusing and that there are numbers that are dropped like section 5 in the supplemental material).

Reply: We thank the reviewer for this comment. Nevertheless, we explained in the Materials and Methods (lines 138-142) the structure of the manuscript, which includes authorship, chronologic and paleobiogeographic ranges, and some historical and evolutionary considerations on each taxon in the main text, and a reduced and emended diagnosis of the species in Supplementary text. At the beginning of this multinational-multiauthored (19) undertaking, we considered the option of including all the information in the main text, but this resulted in a manuscript that was too unwieldy and long and did not concentrate the information central to the taxonomy. We realized from the beginning that the additional information eventually included is important and placed that in Supplementary text. We recognize the interest of the reviewers comment, but we believe that the present structure of the manuscript helps the readers to identify the information they are looking for such as authorship, chronologic and paleobiogeographic ranges, evolutionary considerations and emended diagnosis.  This essential information is necessary to make the most sense of the biogeography and biochronology, paleoecology, climate and evolution and phylogenetic hypotheses. The Supplementary text provides more details that the interested researcher will value.  The parallel numerical ordering of the main and supplementary texts are intended to be a ready guide to the reader.  The numerical headings are very much like that were adopted in the Bernor et al. 1997 Hoewenegg (Hegau, Germany, 10.3 Ma) monograph which likewise help the reader navigate the various sections.

The difficulty of the geographic organization is particularly evident in less robust descriptions of transitional Ares like the Southern Levant, which has both African taxa and European and Eurasian ones. I do not want to go through the complete list of species. Still, I wanted to point out several omissions of some species, which suggest that some identification of the geographic and chronological range of species needs to be amended.

Reply: We thank the reviewer for this comment. We have added the information for the taxa found in the Levant, such as Equus stenonis and Equus tabeti. Considering that these species have been described in Europe and Africa, we did not create a new area such as the Southern Levant. We are describing species, not localities, as in the case of ‘Ubeidiya, Bizat Ruhama and Qafzeh (Levant).  

  1. Equus tabeti is found also in Bitzat Ruhama (1.0 Ma, Yeshurun et al 2011) and Qafzeh (120,000; Tchernov 2002). The 'Ubeidiya specimens were identified as Equus altidens in Gaudzinski 2004; in fact, a multivariate analysis has suggested similarities between the equids from the E. numerous–E tabeti lineage and Equus cf. altidens (Guerreo-Alba and Palmqvist, 1997). This lineage has been named "simplicidens" and includes E. numidicus, E. tabeti, E. altidens, and Equus granatensis. This lineage has also been found in Spain (Venta Micena, Orce, Cúllar de Baza, Cueva Victoria, Huèscar-1), Italy (Pirro Nord, Selvella), France (Sainzelles), and Germany (Süssenborn) (Arribas and Palmqvist, 1999). There has been a discussion with some suggesting an African origin for the species (Gibert et al., 2016), while others offer an Asian origin (Madurell-Malapeira et al., 2014).

Reply: We thank the reviewer for this comment. This information has been included in the sections on Equus tabeti and Equus altidens, respectively. However, no information about a possible Asian origin of Equus altidens has been proposed by Madurell-Malapeira et al. (2014), where the authors say: “The beginning of the Late Villafranchian represents a major faunal renewal, which involved the extinction of several spe­cies (most of them herbivores) as well as the arrival of several new forms of Asian and African origin. Several taxa with a long record during the Villafranchian, such as Stephanorhi­nus etruscus, Equus stenonis, Gazellospira, Gazella, Cer­vus phillisi, Arvernoceros ardei and Croizetoceros ramosus, vanished from the Iberian Peninsula around the Middle-Late Villafranchian boundary. These taxa were replaced by new-coming species, most of them of Asian origin, namely Steph­anorhinus hundsheimensis, Equus altidens, Praemegaceros verticornis, Hemibos, Hemitragus and Soergelia minor, most of them being first recorded at Venta Micena (ca. 1.6-1.4 Ma; Palmqvist et al., 2005)”. The wording “most of them” used by the authors does not necessarily lead to identifying an Asian origin of Equus altidens, and it was probably referred to the bovids. However, a new paragraph has been included in text, based on the latest report of Bernor et al. 2021 on Equus altidens from Dmanisi.

  1. Equus melkiensis also is found in sites of Gesher Benot Ya'akov (Eisenmann 2006, 2012b), Nahal Hesi (Yeshurun et al. 2011), Oumm Qatafa (Middle Plesitocene, Marom, et al., 2022) as well as Yemen and Tajikistan (Eisenmann 2006)

Reply: We thank the reviewer for this comment. We have included this information regarding Equus melkiensis in the text.

  1. Equus mauritanicus is also found in Oumm Qatafa (Marom et al., 2022)

Reply: We thank the reviewer for this comment. We have included this information regarding Equus mauritanics in the text.

  1. E. valeriani is not discussed (Eisenmann, 2002)

Reply: We thank the reviewer for this comment. Equus valeriani was described from Gromova (1946) from Samarkand, Uzbekistan. We have added a new section in Chapter 4.1 (Eastern and Central Asia).

  1. The second issue I find is the analysis of paleoecology. While the study focused on using the NOW database to retrieve data, additional data could have been added to the research that is not available in NOW (e.g., Equus tabeti in 'Ubeidiya, Belmaker and O'Brien 2018). However, a more critical analysis of mesowear has been shown to have a phylogenetic signal, and all linear correlations should be analyzed using phylogenetic contrasts (Belmaker and O'Brien, 2018). This is particularly evident in figure 2, where the regression is pulled by one outlier and should not be viewed as linear regression.

Reply: We sincerely thank the reviewer for these notes on the paleoecology section. Indeed, much of the data used in the paleoecology section is not derived from the NOW database but is instead based on literature (which are cited in the text and in Supplementary Table 2), complemented by direct mesowear scoring and osteological measurements for body mass estimates in museum collections. This aspect of the study is explained in the materials and methods -section, but we note that some sources of data could have been more clearly indicated. We have amended this by adding full reference list in the end of Supplementary Table 2. Note that the mesowear data that is new in this study are indicated as such in the reference column of Supplementary Table 2.

As for the second concern brought up by the reviewer, we acknowledge that there is a phylogenetic signal in mesowear (e.g., Fraser et al. (2018)), but we argue that this only reflects the fact that diets of large mammals are also similarly phylogenetically correlated, which makes sense as mesowear gives a signal of dietary composition (see Fraser et al. (2018a), DeSantis et al. (2018) reply to their study, and Fraser et al. (2008b) response to them). Furthermore, despite the phylogenetic signal in mesowear, it has been shown by several studies that mesowear can reflect drastic differences in dietary composition also within a species, and it is thus clearly primarily controlled by diet and is not biased by phylogenetic relatedness. A powerful example of this is the study by Clauss et al. (2007) who show that while mesowear classifies wild populations of extant giraffes as browsers, grass-fed zoo populations are classified as grazers by the mesowear analysis. Thus, we argue that there is no need to correct the mesowear analyses by phylogenetic contrasts, and indeed doing so introduces a risk of altering some of the true signal of dietary variation (see DeSantis et al., 2018). Furthermore, in some cases the correlations between body mass and mesowear that we present are significant even within species, not only within the genus Equus (see mesowear-body mass regression of E. stenonis in Fig. 1). The fact that we can get such a signal even within a species further demonstrates that the relationship between body size and diet is not phylogenetically constrained. Clauss (2019) brings a further point to the discussion, noting that similar diets could leave a different trace in the dental wear of mammals with different dental morphology. However, that is certainly not the case in our analyses for the genus Equus, as the differences in masticatory apparatus, digestive system and dentition are very subtle within this genus, and for this reason differences in mesowear signal are certainly directly comparable between all species of Equus.

Having said this, we do acknowledge that all the within-species regressions between body mass and mesowear are not statistically significant (probably due to small sample sizes) and should thus be taken as tentative results, but the fact that they show similar direction of the relationship between mesowear and body mass as the genus-level analyses suggest that these patterns do, by and large, apply to intraspecific patterns as well. We do agree with the reviewer that there is no significant linear regression for the African Equus in Figure 2, and indeed the “regression line” in the figure is tentative and should be taken only as showing the one large species having more mixed-feeding mesowear signal than the rest of the species. As Clauss (2019) puts it, lack of statistical significance does not automatically mean that such patterns would not be biologically meaningful (and in our case, they are supported by statistically significant results based on more extensive datasets).

References:

Clauss, M., Franz-Odendaal, T.A., Brasch, J., Castell, J.C., and Kaiser, T. 2007. Tooth wear in captive giraffes (Giraffa camelopardalis): mesowear analysis classifies free-ranging specimens as browsers but captive ones as grazers. Journal of Zoo and Wildlife Medicine 38, 433-445.

Clauss, M. 2019. Phylogenetic signal in tooth wear? A question that can be answered - By testing. Ecology and Evolution 9, 6170–6171.

DeSantis, L., Fortelius, M., Grine, F. E., Janis, C., Kaiser, T. M., Merceron, G., Purnell, M.A.,

Schulz-Kornas, E., Saarinen, J., Teaford, M., Ungar, P.S., ŽliobaitÄ—, I. 2018. The phylogenetic signal in tooth wear: What does it mean? Ecology and Evolution, 8, 11359–11362.

Fraser, D., Haupt, R.J., Barr, W.A. 2018a. Phylogenetic signal in tooth wear dietary niche proxies. Ecology and Evolution, 8, 5355–5368.

Fraser, D., Haupt, R.J., Barr, W.A. 2018b. Phylogenetic signal in tooth wear dietary niche proxies: What it means for those in the field. Ecology and Evolution 8, 11363–11367.

Reviewer 3 Report

The paper "Evolution of the Family Equidae, Subfamily Equinae, in North, Central and South America, Eurasia and Africa during the Plio-Pleistocene" is a high-quality contribution dedicated to the taxonomical diversity, evolution, paleobiogeography, and paleoecology of horses and hippariones in their broad sense from the World. The study is based on extensive fossil and osteological material and is performed by using a combination of several methodological approaches that help tp reveal the complex picture of horses diversity and evolution. The authors propose a bref but comprehensive taxonomic revision of equid species, ass well as some poorly understood forms. The applied meta-analysis approach seems to be the most suitable in this case, knowing the specifics of equid taxonomy issues. The proposed global description of paleobiogeography is the first synthesis of this kind. The discussion of the taxonimical concept of the genus Equus is another interesting and valuable point of the article. The bibliographic sources are complete and up-to-date, the overall concept of paper is well-designed. No doubts, that this paper will be a reference work for the studies of equid evolution, paleoecology, and systematics.

I am happy to recommend this paper in "Biology".

There are just few minor points to correct:

1. Page 4, line 140: "during several museum visits..." - it is necessary to indicate the visited collections. If the list tis too long, you may arrange them as a table in the supplimentary materials.

2. Page 4, line 157: "Because of slight methodological differences..." - a reference or references that provide the evidences on methodological differences are needed here.

3. In the paragraph 4, the terms "holotype" and "lectotype" are capitalized. These terms should be written with a lowercase letter.

4. Page 14, line 647: "Pleistocene species erected based on complete maxillary" - remove the word "erected".

5. Page 16, line 724: "Another extinct species was recognized..." - apparently, you are talking about subspecies here.

6. Page 16, line 744 : "Kuzmina [129] also gives early synonyms of E. caballus and E. ferus" - This is not very clear for me. Does Kuzmina regards E. caballus and E. ferus as synonyms of E. prewalskii? Please, reformulate this statement.

7. Page 17, line 760: Cormohipparion in italics.

8. Page 17, line 779: Cremohipparion in italics.

9. Page 17, line 785: "study currently in prep..." - in preparation.

10. Page 20, lines 928-929: "identifired in Moldavia" - should be "identified (remove "r") and "Moldova".

11. Page 22, line 1016: "subspecies were based..." - use "taxa" instead of "subspecies", since your list also contains several species names.

12. Page 46, line 2054: "E. hemionous" - hemionus (remove "o").

13. Page 47, line 2101: "[3, 411, 55, 412, 58]" - the reference should follow the order in which they appear in the text.

14. Missing references: 145. Spasskaya et al. 2021; 159. Stewart et al. 2019; 164. Poliakov, 1881; 183. Patel et al. 2017.

Author Response

Reviewer 3:

The paper "Evolution of the Family Equidae, Subfamily Equinae, in North, Central and South America, Eurasia and Africa during the Plio-Pleistocene" is a high-quality contribution dedicated to the taxonomical diversity, evolution, paleobiogeography, and paleoecology of horses and hippariones in their broad sense from the World. The study is based on extensive fossil and osteological material and is performed by using a combination of several methodological approaches that help to reveal the complex picture of horses diversity and evolution. The authors propose a brief but comprehensive taxonomic revision of equid species, as well as some poorly understood forms. The applied meta-analysis approach seems to be the most suitable in this case, knowing the specifics of equid taxonomy issues. The proposed global description of paleobiogeography is the first synthesis of this kind. The discussion of the taxonimical concept of the genus Equus is another interesting and valuable point of the article. The bibliographic sources are complete and up-to-date, the overall concept of paper is well-designed. No doubts, that this paper will be a reference work for the studies of equid evolution, paleoecology, and systematics.

Reply: We would like to acknowledge the reviewer to provide us his comments. We have addressed all the comments made by the reviewers to improve the quality of the present manuscript.

I am happy to recommend this paper in "Biology".

There are just few minor points to correct:

  1. Page 4, line 140: "during several museum visits..." - it is necessary to indicate the visited collections. If the list is too long, you may arrange them as a table in the supplementary materials.

Reply: We thank the reviewer for this comment. A list of museum acronyms has been included in the supplementary materials.

  1. Page 4, line 157: "Because of slight methodological differences..." - a reference or references that provide the evidences on methodological differences are needed here.

Reply: We thank the reviewer for this comment. The sentence has been edited with a new reference and indication for the supplementary text.

  1. In the paragraph 4, the terms "holotype" and "lectotype" are capitalized. These terms should be written with a lowercase letter.

Reply: We thank the reviewer for this comment. The terms holotype and lectotype have been uniformed in lowercase in the whole text, as suggested.

  1. Page 14, line 647: "Pleistocene species erected based on complete maxillary" - remove the word "erected".

Reply: We thank the reviewer for this comment. This issue has been resolved as suggested.

  1. Page 16, line 724: "Another extinct species was recognized..." - apparently, you are talking about subspecies here.

Reply: We thank the reviewer for this comment. The word species has been changed with subspecies.

  1. Page 16, line 744 : "Kuzmina [129] also gives early synonyms of E. caballus and E. ferus" -This is not very clear for me. Does Kuzmina regards E. caballus and E. ferus as synonyms of E. prewalskii? Please, reformulate this statement.

Reply: We thank the reviewer for this comment. The sentence was deleted as misleading. Please check the new subchapter 4.1.26 on Equus przewalskii.

  1. Page 17, line 760: Cormohipparion in italics.

Reply: We thank the reviewer for this comment. This issue has been resolved as suggested.

  1. Page 17, line 779: Cremohipparion in italics.

Reply: We thank the reviewer for this comment. This issue has been resolved as suggested.

  1. Page 17, line 785: "study currently in prep..." - in preparation.

Reply: We thank the reviewer for this comment. This issue has been resolved as suggested.

  1. Page 20, lines 928-929: "identifired in Moldavia" - should be "identified (remove "r") and "Moldova".

Reply: We thank the reviewer for this comment. This issue has been resolved as suggested.

  1. Page 22, line 1016: "subspecies were based..." - use "taxa" instead of "subspecies", since your list also contains several species names.

Reply: We thank the reviewer for this comment. This issue has been resolved as suggested.

  1. Page 46, line 2054: "E. hemionous" - hemionus (remove "o").

Reply: We thank the reviewer for this comment. This issue has been resolved as suggested.

  1. Page 47, line 2101: "[3, 411, 55, 412, 58]" - the reference should follow the order in which they appear in the text.

Reply: We thank the reviewer for this comment. This issue has been resolved as suggested.

  1. Missing references: 145. Spasskaya et al. 2021; 159. Stewart et al. 2019; 164. Poliakov,1881; 183. Patel et al. 2017.

Reply: We thank the reviewer for this comment. These references have been included in the main text.

Round 2

Reviewer 2 Report

I appreciate the authors' response to my comments. However, the assumption that a review is a he is wrong :).